# Expressive Power of Implicit Models: Rich Equilibria and Test-Time Scaling

**Jialin Liu**
School of Data, Mathematical, and Statistical Sciences
University of Central Florida
Orlando, FL 32826
jialin.liu@ucf.edu

**Lisang Ding**
Department of Mathematics
University of California, Los Angeles
Los Angeles, CA 90095
lisangding@ucla.edu

**Stanley Osher**
Department of Mathematics
University of California, Los Angeles
Los Angeles, CA 90095
sjo@math.ucla.edu

**Wotao Yin**
Decision Intelligence Lab, DAMO Academy
Alibaba US
Bellevue, WA 98004
wotao.yin@alibaba-inc.com

## ABSTRACT

Implicit models, an emerging model class, compute outputs by iterating a single parameter block to a fixed point. This architecture realizes an infinite-depth, weight-tied network that trains with constant memory, significantly reducing memory needs for the same level of performance compared to explicit models. While it is empirically known that these compact models can often match or even exceed the accuracy of larger explicit networks by allocating more test-time compute, the underlying reasons are not yet well understood.

We study this gap through a non-parametric analysis of expressive power. We provide a strict mathematical characterization, showing that a simple and regular implicit operator can, through iteration, progressively express more complex mappings. We prove that for a broad class of implicit models, this process allows the model's expressive power to grow with test-time compute, ultimately matching a much richer function class. The theory is validated across four domains: imaging, scientific computing, operations research, and LLM reasoning, demonstrating that as test-time iterations increase, the complexity of the learned mapping rises, while the solution quality simultaneously improves and stabilizes.

## 1 INTRODUCTION

Many machine-learning tasks can be cast as learning a mapping $\mathcal{F}$ from input $\boldsymbol{x}$ to the desired output $\boldsymbol{y}_*$, i.e., $\boldsymbol{y}_* = \mathcal{F}(\boldsymbol{x})$. An emerging alternative is the *implicit models*: train an operator $\mathcal{G}$ whose fixed point matches the target, i.e., $\boldsymbol{y}_* = \mathcal{G}(\boldsymbol{y}_*, \boldsymbol{x})$ (Bai et al., 2019; El Ghaoui et al., 2021). At inference time, the fixed point is obtained via a root-finding solver. While advanced algorithms (e.g., Anderson acceleration or Broyden's method) exist, the canonical approach is the Picard iteration:

$$\boldsymbol{y}_1 = \mathcal{G}(\boldsymbol{y}_0, \boldsymbol{x}), \;\; \boldsymbol{y}_2 = \mathcal{G}(\boldsymbol{y}_1, \boldsymbol{x}), \;\; \boldsymbol{y}_3 = \mathcal{G}(\boldsymbol{y}_2, \boldsymbol{x}), \;\; \cdots, \tag{1}$$

and expect $\boldsymbol{y}_t(\boldsymbol{x}) \to \boldsymbol{y}_*(\boldsymbol{x}) = \mathcal{F}(\boldsymbol{x})$ for all $\boldsymbol{x}$. Rather than producing $\boldsymbol{y}_*$ in a single feed-forward pass, implicit models reach the target through gradual equilibrium-seeking updates. By tailoring the structure of $\mathcal{G}$, implicit models have shown strong results across many domains (e.g., imaging (Gilton et al., 2021), scientific computing (Marwah et al., 2023), generative modeling (Pokle et al., 2022; Geng et al., 2023), LLM reasoning (Geiping et al., 2025), etc.).

Behind these successes, the advantages of implicit models include: **(i)** they realize an infinite-depth, weight-tied network trainable with constant memory, which yields efficient training (Fung et al., 2022; Geng et al., 2021); **(ii)** they allow us to "implicitly bake in" domain constraints and structure (e.g., physics, geometry, safety), see Xie et al. (2022); Güngör et al. (2023); Oshin et al. (2024); and, *most surprisingly,* **(iii)** *they can often match or even exceed the accuracy of larger explicit networks by allocating more iterations* (Marwah et al., 2023; Wang et al., 2024; Geiping et al., 2025). Point

(i) stems from the weight-tied architecture and avoiding full back-propagation. Point (ii) arises from the inherently implicit nature of many real-world, equation-based constraints. In contrast, the mechanism underlying the surprising effectiveness of (iii) remains less well understood.

We study this through the lens of expressive power—the set of input–output maps a model family can represent. We ask two questions. First, as a baseline: **(Q1)** *Do implicit models (at least) match the expressive power of explicit ones?* Concretely, for a target map $\mathcal{F} : \boldsymbol{x} \mapsto \boldsymbol{y}_*$, does there always exist an implicit operator $\mathcal{G}$ such that the iterates of (1) satisfy $\boldsymbol{y}_t(\boldsymbol{x}) \to \mathcal{F}(\boldsymbol{x})$ for all $\boldsymbol{x}$? If yes, a more insightful question follows: **(Q2)** *Do implicit models offer an expressive advantage?* In particular, can a relatively *simple* implicit operator $\mathcal{G}$, through iteration, represent a *complex* explicit map $\mathcal{F}$? A positive answer to (Q2) would directly explain phenomenon (iii).

To our knowledge, these questions remain largely open. While universality has been touched upon in specific settings (Bai et al., 2019; Marwah et al., 2023) and separation results have demonstrated advantages over explicit models (Wu et al., 2024), a complete characterization of the representable function class of implicit models (and hence a direct answer to questions (Q1) and (Q2)) is still missing. Unlike studies focusing on infinite-width limits and kernel connections (Gao et al., 2022; Feng & Kolter, 2023; Ling et al., 2024), our work fills this gap from a *nonparametric, function-space perspective*, establishing that an implicit model's expressive power scales with test-time compute. (See Appendix K for broader contextual discussions.) Specifically:

- **Expressive boundary.** We identify locally Lipschitz mappings as a natural target class and prove: every such map $\mathcal{F}$ can be expressed as the fixed point of a "regular" (simple and well-behaved) $\mathcal{G}$, and conversely, every such fixed-point map is locally Lipschitz.
- **Emergent expressive power.** Our theory, combined with iterative solvers' dynamics, yields a new viewpoint on implicit models: the expressive power is progressively unlocked by iterations.
- **Validation across domains.** We validate our theory with case studies in a wide range of applications (e.g., image reconstruction, scientific computing, operations research, and LLM reasoning).

Note that, while explicit networks are capable of expressing locally Lipschitz target maps (Beneventano et al., 2021) by scaling up the model size, implicit models can scale expressivity with test-time compute (i.e., more iterations at inference), which **scales up test-time runtime but not parameters**.

## 2 MAIN RESULTS

We now return to (Q1): given a target map $\mathcal{F}$, does there exist an implicit operator $\mathcal{G}$ whose fixed-point iteration yields $\boldsymbol{y}_t(\boldsymbol{x}) \to \mathcal{F}(\boldsymbol{x})$? A naive construction answers "yes": define, for $0 < \eta < 1$,
$$\mathcal{G}(\boldsymbol{y}, \boldsymbol{x}) := (1 - \eta)\boldsymbol{y} + \eta\mathcal{F}(\boldsymbol{x}). \tag{2}$$
Then the fixed-point iteration reduces to $\boldsymbol{y}_t = (1 - \eta)\boldsymbol{y}_{t-1} + \eta\mathcal{F}(\boldsymbol{x})$, hence $\boldsymbol{y}_t - \mathcal{F}(\boldsymbol{x}) = (1 - \eta)(\boldsymbol{y}_{t-1} - \mathcal{F}(\boldsymbol{x}))$. As $0 < \eta < 1$, it holds that, for all $\boldsymbol{x}$, $\boldsymbol{y}_t(\boldsymbol{x}) - \mathcal{F}(\boldsymbol{x}) \to \boldsymbol{0}$ as $t \to \infty$.

However, (2) is merely a trivial averaging of $\boldsymbol{y}$ and $\mathcal{F}(\boldsymbol{x})$; learning such an implicit model is no different from learning $\mathcal{F}$ directly. This prompts the natural follow-up: is there any *nontrivial* implicit representation that is able to indicate the expressive benefits of implicit models?

**An illustrative example.** Let $\mathcal{F}(x) = 1/x$ on $[-1, 1] \setminus \{0\}$. This function is smooth (differentiable to any order) almost everywhere, but blows up near the singular point $x = 0$:
$$|\mathcal{F}(x)| = \left|\frac{1}{x}\right| \to \infty, \quad \left|\frac{\mathrm{d}\mathcal{F}}{\mathrm{d}x}\right| = \left|-\frac{1}{x^2}\right| \to \infty, \quad \text{as } x \to 0.$$
Neural networks approximating $1/x$ on $[-1, -\delta) \cup (\delta, 1]$ typically demands higher network complexity—i.e., increasing depth/width as $\delta \to 0$ to capture the growing steepness near the singularity (Telgarsky, 2017). If we adopt the naive implicit form (2), $\mathcal{G}(y, x) = (1 - \eta)y + \eta/x$, nothing is gained: the model still inherits the singular behavior $|\partial\mathcal{G}/\partial x| = \eta/x^2 \to \infty$.

What would be a nontrivial implicit representation in this setting? Instead of writing $(1/x)$ explicitly, we can regard it as the solution of the equation $xy - 1 = 0$ **(implicit representation)**. Inspired by this, we apply a fixed-point iteration to $xy - 1 = 0$: $\mathcal{G}(y, x) = y - \eta(xy - 1)$. Using the general scheme in (1), we have $y_t = y_{t-1} - \eta(xy_{t-1} - 1)$. Subtracting the true solution gives
$$y_t - \frac{1}{x} = y_{t-1} - \frac{1}{x} - \eta x \left(y_{t-1} - \frac{1}{x}\right) = (1 - \eta x)\left(y_{t-1} - \frac{1}{x}\right)$$
For any $0 < \eta < 1$ and any $x \in (0, 1]$, we have $0 < (1 - \eta x) < 1$ which implies $y_t \to 1/x$. (For $x < 0$, simply flip the stepsize sign, $\eta$ to $-\eta$.) This implicit formulation is much simpler and more elegant: the operator $\mathcal{G}(y, x) = y - \eta(xy - 1)$ has *no singularity* and *no blow-up*.

The example indicates: intuitively, an implicit representation can realize a complicated map with singularities via a much simpler, smoother update operator $\mathcal{G}$. Next, we make it precise: we formally define what we mean by "simple" versus "complex," and characterize—beyond the $1/x$ example—the class of target functions for which an implicit representation admits such a simple form.

**Definition 2.1** (Lipschitz continuity). Let $(\mathbb{X}, \| \cdot \|)$ and $(\mathbb{Y}, \| \cdot \|)$ be normed spaces, and let $\mathcal{Q} : \mathbb{X} \to \mathbb{Y}$. We say $\mathcal{Q}$ is *L-Lipschitz* (globally Lipschitz) on $\mathbb{X}$ if there exists $L > 0$ such that

$$\|\mathcal{Q}(\boldsymbol{x}_1) - \mathcal{Q}(\boldsymbol{x}_2)\| \ \leq \ L\,\|\boldsymbol{x}_1 - \boldsymbol{x}_2\| \quad \text{for all } \boldsymbol{x}_1, \boldsymbol{x}_2 \in \mathbb{X},$$

and the smallest such $L$ is the *Lipschitz constant* (or *Lipschitz modulus*), denoted as $\mathrm{Lip}(\mathcal{Q})$. If the Lipschitz constant $L < 1$, we say $\mathcal{Q}$ is *L-contractive* on $\mathbb{X}$. Given $\boldsymbol{x} \in \mathbb{X}$, we say $\mathcal{Q}$ is *locally Lipschitz at* $\boldsymbol{x}$ if there exists a neighborhood $\mathbb{U}$ of $\boldsymbol{x}$ on which $\mathcal{Q}$ is $L_{\mathbb{U}}$-Lipschitz continuous for some $L_{\mathbb{U}} > 0$. If $\mathcal{Q}$ is locally Lipschitz at every $\boldsymbol{x} \in \mathbb{X}$, we say $\mathcal{Q}$ is *locally Lipschitz on* $\mathbb{X}$.

Intuitively, Lipschitz continuity limits how quickly a function's value can change. When a function is differentiable, its Lipschitz modulus can be characterized by the norm of its first derivative via the mean-value theorem. For example, $\mathcal{F}(x) = 1/x$ is locally Lipschitz on $[-1, 1] \setminus \{0\}$ but not globally Lipschitz there, since $|\mathrm{d}\mathcal{F}/\mathrm{d}x| = 1/x^2$ is unbounded as $x \to 0$, causing local Lipschitz constants to blow up near the singularity. In contrast, the implicit update $\mathcal{G}(y, x) = y - \eta(xy - 1)$ has simple partial derivatives $|\partial \mathcal{G}/\partial x| = |\eta\, y|$ and $|\partial \mathcal{G}/\partial y| = |1 - \eta x|$ without singularity.

Locally Lipschitz mappings form a much richer class than globally Lipschitz ones. Typical examples (locally Lipschitz everywhere in their domains but not globally Lipschitz on the whole set) include: $\log x$ in $(0, 1]$, $\tan x$ in $\left(-\frac{\pi}{2}, \frac{\pi}{2}\right)$, $\sqrt{x}$ in $(0, 1]$, $\Gamma(x)$ in $\mathbb{R} \setminus \{0, -1, -2, \cdots\}$, etc.

For this reason, we refer to globally Lipschitz maps as "simple" operators and locally Lipschitz maps (which may exhibit large local slopes near certain inputs) as "complex." Next, we formally state our main result: identifying a broad family of target functions for which implicit representations provide such simple update operators while expressing complex fixed-point mappings.

**Assumption 2.2.** Let $\mathbb{X} \subset \mathbb{R}^d$ and $\mathcal{F} : \mathbb{X} \to \mathbb{R}^n$ be locally Lipschitz on $\mathbb{X}$.

We do NOT assume the domain $\mathbb{X}$ to be bounded, compact, closed, or connected. For instance, $\mathbb{X} = \mathbb{R} \setminus \{0\}$ excludes the singular point and permits $\mathcal{F}(x) = 1/x$ to blow up at the interior gap $x = 0$ while remaining locally Lipschitz on $\mathbb{X}$. Another example is $\mathbb{X} = \bigcup_{k \in \mathbb{Z}} \left(k\pi - \frac{\pi}{2}, k\pi + \frac{\pi}{2}\right)$, where $\mathcal{F}(x) = \tan x$ remains locally Lipschitz despite blowing up at the singularity points $\left\{k\pi + \frac{\pi}{2}\right\}_{k \in \mathbb{Z}}$.

We now formalize what we mean by "simple" update rules—namely, *regular implicit operators*.

**Definition 2.3** (Regular implicit operator). Let $\mathbb{X} \subset \mathbb{R}^d$ be bounded. An operator $\mathcal{G} : \mathbb{R}^n \times \mathbb{X} \to \mathbb{R}^n$ is *regular* if: (i) For any $\boldsymbol{y} \in \mathbb{R}^n$, the map $\boldsymbol{x} \mapsto \mathcal{G}(\boldsymbol{y}, \boldsymbol{x})$ is globally Lipschitz (w.r.t. $\boldsymbol{x}$) on $\mathbb{X}$, and the Lipschitz constant grows linearly w.r.t. $\|\boldsymbol{y}\|$, and (ii) For each $\boldsymbol{x} \in \mathbb{X}$, there exists $\mu(\boldsymbol{x}) \in (0, 1)$, the map $\boldsymbol{y} \mapsto \mathcal{G}(\boldsymbol{y}, \boldsymbol{x})$ is $\mu(\boldsymbol{x})$-contractive on $\mathbb{R}^n$, and $\mu(\boldsymbol{x})$ is continuous w.r.t. $\boldsymbol{x}$.

A regular $\mathcal{G}$ satisfies: (i) Fixing $\boldsymbol{y}$, $\mathcal{G}(\boldsymbol{y}, \cdot)$ *is globally Lipschitz in* $\boldsymbol{x}$, this makes it a "simple" operator, and (ii) Fixing $\boldsymbol{x}$, $\mathcal{G}(\cdot, \boldsymbol{x})$ *is contractive in* $\boldsymbol{y}$; by Banach's theorem, this yields a unique fixed point $\boldsymbol{y}_*(\boldsymbol{x})$ and guarantees that iterates of (1) converge to it: $\boldsymbol{y}_t(\boldsymbol{x}) \to \boldsymbol{y}_*(\boldsymbol{x})$. An example of regular $\mathcal{G}$ is the aforementioned $\mathcal{G}(y, x) = y - \eta(xy - 1)$ on $x \in (0, 1]$ with $0 < \eta < 1$. Moreover, regularity does not require joint Lipschitz properties. With this definition, we present our main results.

**Theorem 2.4** (Sufficiency). *Under Assumption 2.2, for any $\mathcal{F}$ there exists a* regular *implicit operator* $\mathcal{G} : \mathbb{R}^n \times \mathbb{X} \to \mathbb{R}^n$ *whose fixed-point map reproduces $\mathcal{F}$: $\mathrm{Fix}\big(\mathcal{G}(\cdot, \boldsymbol{x})\big) = \mathcal{F}(\boldsymbol{x})$ for all $\boldsymbol{x} \in \mathbb{X}$.*

**Theorem 2.5** (Necessity). *Let $\mathbb{X} \subset \mathbb{R}^d$ and let $\mathcal{G} : \mathbb{R}^n \times \mathbb{X} \to \mathbb{R}^n$ be regular. For every $\boldsymbol{x} \in \mathbb{X}$, $\mathcal{G}(\cdot, \boldsymbol{x})$ has a unique fixed point $\boldsymbol{y}_*$, and the fixed-point map $\boldsymbol{x} \mapsto \boldsymbol{y}_*(\boldsymbol{x})$ is locally Lipschitz on $\mathbb{X}$.*

Proofs are deferred to Appendix A. Theorem 2.4 provides an affirmative answer to (Q1) and (Q2) posed in the introduction. It proves that for any locally Lipschitz target $\mathcal{F}$ on a bounded domain, there exists a *regular* implicit operator $\mathcal{G}$, whose iterations converge to the target $\boldsymbol{y}_t(\boldsymbol{x}) \to \mathcal{F}(\boldsymbol{x})$ for all $\boldsymbol{x}$. This demonstrates that the expressive power of implicit models **not only matches** that of explicit models **but also provides a distinct expressive benefit**: *a relatively simple (regular) implicit representation can yield a complex fixed-point mapping.* Complementarily, Theorem 2.5 shows the boundary is tight: fixed points induced by any regular $\mathcal{G}$ are *necessarily* locally Lipschitz. Together, the two results give an exact expressivity characterization for regular implicit models.

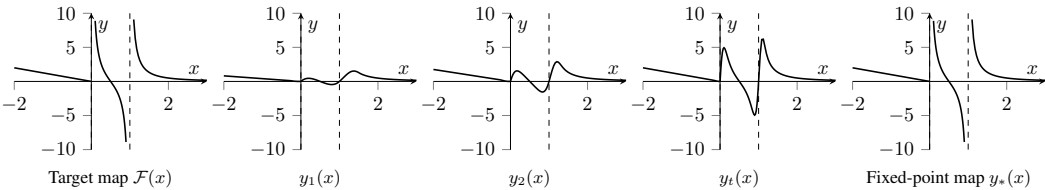

Figure 1: (Conceptual diagram) A simple implicit update expresses a complex map via iteration.

**What does our theory imply?** Take a locally Lipschitz target $\mathcal{F}$ (e.g., the curve in Fig. 1). Our results guarantee the existence of a *regular* implicit operator $\mathcal{G}$ such that the iteration $\boldsymbol{y}_t = \mathcal{G}(\boldsymbol{y}_{t-1}, \boldsymbol{x})$ with $\boldsymbol{y}_0 = \boldsymbol{0}$ converges: $\boldsymbol{y}_t(\boldsymbol{x}) \to \mathcal{F}(\boldsymbol{x})$. Consider the first iterate and its Lipschitz property:

$$\boldsymbol{y}_1(\boldsymbol{x}) = \mathcal{G}(\boldsymbol{0}, \boldsymbol{x}) \quad \Longrightarrow \quad \mathrm{Lip}(\boldsymbol{y}_1) = \sup_{\boldsymbol{x}, \boldsymbol{x}'} \frac{\|\mathcal{G}(\boldsymbol{0}, \boldsymbol{x}) - \mathcal{G}(\boldsymbol{0}, \boldsymbol{x}')\|}{\|\boldsymbol{x} - \boldsymbol{x}'\|} = \mathrm{Lip}(\mathcal{G}(\boldsymbol{0}, \cdot)).$$

Because a regular operator $\mathcal{G}$ is globally Lipschitz by definition, $\boldsymbol{y}_1(\cdot)$ is restricted to representing "simple," globally smooth mappings. However, as iterations progress, $\boldsymbol{y}_t$ converges toward $\mathcal{F}$. If the target $\mathcal{F}$ features singularities (regions where local slopes become large or unbounded), the effective Lipschitz constant of the iterate $\boldsymbol{y}_t(\cdot)$ naturally grows with $t$ to match that complexity:

$$\lim_{t \to \infty} \frac{\|\boldsymbol{y}_t(\boldsymbol{x}) - \boldsymbol{y}_t(\boldsymbol{x}')\|}{\|\boldsymbol{x} - \boldsymbol{x}'\|} = \frac{\|\mathcal{F}(\boldsymbol{x}) - \mathcal{F}(\boldsymbol{x}')\|}{\|\boldsymbol{x} - \boldsymbol{x}'\|}.$$

This dynamic highlights a fundamental distinction: while explicit networks scale *model size* to approximate locally Lipschitz targets (Beneventano et al., 2021), implicit models scale expressivity with *test-time compute*. A regular implicit operator can yield a complex equilibrium, and iterating this single operator at test time progressively unlocks that complexity *without adding parameters*.

**Generalization.** Someone may ask: does a large Lipschitz constant of the fixed-point map $\boldsymbol{y}_*(\boldsymbol{x})$ imply sensitivity or poor generalization (cf. Pabbaraju et al. (2021))? Our view is that this sensitivity is *inherent to the target $\mathcal{F}$*, not to the implicit representation—any faithful model, explicit or implicit, must track $\mathcal{F}$'s sharp variations. Our case studies in Section 3 confirm this: the target $\mathcal{F}$ in many tasks is indeed steep somewhere and the effective Lipschitz grows as accuracy improves. Crucially, the implicit representation can realize such targets with a *simple* operator $\mathcal{G}$, which regularizes training and supports good generalization in practice.

**Insights for practitioners.** A substantial line of work (e.g., El Ghaoui et al. (2021); Winston & Kolter (2020); Jafarpour et al. (2021); Revay et al. (2020); Havens et al. (2023)) enforces a global Lipschitz bound on the fixed-point map $\boldsymbol{y}_*(\boldsymbol{x})$. Typically, the model is parameterized as $\mathcal{G}(\boldsymbol{y}, \boldsymbol{x}) = \sigma(\boldsymbol{A}\boldsymbol{y} + \boldsymbol{B}\boldsymbol{x} + \boldsymbol{b})$, and by imposing specific algebraic structure on $\boldsymbol{A}$ and $\boldsymbol{B}$, one ensures that $\boldsymbol{y}_*(\boldsymbol{x})$ is globally Lipschitz in $\boldsymbol{x}$. While this indeed improves robustness, our theory shows it **constrains expressivity and undercuts the unique advantage of implicit models**. Our recommendation is different: rather than imposing uniform Lipschitz constraints, incorporate case-by-case *domain-specific knowledge, priors, or constraints* (as illustrated in our case studies Sec. 3). This method provides effective regularization, leading to robustness and strong test performance while unlocking the full power of implicit models—representing complex maps with relatively simple operators.

## 3 CASE STUDIES

In this section, we present four case studies. For the first three tasks, we (i) verify that the target satisfies Assumption 2.2; (ii) specify a domain-informed architecture for $\mathcal{G}$; (iii) confirm empirically that, under standard training without explicitly enforcing $\mathcal{G}$ to be regular, *the learned operators $\mathcal{G}$ exhibit these properties*—i.e., $\mathcal{G}$ is Lipschitz in $\boldsymbol{x}$ and iterates $\boldsymbol{y}_t$ converge (see Appendix F for training strategies and discussions regarding regularity guarantees); and (iv) demonstrate that expressive power scales with test-time iterations. Finally, we extend this analysis to LLM reasoning to validate our predictions in a domain where strict mathematical definitions are less applicable.

### 3.1 CASE STUDY 1: IMAGE RECONSTRUCTION (INVERSE PROBLEMS)

Inverse problems in imaging seek to recover an image $\boldsymbol{y}_* \in \mathbb{R}^n$ from partial, noisy measurements $\boldsymbol{x} = \boldsymbol{A}\boldsymbol{y}_* + \boldsymbol{n} \in \mathbb{R}^d$ $(d < n)$, where $\boldsymbol{A}$ is a known linear operator and $\boldsymbol{n}$ is noise. A common prior is that $\boldsymbol{y}_*$ lies near a smooth data manifold $\mathbb{M} \subset \mathbb{R}^n$. To recover $\boldsymbol{y}_*$, a standard estimator solves

$$\min_{\boldsymbol{y} \in \mathbb{R}^n} \ \frac{1}{2}\|\boldsymbol{x} - \boldsymbol{A}\boldsymbol{y}\|^2 \ + \ \frac{\alpha}{2} \mathrm{dist}^2(\boldsymbol{y}, \mathbb{M}), \tag{3}$$

or, equivalently, a variable–splitting surrogate

$$\min_{\boldsymbol{y},\boldsymbol{z}\in\mathbb{R}^n} \frac{1}{2}\|\boldsymbol{x} - \boldsymbol{A}\boldsymbol{y}\|^2 + \frac{\alpha}{2}\operatorname{dist}^2(\boldsymbol{z},\mathbb{M}) + \frac{\beta}{2}\|\boldsymbol{y} - \boldsymbol{z}\|^2. \tag{4}$$

Next we will show that, under mild assumptions, both (3) and (4) admit a unique minimizer for each $\boldsymbol{x}$ in a bounded set, and the solution map $\boldsymbol{x} \mapsto \hat{\boldsymbol{y}}(\boldsymbol{x})$ is *locally Lipschitz*. Hence the reconstruction target falls within Assumption 2.2 and is covered by our expressivity results in Section 2.

**Assumption 3.1.** Let $\mathbb{M} \subset \mathbb{R}^n$ be a compact, $\mathcal{C}^2$, embedded (possibly nonconvex) submanifold with positive reach $\tau > 0$. Assume the forward operator $\boldsymbol{A} : \mathbb{R}^n \to \mathbb{R}^d$ is $(\mu, L)$–bi-Lipschitz when restricted to $\mathbb{M}$ and let $\sigma_{\max}$ denote the maximal singular value of $\boldsymbol{A}$.

These assumptions are modest: they are standard in prior work and supported by existing theory. Formal definitions (reach and bi-Lipschitz continuity) and relevant literature appear in Appendix C.

**Definition 3.2.** Define the admissible set of observations $\boldsymbol{x}$ for (3) and (4):

$$\mathbb{X} := \left\{ \boldsymbol{x} : \boldsymbol{x} = \boldsymbol{A}\boldsymbol{y}_* + \boldsymbol{n}, \quad \text{for some } \boldsymbol{y}_* \in \mathbb{M}, \ \|\boldsymbol{n}\| < \frac{1}{80}\frac{\mu^5}{\sigma_{\max}^2 L^2}\tau. \right\}$$

**Theorem 3.3.** *Under Assumption 3.1, there exists $\alpha > 0$ for all $\boldsymbol{x} \in \mathbb{X}$ such that the minimization problem* (3) *yields a unique minimizer $\hat{\boldsymbol{y}}$. Let $\mathcal{F}_{1a}: \boldsymbol{x} \mapsto \hat{\boldsymbol{y}}$ denote the associated solution map from input $\boldsymbol{x}$ to the recovery $\hat{\boldsymbol{y}}$. Then $\mathcal{F}_{1a}$ is locally Lipschitz continuous on $\mathbb{X}$.*

**Theorem 3.4.** *Under Assumption 3.1, there exist $\alpha, \beta > 0$ for all $\boldsymbol{x} \in \mathbb{X}$ such that the minimization problem* (4) *yields a unique minimizer $(\hat{\boldsymbol{y}}, \hat{\boldsymbol{z}})$. Let $\mathcal{F}_{1b}: \boldsymbol{x} \mapsto \hat{\boldsymbol{y}}$ denote the associated solution map from input $\boldsymbol{x}$ to the recovery $\hat{\boldsymbol{y}}$. Then $\mathcal{F}_{1b}$ is locally Lipschitz continuous on $\mathbb{X}$.*

**Corollary 3.5.** *There must be a regular implicit operator $\mathcal{G}(\boldsymbol{y}, \boldsymbol{x})$ such that $\operatorname{Fix}(\mathcal{G}(\cdot, \boldsymbol{x})) = \mathcal{F}_{1a}(\boldsymbol{x})$ for all $\boldsymbol{x} \in \mathbb{X}$. The same conclusion holds for $\mathcal{F}_{1b}(\boldsymbol{x})$.*

Proofs of the theorems are deferred to Appendix C, and Corollary 3.5 follows immediately from Theorems 2.4, 3.3, and 3.4. This corollary guarantees the existence of regular implicit models $\mathcal{G}$ for image reconstruction. Next, we present how to implement $\mathcal{G}$ in this context.

**Problem-specific $\mathcal{G}$.** We adopt *algorithm-inspired* designs that mirror classical solvers for (3) and (4). Parameterizing these iterative solvers gives problem-tailored implicit models. In particular,

- Option I (PGD-style). To solve (3), if $\mathbb{M}$ were known, one would use proximal gradient descent (PGD): $\boldsymbol{y}_{t+1} = \operatorname{prox}_\sigma(\boldsymbol{y}_t - \gamma \boldsymbol{A}^\top(\boldsymbol{A}\boldsymbol{y}_t - \boldsymbol{x}))$, with parameters $\sigma, \gamma > 0$, where $\operatorname{prox}_\sigma$ is the proximal map of $(\sigma/2)\operatorname{dist}^2(\boldsymbol{y}, \mathbb{M})$ (see Appendix C.1). In practice, we replace $\operatorname{prox}_\sigma$ by a learnable neural network denoiser $\mathcal{H}_{\theta,\sigma}$ (parameters $\theta$ and noise level input $\sigma$) and obtain

$$\mathcal{G}_\Theta(\boldsymbol{y}, \boldsymbol{x}) = \mathcal{H}_{\theta,\sigma}\Big(\boldsymbol{y} - \gamma \boldsymbol{A}^\top(\boldsymbol{A}\boldsymbol{y} - \boldsymbol{x})\Big), \qquad \Theta = \{\theta, \sigma, \gamma\}. \tag{5}$$

- Option II (HQS-style). For (4), a standard solver is half–quadratic splitting (HQS, see Appendix C.2). Similar to Option I, we replace the proximal map by a learned module and obtain

$$\mathcal{G}_\Theta(\boldsymbol{y}, \boldsymbol{x}) = \mathcal{H}_{\theta,\sigma}\Big( (\boldsymbol{A}^\top\boldsymbol{A} + \beta\boldsymbol{I})^{-1}(\boldsymbol{A}^\top\boldsymbol{x} + \beta\boldsymbol{y}) \Big), \qquad \Theta = \{\theta, \sigma, \beta\}. \tag{6}$$

Here we adopt the long-standing "plug-in denoiser" idea from Venkatakrishnan et al. (2013), which replaces a proximal operator with an off-the-shelf denoiser inside an iterative solver; in our case, we train the *entire* $\mathcal{G}_\Theta$ as an implicit model. Details and bibliography are given in Appendix C.2.

**Experiment settings.** We study image deblurring, $\boldsymbol{x} = \boldsymbol{A}(\boldsymbol{y}_*) + \boldsymbol{n}$, where $\boldsymbol{A}$ is a motion-blur operator and $\boldsymbol{n}$ is additive Gaussian noise. Using BSDS500 (Martin et al., 2001), we construct 200 training, 100 validation, and 200 test pairs $(\boldsymbol{x}, \boldsymbol{y}_*)$, yielding datasets $\mathbb{D}_{\text{inv,train}}$, $\mathbb{D}_{\text{inv,val}}$, and $\mathbb{D}_{\text{inv,test}}$. Implementation details (data preprocessing, model choices, and training) are in Appendix G.

For evaluation, we analyze 100 iterations of the learned dynamics, $\boldsymbol{y}_{t+1}(\boldsymbol{x}) = \mathcal{G}_\Theta(\boldsymbol{y}_t(\boldsymbol{x}), \boldsymbol{x}), 0 \leq t \leq 99$ and $\boldsymbol{y}_0 = \boldsymbol{0}$, on the test set $\mathbb{D}_{\text{inv,test}} = \{(\boldsymbol{x}_i, \boldsymbol{y}_i^*)\}_{i=1}^{200}$. For each $i$, we create 5 perturbed ground truths $\boldsymbol{y}_{i,j}^*, 1 \leq j \leq 5$, and for each $\boldsymbol{y}_{i,j}^*$, we apply $\boldsymbol{A}$, add noise, and then obtain $\boldsymbol{x}_{i,j}$. The perturbed pairs $\{(\boldsymbol{x}_{i,j}, \boldsymbol{y}_{i,j}^*)\}_{i,j}$ form the perturbed dataset $\mathbb{D}'_{\text{inv,test}}$. Details appear in Appendix G. We track two metrics, including an empirical Lipschitz estimate and reconstruction quality in PSNR (i.e., Peak Signal-to-Noise Ratio, higher PSNR means more accurate reconstruction):

$$L_t := \max_{1 \leq i \leq 200} \max_{1 \leq j \leq 5} \frac{\|\boldsymbol{y}_t(\boldsymbol{x}_i) - \boldsymbol{y}_t(\boldsymbol{x}_{i,j})\|}{\|\boldsymbol{x}_i - \boldsymbol{x}_{i,j}\|}, \quad \text{and} \quad P_t(i,j) := \operatorname{PSNR}(\boldsymbol{y}_t(\boldsymbol{x}_{i,j}), \boldsymbol{y}_{i,j}^*),$$

for $1 \leq i \leq 200, 0 \leq j \leq 5$, where $j = 0$ means the original (unperturbed) sample, $\boldsymbol{x}_{i,0} := \boldsymbol{x}_i, \boldsymbol{y}_{i,0}^* := \boldsymbol{y}_i^*$. Here, $L_t$ estimates how complex the t-th iterate map $\boldsymbol{y}_t(\cdot)$ is, while $P_t$ measures the reconstruction quality on *both* the original dataset $\mathbb{D}_{\text{inv,test}}$ and the perturbed set $\mathbb{D}'_{\text{inv,test}}$.

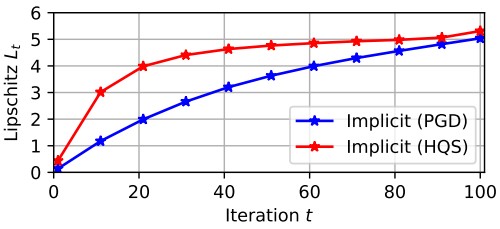
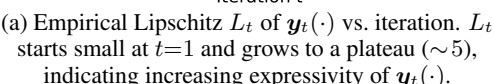
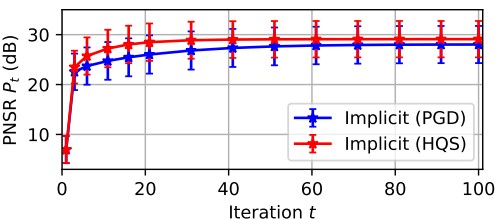

(a) Empirical Lipschitz $L_t$ of $\boldsymbol{y}_t(\cdot)$ vs. iteration. $L_t$ starts small at $t{=}1$ and grows to a plateau ($\sim 5$), indicating increasing expressivity of $\boldsymbol{y}_t(\cdot)$.

(b) Reconstruction quality $P_t$ (mean $\pm$ std over the original and perturbed test samples) increases and stabilizes: $\boldsymbol{y}_t(\boldsymbol{x})$ converges toward the truth.

Figure 2: Validation on image deblurring. Iterating a simple operator $\mathcal{G}_\Theta$ produces a complex fixed-point mapping: Lipschitz (a) grows, while accuracy (b) improves and stabilizes.

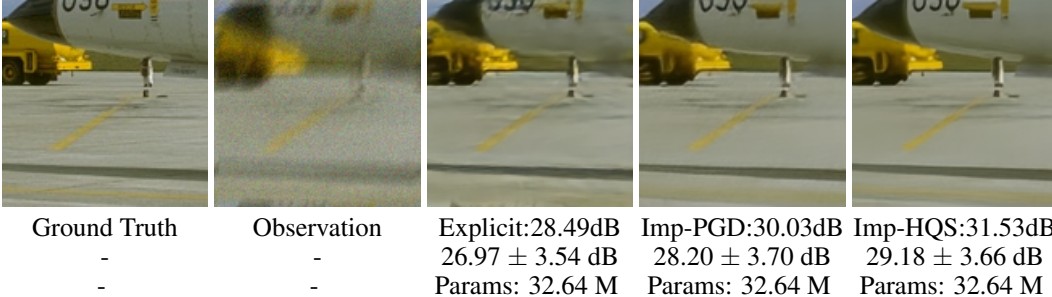

| Ground Truth | Observation | Explicit:28.49dB | Imp-PGD:30.03dB | Imp-HQS:31.53dB |
|---|---|---|---|---|
| - | - | $26.97 \pm 3.54$ dB | $28.20 \pm 3.70$ dB | $29.18 \pm 3.66$ dB |
| - | - | Params: 32.64 M | Params: 32.64 M | Params: 32.64 M |

Figure 3: Visual results for deblurring. The top PSNR values (28.49, 30.03, or 31.53 dB) correspond to the single visualized image; the second line shows the average ($\pm$ std) over all test samples.

**Experiment results.** *(i)* Results in Figure 2 support our theory. Figure 2a plots $L_t$ versus $t$, while Figure 2b reports the mean $\pm$ std of $\{P_t(i,j)\}_{i,j}$ versus $t$. At $t = 1$, the mapping $\boldsymbol{y}_1(\boldsymbol{x}) = \mathcal{G}_\Theta(\boldsymbol{0}, \boldsymbol{x})$ reflects a *single application of $\mathcal{G}_\Theta$ and exhibits low Lipschitz constant*: $L_1 = 0.140$ for PGD and $L_1 = 0.436$ for HQS. As $t$ increases, $\boldsymbol{y}_t$ approaches the fixed point and $L_t$ grows substantially, saturating around $\approx 5.0$ for both models (Figure 2a). Meanwhile, the PSNR rises and stabilizes, indicating that $\boldsymbol{y}_t(\boldsymbol{x})$ converges toward the ground truth (Figure 2b). Thus, the increase in $L_t$ does not reflect divergence or instability; rather, *it captures the greater complexity of the underlying target mapping $\boldsymbol{x} \mapsto \boldsymbol{y}_*$, which is progressively expressed through iteration. (ii)* We also provide a comparison (both visually and quantitatively) to an explicit model in Figure 3. This baseline uses the identical DRUnet and is trained on the deblurring dataset with an end-to-end MSE loss. A visual inspection reveals that implicit models, particularly implicit HQS (6), produce sharper images with better-recovered textures and fewer artifacts than the explicit baseline. This perceptual advantage is corroborated by the quantitative metrics, where the DEQ-HQS model achieves a significant PSNR gain of over 2dB on average across the entire test set. *(iii)* Additional experiments showing a small implicit model outperforming larger explicit ones appear in Appendix G.

### 3.2 Case Study 2: Scientific Computing

The Navier-Stokes (NS) equations are foundational to computational fluid dynamics. We focus on the 2D steady-state incompressible case on a periodic domain $\Omega := [0, 2\pi]^2$:

$$(u \cdot \nabla)u + \nabla p = \nu \Delta u + f, \quad \nabla \cdot u = 0 \quad \text{on } \Omega \tag{7}$$

where $u : \Omega \to \mathbb{R}^2$ is the velocity field, $p : \Omega \to \mathbb{R}$ is the pressure, $\nu > 0$ is the viscosity, $f : \Omega \to \mathbb{R}^2$ is the external force. Solving NS equations refers to determining $u$ given $f$. Although global existence/smoothness of the solution given general forcings is famously open, classical results guarantee well-posedness under suitable conditions on $f$.

**Theorem 3.6** (Temam (1995)). *There exists a constant $c > 0$ depending only on $\Omega$ such that, if $\|f\|_{L^2(\Omega)} \leq c\nu^2$, then (7) admits a unique solution $u_*(f)$. Let $\mathbb{H}$ denote the space of admissible forcings[1], and set $\mathbb{B}_\nu := \{f \in \mathbb{H} : \|f\|_{L^2(\Omega)} \leq c\nu^2\}$. Then there exists a subset $\mathbb{H}_\nu \subset \mathbb{B}_\nu$ that is dense in $\mathbb{B}_\nu$, on which the solution map $f \mapsto u_*(f)$ is locally Lipschitz.*

---

[1]Details regarding the function spaces are provided in Appendix D.

**Vorticity form.** Let $\omega := \nabla \times u$ (and hence $\omega_* := \nabla \times u_*$). Under periodic boundary and zero-mean conditions, one can recover the velocity $u$ from vorticity $\omega$ by solving a Poisson equation (Majda et al., 2002). We hence focus on the solution map in vorticity: $f \mapsto \omega_*$.

While Theorem 3.6 gives a local Lipschitz result in function spaces, our expressivity results (Section 2) are stated for finite-dimensional spaces. To bridge this gap, we discretize the NS equations.

**Discretization.** Partition $\Omega$ into $N_h$ cells $\Omega_h := \{C_i\}_{i=1}^{N_h}$ and define the cell–average restriction $\mathcal{R}_h(f)|_C := \frac{1}{|C|} \int_C f(\xi) \mathrm{d}\xi$ (similarly for $\omega$). We work with the discrete forcings and vorticities:

$$\boldsymbol{x} := \mathcal{R}_h(f) \in \mathbb{R}^{N_h \times 2}, \quad \boldsymbol{y} := \mathcal{R}_h(\omega) \in \mathbb{R}^{N_h}$$

and aim to learn $\boldsymbol{x} \mapsto \boldsymbol{y}_*$ where $\boldsymbol{y}_* := \mathcal{R}_h(\omega_*)$ is the discrete solution in vorticity form. Back to the continuum setting, let the lifting operator $\mathcal{E}_h$ be the piecewise–constant reconstruction $\mathcal{E}_h(\boldsymbol{x}) := \sum_{C \in \Omega_h} x_C \mathbf{1}_C$, and let $\mathcal{P}$ be the orthogonal projection onto divergence–free, zero–mean fields.

**Corollary 3.7.** $\mathcal{F}_2 : \boldsymbol{x} \mapsto \boldsymbol{y}_*$ *is locally Lipschitz on* $\mathbb{X}_{\nu,h} := \{\boldsymbol{x} \in \mathbb{R}^{N_h \times 2} : \mathcal{P}(\mathcal{E}_h(\boldsymbol{x})) \in \mathbb{H}_\nu\}$ *and there exists a regular implicit operator* $\mathcal{G}(\boldsymbol{y}, \boldsymbol{x})$ *satisfying* $\mathrm{Fix}(\mathcal{G}(\cdot, \boldsymbol{x})) = \mathcal{F}_2(\boldsymbol{x})$ *on* $\mathbb{X}_{\nu,h}$.

The corollary instantiates our expressivity theory for steady-state NS, guaranteeing the existence of a *regular* implicit model $\mathcal{G}$. As in the image–reconstruction case, we now (i) choose a problem-specific parameterization of $\mathcal{G}$ and (ii) verify our theory numerically on this architecture.

**Parameterization.** We use Marwah et al. (2023) as our code base: $\boldsymbol{z}_* = \mathcal{G}_\Theta(\boldsymbol{z}_*, \mathcal{Q}_\Phi(\boldsymbol{x}))$, and $\boldsymbol{y}_* = \mathcal{Q}_\Psi(\boldsymbol{z}_*)$. The core $\mathcal{G}_\Theta$ is implemented as a Fourier Neural Operator (FNO) (Li et al., 2021), and both the encoder $\mathcal{Q}_\Phi$ and decoder $\mathcal{Q}_\Psi$ use pointwise MLPs[2]. Details appear in Appendix H.

**Experiments.** We use the dataset of Marwah et al. (2023) with viscosity $\nu = 0.01$, which provides 4500 training pairs and 500 test pairs $(\boldsymbol{x}, \boldsymbol{y}^*)$, where $\boldsymbol{x}$ is the discretized force and $\boldsymbol{y}^*$ is the corresponding vorticity; we denote these sets by $\mathbb{D}_{\mathrm{pde,train}}$ and $\mathbb{D}_{\mathrm{pde,test}}$. Details are given in Appendix H.

We test iteration-wise behavior for 50 steps starting from $\boldsymbol{z}_0 = \boldsymbol{0}$: $\boldsymbol{z}_{t+1} = \mathcal{G}_\Theta(\boldsymbol{z}_t, \mathcal{Q}_\Phi(\boldsymbol{x}))$ for $0 \le t \le 49$, and $\boldsymbol{y}_t(\boldsymbol{x}) = \mathcal{Q}_\Psi(\boldsymbol{z}_t)$. Analogous to the inverse-problem study, we augment the test set with perturbations. For each $(\boldsymbol{x}_i, \boldsymbol{y}_i^*) \in \mathbb{D}_{\mathrm{pde,test}}$, we construct 15 perturbed vorticities $\{\boldsymbol{y}_{i,j}^*\}_{j=1}^{15}$; we then compute compatible forces $\{\boldsymbol{x}_{i,j}\}_{j=1}^{15}$ by evaluating the NS operator (see Appendix H for details). The perturbed test set is $\mathbb{D}'_{\mathrm{pde,test}} = \{(\boldsymbol{x}_{i,j}, \boldsymbol{y}_{i,j}^*) : 1 \le i \le 500, \ 1 \le j \le 15\}$. Across iterations we report an empirical Lipschitz estimate $L_t$ and relative reconstruction error $E_t$:

$$L_t := \max_{1 \le i \le 500} \max_{1 \le j \le 15} \frac{\|\boldsymbol{y}_t(\boldsymbol{x}_i) - \boldsymbol{y}_t(\boldsymbol{x}_{i,j})\|}{\|\boldsymbol{x}_i - \boldsymbol{x}_{i,j}\|}, \quad \text{and} \quad E_t(i,j) := \frac{\|\boldsymbol{y}_t(\boldsymbol{x}_{i,j}) - \boldsymbol{y}_{i,j}^*\|}{\|\boldsymbol{y}_{i,j}^*\| + \epsilon},$$

for $1 \le i \le 500, 0 \le j \le 15$, where $j = 0$ means the original (unperturbed) sample, $\boldsymbol{x}_{i,0} := \boldsymbol{x}_i, \boldsymbol{y}_{i,0}^* := \boldsymbol{y}_i^*$. Therefore, $E_t$ evaluates accuracy on both $\mathbb{D}_{\mathrm{pde,test}}$ and $\mathbb{D}'_{\mathrm{pde,test}}$.

The results in Figure 4 align with our theory. At $t = 1$, the mapping $\boldsymbol{y}_1(\boldsymbol{x})$ reflects a single application of $\mathcal{G}_\Theta$ and exhibits low Lipschitz constant: $L_1 = 23.1$. As iterations proceed toward the fixed point, the complexity grows markedly: $L_t$ increases to $\approx 367$ by $t = 50$ (Figure 4a). Meanwhile, the relative error $E_t$ decreases monotonically and stabilizes at $0.078 \pm 0.028$ (Figure 4b), indicating convergence to a good approximation of $\boldsymbol{y}_*$. Thus, *the learned operator $\mathcal{G}_\Theta$ is simple (Lipschitz in $\boldsymbol{x}$), while additional test-time iterations let $\boldsymbol{y}_t$ realize progressively more complex mappings.* In addition, a comparison with an explicit baseline (vanilla FNO) in Figure 5 shows the implicit model produces more accurate solutions, both visually and quantitatively. Additional experiments showing a small implicit model outperforming larger explicit ones appear in Appendix H.

### 3.3 CASE STUDY 3: OPERATIONS RESEARCH

Linear program (LP) is fundamental to operations research, of which a general form is given by

$$\min_{\boldsymbol{y} \in \mathbb{R}^n} \boldsymbol{c}^\top \boldsymbol{y}, \quad \text{s.t. } \boldsymbol{A}\boldsymbol{y} \circ \boldsymbol{b}, \ \boldsymbol{l} \le \boldsymbol{y} \le \boldsymbol{u}. \tag{8}$$

Here, $\boldsymbol{A} \in \mathbb{R}^{m \times n}, \boldsymbol{b} \in \mathbb{R}^m, \boldsymbol{c} \in \mathbb{R}^n, \boldsymbol{l} \in \mathbb{R}^n, \boldsymbol{u} \in \mathbb{R}^n$, and $\circ \in \{=, \le\}^m$ denotes componentwise relations, i.e., each $\circ_i \in \{=, \le\}$ specifies whether $(\boldsymbol{A}\boldsymbol{y})_i$ equals or is bounded above by $b_i$. Let $\boldsymbol{x} := (\boldsymbol{A}, \boldsymbol{b}, \boldsymbol{c}, \circ, \boldsymbol{l}, \boldsymbol{u})$ as the input that describes the LP in (8). In addition, we consider those feasible and bounded LPs (which admit an optimal solution (Bertsimas & Tsitsiklis, 1997)):

---

[2]Introducing additional encoder and decoder is common in practice. Compared to the vanilla formulation $\boldsymbol{y}_* = \mathcal{G}(\boldsymbol{y}_*, \boldsymbol{x})$, it does not change our expressivity results in Section 2. Details appear in Appendix B.

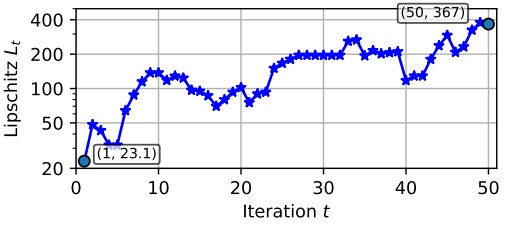
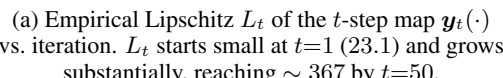
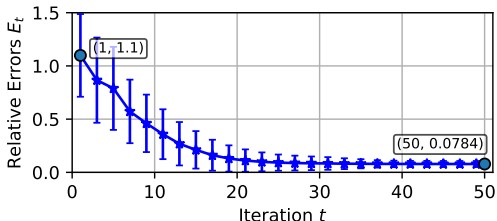

(a) Empirical Lipschitz $L_t$ of the $t$-step map $\boldsymbol{y}_t(\cdot)$ vs. iteration. $L_t$ starts small at $t{=}1$ (23.1) and grows substantially, reaching $\sim 367$ by $t{=}50$.

(b) Relative error $E_t$ (mean $\pm$ std over original and perturbed inputs) vs. iteration. $E_t$ decreases to $0.078 \pm 0.028$—$\boldsymbol{y}_t$ converges towards $\boldsymbol{y}_*$.

Figure 4: Validation on the steady Navier–Stokes task. Iterating a simple operator $\mathcal{G}_\Theta$ yields a complex fixed-point mapping: Lipschitz constant (a) increases, while error (b) decreases.

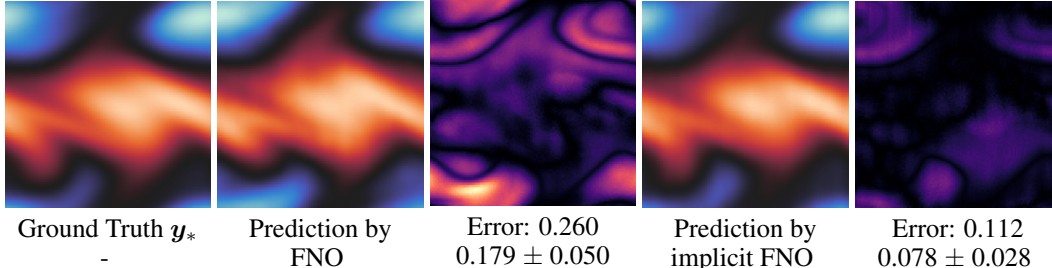

| Ground Truth $\boldsymbol{y}_*$ | Prediction by | Error: 0.260 | Prediction by | Error: 0.112 |
| - | FNO | $0.179 \pm 0.050$ | implicit FNO | $0.078 \pm 0.028$ |

Figure 5: Visual results for NS equations. The top value (0.260 and 0.112) means the relative error (between the prediction and the ground truth) on the single visualized sample; the second line shows the average relative error ($\pm$ std) over all test samples. Both models have 2.376 M parameters.

$$\mathbb{X} := \{(\boldsymbol{A}, \boldsymbol{b}, \boldsymbol{c}, \circ, \boldsymbol{l}, \boldsymbol{u}) : \text{The resulting LP is feasible and bounded}\}$$

Within $\mathbb{X}$, there are some LPs where the solution mapping is not single-valued or not continuous. By excluding these LPs, it forms a subset $\mathbb{X}_{\text{sub}} \subset \mathbb{X}$ on which $\mathcal{F}_3$ is single-valued and locally Lipschitz. The strict definition of $\mathbb{X}_{\text{sub}}$ and the proof of Theorem 3.8 are provided in Appendix E.

**Theorem 3.8.** *There is a subset $\mathbb{X}_{\text{sub}} \subset \mathbb{X}$ that is dense in $\mathbb{X}$, on which each LP admits a unique solution $\boldsymbol{y}_*$, and the solution map $\mathcal{F}_3 : \boldsymbol{x} \mapsto \boldsymbol{y}_*$ is locally Lipschitz continuous on $\mathbb{X}_{\text{sub}}$.*

**Corollary 3.9.** *There exists a regular implicit model $\mathcal{G}(\boldsymbol{y}, \boldsymbol{x})$ with $\text{Fix}(\mathcal{G}(\cdot, \boldsymbol{x})) = \mathcal{F}_3(\boldsymbol{x})$ on $\mathbb{X}_{\text{sub}}$.*

Corollary 3.9 follows immediately from Theorems 2.4 and 3.8. It indicates the existence of implicit models with desired properties that solves LP. As in the previous case studies, we now (i) choose a problem-specific parameterization of $\mathcal{G}$ and (ii) verify the theory numerically on this architecture.

**Implicit GNN parameterization.** We model the implicit operator $\mathcal{G}$ for LP with a *graph neural network (GNN)*. First, express an LP instance $\boldsymbol{x} = (\boldsymbol{A}, \boldsymbol{b}, \boldsymbol{c}, \circ, \boldsymbol{l}, \boldsymbol{u})$ as a bipartite graph (Figure 6). We create $n$ variable nodes $\{V_j\}_{j=1}^n$ and $m$ constraint nodes $\{W_i\}_{i=1}^m$. Node features collect the data of the LP: each $V_j$ stores $(c_j, l_j, u_j)$; each $W_i$ stores $(b_i, \circ_i)$. We connect $W_i$ to $V_j$ if $A_{ij} \neq 0$, and place $A_{ij}$ on that edge as its feature. Given this representation, an (explicit) GNN can map the LP to a solution, i.e., $\boldsymbol{y}_* = \text{GNN}(\boldsymbol{x})$ where $\boldsymbol{x}$ denotes the graph-encoded LP. This approach was proposed in Gasse et al. (2019), and Chen et al. (2023) subsequently showed that (explicit) GNNs offer a universal framework for representing LPs. Built on this, we propose an *implicit* GNN:

$$\boldsymbol{z}_* = \mathcal{G}_\Theta(\boldsymbol{z}_*, \mathcal{Q}_\Phi(\boldsymbol{x})), \quad \boldsymbol{y}_* = \mathcal{Q}_\Psi(\boldsymbol{z}_*) \tag{9}$$

where $\mathcal{G}_\Theta$ is the core GNN, $\mathcal{Q}_\Phi$ encodes instance-specific (static) features from $\boldsymbol{x}$, and $\mathcal{Q}_\Psi$ decodes per-variable states to the solution. Both $\mathcal{Q}_\Phi$ and $\mathcal{Q}_\Psi$ are small MLPs shared across all nodes. At inference, we repeatedly call $\mathcal{G}_\Theta$ with initialization $\boldsymbol{z}_0 = \boldsymbol{0}$: $\boldsymbol{z}_t = \mathcal{G}_\Theta(\boldsymbol{z}_{t-1}, \mathcal{Q}_\Phi(\boldsymbol{x}))$ for $t = 1, 2, \cdots, T$, and finally output $\boldsymbol{y}_t = \mathcal{Q}_\Psi(\boldsymbol{z}_t)$. Relative to prior work, our only modification is to attach to each variable node an additional *dynamic* state $z_j^{(t)} \in \mathbb{R}$. Details appear in Appendix I.

There is a rapidly growing literature on implicit GNNs with diverse applications and theories (Gu et al., 2020; Park et al., 2022; Chen et al., 2022a;b; Baker et al., 2023; Lin et al., 2024; Nastorg et al., 2024; Zhong et al., 2024; Yang et al., 2025). Our LP case study is complementary to this line of work: rather than adopting a particular implicit-GNN architecture, we start from a standard explicit GNN for LP and convert it into a fixed-point formulation tailored to linear programs.

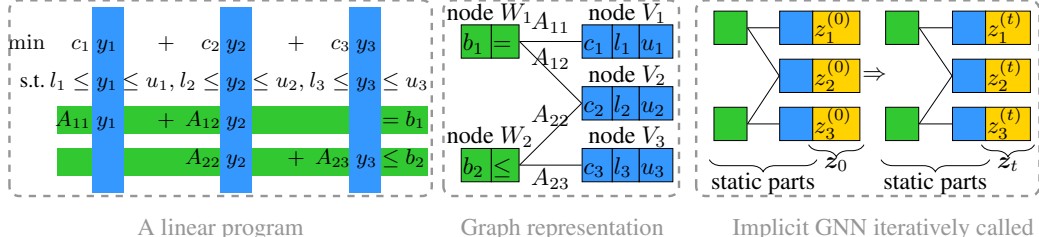

Figure 6: The graph representation of LP and implicit GNN applied on this graph

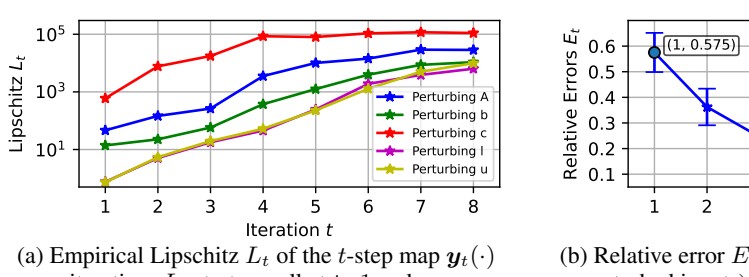

(a) Empirical Lipschitz $L_t$ of the $t$-step map $\boldsymbol{y}_t(\cdot)$ vs. iteration. $L_t$ starts small at $t=1$ and grows substantially by $t=8$ for all perturbation modes.

(b) Relative error $E_t$ (mean $\pm$ std over original and perturbed inputs) vs. iteration. $E_t$ decreases to $0.146 \pm 0.091$—$\boldsymbol{y}_t$ converges towards $\boldsymbol{y}_*$.

Figure 7: Numerical validation on the linear-program task.

**Experiments.** We sample LP instances $\boldsymbol{x} = (\boldsymbol{A}, \boldsymbol{b}, \boldsymbol{c}, \circ, \boldsymbol{l}, \boldsymbol{u})$, solve it to obtain an optimal solution $\boldsymbol{y}_*$, and form 2,500 training pairs and 1,000 test pairs like $(\boldsymbol{x}, \boldsymbol{y}_*)$, denoted $\mathbb{D}_{\text{LP,train}}$ and $\mathbb{D}_{\text{LP,test}}$. We also create five perturbed test sets $\{\mathbb{D}_{\text{LP,test}}^{(j)}\}_{j=1}^5$ by altering exactly one block among ($\boldsymbol{A}$, $\boldsymbol{b}$, $\boldsymbol{c}$, $\boldsymbol{l}$, or $\boldsymbol{u}$). For each $(\boldsymbol{x}_i, \boldsymbol{y}_i^*) \in \mathbb{D}_{\text{LP,test}}$ and each perturbation type $j$, we form a perturbed instance $\boldsymbol{x}_{i,j}$, solve it to obtain $\boldsymbol{y}_{i,j}^*$, and collect $\mathbb{D}_{\text{LP,test}}^{(j)} = \{(\boldsymbol{x}_{i,j}, \boldsymbol{y}_{i,j}^*)\}_{i=1}^{1000}$. Details in Appendix I. We report:

$$L_t(j) := \max_{1 \le i \le 1000} \frac{\|\boldsymbol{y}_t(\boldsymbol{x}_i) - \boldsymbol{y}_t(\boldsymbol{x}_{i,j})\|}{\|\boldsymbol{x}_i - \boldsymbol{x}_{i,j}\|}, \quad \text{and} \quad E_t(i,j) := \frac{\|\boldsymbol{y}_t(\boldsymbol{x}_{i,j}) - \boldsymbol{y}_{i,j}^*\|}{\|\boldsymbol{y}_{i,j}^*\| + \epsilon},$$

for $1 \le i \le 1000, 0 \le j \le 5$, where $j = 0$ denotes the unperturbed pair $(\boldsymbol{x}_{i,0}, \boldsymbol{y}_{i,0}^*) := (\boldsymbol{x}_i, \boldsymbol{y}_i^*)$. Results support our theory. *(i)* Figure 7a plots the five curves $L_t(j)$ (one for each perturbation type). At $t = 1$, a single application of (9) yields relatively small empirical Lipschitz constants for *all* perturbation modes. As iterations proceed toward the fixed point, Lipschitz constants grow markedly. *(ii)* Figure 7b reports the mean$\pm$std of $E_t(i,j)$: $E_t$ decreases and stabilizes at $0.146$, indicating that the growth of $L_t$ reflects the higher intrinsic complexity of the solution mapping $\boldsymbol{y}_*(\boldsymbol{x})$ rather than divergence or instability. *(iii)* Table 1 contrasts implicit and explicit GNNs. At matched embedding sizes, implicit GNNs match or beat explicit ones—most clearly at small/mid sizes (4/8/16). In addition, a smaller implicit model can outperform a larger explicit model on training error. For example, implicit–4 vs. explicit–8 (0.203 vs. 0.233) and implicit–8 vs. explicit–16 (0.162 vs. 0.183). This supports our theory that iterating a simple implicit operator can yield strong expressivity.

**Discussion on generalization.** While generalization is not our main focus, a trend in Table 1 is informative: explicit GNNs improve as width increases from 4 to 8 but then *overfit* (test error significantly rises at 16/32), whereas implicit GNNs improve from 4 to 8 to 16 and only tick up slightly at 32. We attribute this to: (i) LP constraints $\boldsymbol{Ay} \circ \boldsymbol{b}$ in (8) are specified implicitly rather than as an explicit set; implicit models align naturally with such a structure, and (ii) while fixed-point maps $\boldsymbol{y}_*(\boldsymbol{x})$

Table 1: Comparison between explicit GNNs and implicit GNNs on the LP task.

| | Emb. size | 4 | 8 | 16 | 32 |
|---|---|---|---|---|---|
| Exp-GNNs | # Params. | 580 | 2,088 | 7,888 | 30,624 |
| | Err (Train) | $0.387 \pm 0.103$ | $0.233 \pm 0.084$ | $0.183 \pm 0.070$ | $0.112 \pm 0.049$ |
| | Err (Test) | $0.397 \pm 0.107$ | $0.273 \pm 0.104$ | $0.283 \pm 0.111$ | $0.318 \pm 0.122$ |
| Imp-GNNs | Emb. size | 4 | 8 | 16 | 32 |
| | # Params. | 722 | 2,350 | 8,390 | 31,606 |
| | Err (Train) | $0.203 \pm 0.107$ | $0.162 \pm 0.094$ | $0.131 \pm 0.080$ | $0.118 \pm 0.073$ |
| | Err (Test) | $0.218 \pm 0.117$ | $0.177 \pm 0.105$ | $0.152 \pm 0.098$ | $0.156 \pm 0.109$ |

can be sensitive to inputs $x$, the implicit formulation allows us realize them via a simpler, smaller operator $\mathcal{G}$, which "implicitly" regularizes training and support good generalization in practice.

### 3.4 CASE STUDY 4: LLM REASONING

While previous case studies focused on domains with strict mathematical definitions (inverse problems, PDEs, LPs), we now investigate if our theory extends to broader applications where metrics like "smoothness" and "Lipschitz continuity" are less formally defined. We examine the looped transformer for LLM reasoning, utilizing the pre-trained model from Geiping et al. (2025). Unlike standard feed-forward transformers, this architecture recycles a shared block $\mathcal{G}_\Theta$ to iteratively update a latent "thought" vector $z$: $z_t = \mathcal{G}_\Theta(z_{t-1}, \mathcal{Q}_\Phi(x))$, $y_t = \mathcal{Q}_\Psi(z_t)$ where $\mathcal{Q}_\Phi$ encodes the input $x$, and $\mathcal{Q}_\Psi$ decodes the latent state into the output sequence $y_t$ obtained after $t$ recurrent blocks.

Strictly extending our Lipschitz theory to the discrete space of language tokens is challenging, as standard norms do not apply. However, we can empirically test the core prediction of our theory: can the model express increasingly complex mappings as iterations increase? In this context, complexity implies: *subtle differences in the input correspond to substantial shifts in context.* A capable model must effectively distinguish these semantic nuances and produce vastly different responses.

**Qualitative Results.** Table 2 visualizes the evolution of reasoning on a typical example. At early iterations ($t = 2, 4$), the model fails to differentiate context (Physics vs. Finance), producing repetitions or shallow associations. Conversely, with more iterations ($t = 6, 8$ or more), the model utilizes increased test-time compute to resolve this ambiguity, correctly defining "charge" as electric potential versus financial debt. This confirms the implicit operator's ability to progressively realize complex, context-sensitive mappings.

Table 2: Evolution of model outputs for the inputs "charge and voltage" vs. "charge and pay", showing how the semantic difference emerges over iterations.

| Input/Output 1 | Input/Output 2 | Comments |
|---|---|---|
| $x$: explain the difference between charge and **voltage**. | $x'$: explain the difference between charge and **pay**. | Prompts differ in a *single word*, but the context shifts from Physics to Finance. |
| $y_2(x)$: explain the difference between charge and voltage. | $y_2(x')$: explain the difference between charge and pay. | Both outputs merely echo inputs. |
| $y_4(x)$: explain the difference between charge and voltage. explain the difference between current and voltage. | $y_4(x')$: explain the difference between charge and pay. explain the difference between the two. | Outputs repeat the inputs with slight variations; contexts are not yet separated. |
| $y_6(x)$: The difference between voltage and charge is that voltage is the difference between the potential difference between two points, while charge is the difference between the electric field between two points. | $y_6(x')$: explain the difference between charge and pay. Explain the difference between charge and pay. | Separation begins: Output 1 moves into a Physics explanation (potential difference), while Output 2 still stays near the prompt. |
| $y_8(x)$: The difference between charge and voltage is that voltage is the difference in electric potential between two points in an electric field, while charge is the amount of electric charge in a system. | $y_8(x')$: Charge is the amount of money that a person owes to a company or organization. Pay is the amount of money that a person receives from a company or organization. | Full separation: Output 1 gives a Physics definition; Output 2 correctly adopts the Financial interpretation (money owed vs. received). |
| $y_{32}(x)$: Charge is the amount of electric charge present in a body. Voltage is the difference in electric potential between two points. | $y_{32}(x')$: Charge is the amount of money that a person or company owes to a credit card company. Pay is the amount of money that a person or company has paid to the credit card company. | Refinement: both domains have stable, concise, and accurate definitions specialized to Physics versus Finance. |

**Quantitative Results.** To quantify this effect, we define an empirical Lipschitz measure based on Levenshtein distance over paired prompts that differ by 1–2 words, and track its geometric mean across iterations. Details are given in Appendix J. The observed rise with $t$ indicates increasing sensitivity to subtle input changes, consistent with progressively sharper context separation.

## 4 CONCLUSIONS AND FUTURE DIRECTIONS

We have provided a strict characterization of the representational capacity of regular implicit models. Our analysis reveals that iterating a simple operator allows the model to progressively realize increasingly complex mappings, ultimately covering the entire class of locally Lipschitz functions. This theory is validated through four diverse case studies, showing that the empirical Lipschitz constant rises alongside solution quality. Our codes are available online [3].

---

[3] Available at: `https://github.com/liujl11git/IMP-Power`

ACKNOWLEDGMENTS

S. Osher was supported by DARPA HR00112590074, DoE DE-SC0026262. and NSF 2208272.

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

# APPENDIX

## CONTENTS

# A  PROOFS OF MAIN RESULTS

The core intuition behind our proofs is an extension of the $1/x$ example discussed in the introduction.

For Theorem 2.4 (Sufficiency), we construct the implicit operator $\mathcal{G}$ as a dynamic interpolation: $\mathcal{G}(\boldsymbol{y}, \boldsymbol{x}) = (1 - \varepsilon(\boldsymbol{x}))\boldsymbol{y} + \varepsilon(\boldsymbol{x})\mathcal{F}(\boldsymbol{x})$, which iteratively pulls the state $\boldsymbol{y}$ toward the target $\mathcal{F}(\boldsymbol{x})$ with a step size $\varepsilon(\boldsymbol{x})$. The key theoretical innovation is making this step size adaptive: we construct $\varepsilon(\boldsymbol{x})$ to be inversely proportional to the local steepness (Lipschitz constant) of the target $\boldsymbol{y}_*(\boldsymbol{x})$. In regions where the target function becomes extremely steep or singular (like $x \to 0$ for $1/x$), our constructed $\varepsilon(\boldsymbol{x})$ naturally vanishes. This effectively "slows down" the dynamics, ensuring the operator $\mathcal{G}$ itself remains globally smooth and contractive.

Theorem 2.5 (Necessity) establishes the converse: we show that for any regular operator, the local steepness of the fixed point is mathematically bounded by the operator's parameters ($y$-contraction modulus $\mu(\boldsymbol{x})$); and the fixed point map $\boldsymbol{y}_*(\boldsymbol{x})$ can only become singular if the convergence rate slows down (contraction modulus $\to 1$), perfectly matching the mechanism used in our sufficiency construction.

## A.1  PROOF OF SUFFICIENCY

*Proof of Theorem 2.4.* Given any $\mathcal{F}$ satisfying Assumption 2.2, the existence of $\mathcal{G}$ is proved by the following construction:

$$\mathcal{G}(\boldsymbol{y}, \boldsymbol{x}) = \mathcal{F}(\boldsymbol{x}) + (1 - \varepsilon(\boldsymbol{x}))(\boldsymbol{y} - \mathcal{F}(\boldsymbol{x})). \tag{10}$$

The proof will be done by choosing a function $\varepsilon : \mathbb{X} \to \mathbb{R}$ such that

- Functions $\varepsilon(\boldsymbol{x})$ and $\varepsilon(\boldsymbol{x})\mathcal{F}(\boldsymbol{x})$ are both globally Lipschitz continuous on $\mathbb{X}$.
- $0 < \varepsilon(\boldsymbol{x}) < 1$ for any $\boldsymbol{x} \in \mathbb{X}$.

The existence of such a $\varepsilon$ function is deferred to Theorem A.4. Now let's suppose such a $\varepsilon(\boldsymbol{x})$ is given and finish the whole proof. First let's check the contractivity of $\mathcal{G}$ in (10) as $\boldsymbol{x}$ fixed. For any $\boldsymbol{y}, \hat{\boldsymbol{y}} \in \mathbb{R}^n$, it holds that

$$\mathcal{G}(\boldsymbol{y}, \boldsymbol{x}) - \mathcal{G}(\hat{\boldsymbol{y}}, \boldsymbol{x}) = (1 - \varepsilon(\boldsymbol{x}))(\boldsymbol{y} - \mathcal{F}(\boldsymbol{x})) - (1 - \varepsilon(\boldsymbol{x}))(\hat{\boldsymbol{y}} - \mathcal{F}(\boldsymbol{x})) = (1 - \varepsilon(\boldsymbol{x}))(\boldsymbol{y} - \hat{\boldsymbol{y}}).$$

Since $0 < \varepsilon(\boldsymbol{x}) < 1$ for $\boldsymbol{x} \in \mathbb{X}$, we conclude that $\mathcal{G}(\cdot, \boldsymbol{x})$ is a contractor for $\boldsymbol{x} \in \mathbb{X}$. In addition, the continuity of the contractive factor $(1 - \varepsilon(\boldsymbol{x}))$ is directly resulted from the continuity of $\varepsilon(\boldsymbol{x})$. Finally, we check the Lipschitz continuity as $\boldsymbol{y}$ fixed. For any $\boldsymbol{x}, \hat{\boldsymbol{x}} \in \mathbb{X}$ and any $\boldsymbol{y} \in \mathbb{R}^n$, it holds that

$$\begin{aligned}
&\mathcal{G}(\boldsymbol{y}, \boldsymbol{x}) - \mathcal{G}(\boldsymbol{y}, \hat{\boldsymbol{x}}) \\
=& \Big(\mathcal{G}(\boldsymbol{y}, \boldsymbol{x}) - \boldsymbol{y}\Big) - \Big(\mathcal{G}(\boldsymbol{y}, \hat{\boldsymbol{x}}) - \boldsymbol{y}\Big) \\
=& \Big(\mathcal{F}(\boldsymbol{x}) - \boldsymbol{y} + (1 - \varepsilon(\boldsymbol{x}))(\boldsymbol{y} - \mathcal{F}(\boldsymbol{x}))\Big) - \Big(\mathcal{F}(\hat{\boldsymbol{x}}) - \boldsymbol{y} + (1 - \varepsilon(\hat{\boldsymbol{x}}))(\boldsymbol{y} - \mathcal{F}(\hat{\boldsymbol{x}}))\Big) \\
=& -\varepsilon(\boldsymbol{x})(\boldsymbol{y} - \mathcal{F}(\boldsymbol{x})) + \varepsilon(\hat{\boldsymbol{x}})(\boldsymbol{y} - \mathcal{F}(\hat{\boldsymbol{x}})) \\
=& \Big(-\varepsilon(\boldsymbol{x}) + \varepsilon(\hat{\boldsymbol{x}})\Big)\boldsymbol{y} + \Big(\varepsilon(\boldsymbol{x})\mathcal{F}(\boldsymbol{x}) - \varepsilon(\hat{\boldsymbol{x}})\mathcal{F}(\hat{\boldsymbol{x}})\Big)
\end{aligned}$$

With a fixed $\boldsymbol{y} \in \mathbb{R}^n$, the Lipschitz continuity of $\mathcal{G}(\boldsymbol{y}, \cdot)$ follows from the Lipschitz continuity of $\varepsilon(\boldsymbol{x})$ and $\varepsilon(\boldsymbol{x})\mathcal{F}(\boldsymbol{x})$. In particular, by denoting the Lipschitz constants of $\varepsilon(\boldsymbol{x})$ and $\varepsilon(\boldsymbol{x})\mathcal{F}(\boldsymbol{x})$ as $L_\varepsilon$ and $L_{\varepsilon\mathcal{F}}$ respectively, we have

$$\|\mathcal{G}(\boldsymbol{y}, \boldsymbol{x}) - \mathcal{G}(\boldsymbol{y}, \hat{\boldsymbol{x}})\| \leq L_\varepsilon \|\boldsymbol{x} - \hat{\boldsymbol{x}}\| \cdot \|\boldsymbol{y}\| + L_{\varepsilon\mathcal{F}} \|\boldsymbol{x} - \hat{\boldsymbol{x}}\| \leq \Big(L_\varepsilon \|\boldsymbol{y}\| + L_{\varepsilon\mathcal{F}}\Big) \|\boldsymbol{x} - \hat{\boldsymbol{x}}\|$$

where the Lipschitz constant of $\mathcal{G}$, $L := L_\varepsilon \|\boldsymbol{y}\| + L_{\varepsilon\mathcal{F}}$, grows linearly w.r.t. $\|\boldsymbol{y}\|$, which finishes the whole proof. $\square$

Below we provide the core theorems used in the proof of Theorem 2.4. We first consider $\mathbb{X}$ to be bounded (Theorem A.1) and then extend the results to the unbounded domain (Theorem A.4).

**Theorem A.1.** *For any $\mathcal{F}$ satisfying Assumption 2.2 defined on a bounded domain $\mathbb{X} \subset \mathbb{R}^d$, there exists a function $\varepsilon : \mathbb{X} \to \mathbb{R}$ such that $0 < \varepsilon(\boldsymbol{x}) < 1$ for $\boldsymbol{x} \in \mathbb{X}$, and $\varepsilon(\boldsymbol{x})$ and $\varepsilon(\boldsymbol{x})\mathcal{F}(\boldsymbol{x})$ are both globally Lipschitz continuous on $\mathbb{X}$.*

*Proof.* Let $\overline{\mathbb{X}}$ be the closure of set $\mathbb{X}$. In this proof, we will first extend $\mathcal{F}$ to $\overline{\mathbb{X}}$, construct the $\varepsilon$ function on $\overline{\mathbb{X}}$, and finally prove the global Lipschitz continuity of $\varepsilon(\boldsymbol{x})$ and $\varepsilon(\boldsymbol{x})\mathcal{F}(\boldsymbol{x})$ on $\overline{\mathbb{X}}$.

**Step 1: Extension to $\overline{\mathbb{X}}$.** First we extend $\mathcal{F}$ to $\bar{\boldsymbol{x}} \in \overline{\mathbb{X}} \backslash \mathbb{X}$ by the limit relative to $\mathbb{X}$:

$$\mathcal{F}(\bar{\boldsymbol{x}}) = \begin{cases} \lim\limits_{\mathbb{X} \ni \boldsymbol{x} \to \bar{\boldsymbol{x}}} \mathcal{F}(\boldsymbol{x}), & \text{if } \lim\limits_{\mathbb{X} \ni \boldsymbol{x} \to \bar{\boldsymbol{x}}} \mathcal{F}(\boldsymbol{x}) \text{ exists,} \\ 0, & \text{otherwise.} \end{cases}$$

Note that even if $\mathcal{F}$ is continuously extendable to $\bar{\boldsymbol{x}}$, it is still possible that $\mathcal{F}$ is not locally Lipschitz continuous at the point $\bar{\boldsymbol{x}}$. A simple example is the function $\sqrt{x}$, which is continuous as $x \geq 0$ and locally Lipschitz continuous for all points $x > 0$ but NOT locally Lipschitz at $x = 0$. We collect all these points (where $\mathcal{F}$ is not locally Lipschitz) into the set $\mathbb{D}(\mathcal{F})$:

$$\mathbb{D}(\mathcal{F}) := \left\{ \boldsymbol{x} \in \overline{\mathbb{X}} : \mathcal{F} \text{ is not locally Lipschitz continuous at } \boldsymbol{x} \right\}$$

For brevity, we will use $\mathbb{D}$ to denote $\mathbb{D}(\mathcal{F})$. It holds that $\mathbb{D}$ is a closed set (ref. to Lemma A.2) and $\mathbb{D} \subset \overline{\mathbb{X}} \backslash \mathbb{X}$.

**Step 2: Constructing a function** $\varepsilon : \overline{\mathbb{X}} \to \mathbb{R}_{\geq 0}$**.** Now let's define a set including all points that are very "safe", i.e., sufficiently far from the discontinuity set $\mathbb{D}$. In particular, given a positive real number $r > 0$, the set $\mathbb{D}_r$ is define by

$$\mathbb{D}_r := \left\{ \boldsymbol{x} \in \overline{\mathbb{X}} : d(\boldsymbol{x}, \mathbb{D}) \geq r \right\},$$

where $d(\boldsymbol{x}, \mathbb{D})$ means the distance of $\boldsymbol{x}$ and $\mathbb{D}$, and the closedness of $\mathbb{D}_r$ can be derived from the continuity of the distance function. Since $\mathbb{D}_r \subset \overline{\mathbb{X}}$ and $\overline{\mathbb{X}}$ is compact, $\mathbb{D}_r$ must be compact. Note that $\mathbb{D}_r$ and $\mathbb{D}$ are disjoint, hence $\mathcal{F}$ is locally Lipschitz continuous everywhere on $\mathbb{D}_r$. Thanks to the fact that local Lipschitz continuity on a compact set implies global Lipschitz continuity (ref to Lemma A.3), we can conclude that $\mathcal{F}$ is bounded and globally Lipschitz continuous on $\mathbb{D}_r$ for all $r > 0$. Therefore, the following two supremums exist, as long as the cardinality (number of elements) of $\mathbb{D}_r$ is large enough:

$$h_1(r) = \begin{cases} \sup\limits_{\boldsymbol{x}_1, \boldsymbol{x}_2 \in \mathbb{D}_r, \boldsymbol{x}_1 \neq \boldsymbol{x}_2} \dfrac{\|\mathcal{F}(\boldsymbol{x}_1) - \mathcal{F}(\boldsymbol{x}_2)\|}{\|\boldsymbol{x}_1 - \boldsymbol{x}_2\|}, & \text{card}(\mathbb{D}_r) \geq 2, \\ 0, & \text{otherwise.} \end{cases}$$

$$h_2(r) = \begin{cases} \sup\limits_{\boldsymbol{x} \in \mathbb{D}_r} \|\mathcal{F}(\boldsymbol{x})\|, & \text{card}(\mathbb{D}_r) \geq 1, \\ 0, & \text{otherwise.} \end{cases}$$

Here, both $h_1$ and $h_2$ are non-negative and monotone non-increasing on $(0, +\infty)$. Then we define:

$$\hat{h}(r) = \frac{1}{h_1(r) + h_2(r) + 1}.$$

It has the following properties:

- Bounded: $0 < \hat{h}(r) \leq 1$ as $r > 0$.

- Monotone : $\hat{h}(r_1) \leq \hat{h}(r_2)$ as $0 < r_1 \leq r_2$. (Due to the monotonicity of $h_1$ and $h_2$)

- Naturally extended to $r = 0$: $\lim_{r \to 0_+} \hat{h}(r)$ exists. (Due to the monotonicity of $\hat{h}$)

- $\hat{h}(r)h_i(r) < 1$ for $r \geq 0$ and $i = 1, 2$.

These properties implies that $\hat{h}$ is Riemann integrable on $[0, +\infty)$. Then we can define the following function:

$$\hat{\varepsilon}(r) := \int_0^r \hat{h}(s)\mathrm{d}s$$

with the following properties:

- $\hat{\varepsilon}(0) = 0$.

- Monotone increasing. This is a straightforward result of the fact that $\hat{h}(s) > 0$ for $s > 0$.

- Strictly positive as $r > 0$. This is also straightforward as $\hat{h}(s) > 0$ for $s > 0$.

- 1-Lipschitz continuous on $[0, +\infty)$. For any $r_1, r_2$ with $0 \leq r_1 \leq r_2 < +\infty$, we have

$$|\hat{\varepsilon}(r_1) - \hat{\varepsilon}(r_2)| = \hat{\varepsilon}(r_2) - \hat{\varepsilon}(r_1) = \int_{r_1}^{r_2} \hat{h}(s)\mathrm{d}s \leq \left(\sup_{r \geq 0} \hat{h}(r)\right)|r_1 - r_2| = |r_1 - r_2|.$$

With such a $\hat{\varepsilon}(r)$, we can define $\varepsilon(\boldsymbol{x})$ by

$$\varepsilon(\boldsymbol{x}) = \frac{\hat{\varepsilon}\Big(d(\boldsymbol{x}, \mathbb{D})\Big)}{1 + \hat{\varepsilon}\Big(d(\boldsymbol{x}, \mathbb{D})\Big)}.$$

It holds that $\varepsilon(\boldsymbol{x}) = 0$ for $\boldsymbol{x} \in \mathbb{D}$ and $0 < \varepsilon(\boldsymbol{x}) < 1$ for $\boldsymbol{x} \in \overline{\mathbb{X}} \backslash \mathbb{D}$. As $\mathbb{D} \subset \overline{\mathbb{X}} \backslash \mathbb{X}$, we have $0 < \varepsilon(\boldsymbol{x}) < 1$ for $\boldsymbol{x} \in \mathbb{X}$.

**Step 3: Establishing the Lipscthiz continuity.** Since the distance function $d(\boldsymbol{x}, \mathbb{D})$ is 1-Lipschitz continuous (Federer, 1959, Theorem 4.8 (1)), the Lipschitz continuity of $\hat{\varepsilon}$ implies the Lipschitz continuity of $\varepsilon$. In particular, for all $\boldsymbol{x}_1, \boldsymbol{x}_2 \in \overline{\mathbb{X}}$, it holds that

$$\left|\varepsilon(\boldsymbol{x}_1) - \varepsilon(\boldsymbol{x}_2)\right|$$

$$= \left|\frac{\hat{\varepsilon}\Big(d(\boldsymbol{x}_1, \mathbb{D})\Big)}{1 + \hat{\varepsilon}\Big(d(\boldsymbol{x}_1, \mathbb{D})\Big)} - \frac{\hat{\varepsilon}\Big(d(\boldsymbol{x}_2, \mathbb{D})\Big)}{1 + \hat{\varepsilon}\Big(d(\boldsymbol{x}_2, \mathbb{D})\Big)}\right| \quad \left(\frac{x}{1+x} \text{ is 1-Lipschitz as } \left(\frac{x}{1+x}\right)' = \frac{1}{(1+x)^2}\right)$$

$$\leq \left|\hat{\varepsilon}\Big(d(\boldsymbol{x}_1, \mathbb{D})\Big) - \hat{\varepsilon}\Big(d(\boldsymbol{x}_2, \mathbb{D})\Big)\right| \qquad \text{(Lipschitz continuity of } \hat{\varepsilon})$$

$$\leq \left|d(\boldsymbol{x}_1, \mathbb{D}) - d(\boldsymbol{x}_2, \mathbb{D})\right| \leq \|\boldsymbol{x}_1 - \boldsymbol{x}_2\| \qquad \text{(Lipschitz continuity of } d)$$

Therefore, to complete the whole proof, it's enough to show the global Lipschitz continuity of $\varepsilon\mathcal{F}$ on $\overline{\mathbb{X}}$. As $\overline{\mathbb{X}}$ is compact, and thanks to Lemma A.3, it's enough to show $\varepsilon\mathcal{F}$ is locally Lipschitz everywhere on $\overline{\mathbb{X}}$.

First, we consider the local Lipschitz continuity of $\varepsilon\mathcal{F}$ on $\overline{\mathbb{X}} \backslash \mathbb{D}$. Due to Lemma A.2, $\overline{\mathbb{X}} \backslash \mathbb{D}$ must be open relative to $\overline{\mathbb{X}}$. For any $\boldsymbol{x} \in \overline{\mathbb{X}} \backslash \mathbb{D}$, there must be a small enough $r > 0$ such that $\mathbb{U} := \mathbb{B}(\boldsymbol{x}, r) \cap \overline{\mathbb{X}} \subset \overline{\mathbb{X}} \backslash \mathbb{D}$. Pick $\boldsymbol{x}_1, \boldsymbol{x}_2 \in \mathbb{U}$. For any $\boldsymbol{x}_1, \boldsymbol{x}_2$, it holds that

$$\|\varepsilon(\boldsymbol{x}_1)\mathcal{F}(\boldsymbol{x}_1) - \varepsilon(\boldsymbol{x}_2)\mathcal{F}(\boldsymbol{x}_2)\|$$
$$= \|\varepsilon(\boldsymbol{x}_1)\mathcal{F}(\boldsymbol{x}_1) - \varepsilon(\boldsymbol{x}_1)\mathcal{F}(\boldsymbol{x}_2) + \varepsilon(\boldsymbol{x}_1)\mathcal{F}(\boldsymbol{x}_2) - \varepsilon(\boldsymbol{x}_2)\mathcal{F}(\boldsymbol{x}_2)\| \qquad (11)$$
$$\leq \varepsilon(\boldsymbol{x}_1)\|\mathcal{F}(\boldsymbol{x}_1) - \mathcal{F}(\boldsymbol{x}_2)\| + |\varepsilon(\boldsymbol{x}_1) - \varepsilon(\boldsymbol{x}_2)| \cdot \|\mathcal{F}(\boldsymbol{x}_2)\|.$$

Since both $\varepsilon$ and $\mathcal{F}$ are locally Lipschitz and locally bounded everywhere on $\overline{\mathbb{X}} \backslash \mathbb{D}$, they must be Lipschitz and bounded within $\mathbb{U}$. Then the local Lipschitz continuity of $\varepsilon\mathcal{F}$ at $\boldsymbol{x}$ immediately follows from (11). Note that $\boldsymbol{x}$ is arbitrarily picked from $\overline{\mathbb{X}} \backslash \mathbb{D}$, hence $\varepsilon\mathcal{F}$ is locally Lipschitz everywhere on $\overline{\mathbb{X}} \backslash \mathbb{D}$.

Next, we consider the local Lipschitz continuity of $\varepsilon\mathcal{F}$ on $\mathbb{D}$. For any $\boldsymbol{x} \in \mathbb{D}$, we consider its neighborhood $\mathbb{U} := \mathbb{B}(\boldsymbol{x}, 1) \cap \overline{\mathbb{X}}$ and pick $\boldsymbol{x}_1, \boldsymbol{x}_2 \in \mathbb{U}$. Then we need to consider three cases. The first case is both $\boldsymbol{x}_1, \boldsymbol{x}_2$ belong to the discontinuity set $\mathbb{D}$: $\boldsymbol{x}_1, \boldsymbol{x}_2 \in \mathbb{D}$. In this case, it holds that $\varepsilon(\boldsymbol{x}_1) = \varepsilon(\boldsymbol{x}_2) = 0$ and hence

$$\left\|\varepsilon(\boldsymbol{x}_1)\mathcal{F}(\boldsymbol{x}_1) - \varepsilon(\boldsymbol{x}_2)\mathcal{F}(\boldsymbol{x}_2)\right\| = 0 \leq \|\boldsymbol{x}_1 - \boldsymbol{x}_2\|.$$

The second case is that one of the point is in $\mathbb{D}$ while the other is not, we suppose $\boldsymbol{x}_1 \in \mathbb{D}, \boldsymbol{x}_2 \in \overline{\mathbb{X}} \backslash \mathbb{D}$, then

$$\left\|\varepsilon(\boldsymbol{x}_1)\mathcal{F}(\boldsymbol{x}_1) - \varepsilon(\boldsymbol{x}_2)\mathcal{F}(\boldsymbol{x}_2)\right\|$$

$$
\begin{aligned}
&= \left\| \varepsilon(\boldsymbol{x}_2)\mathcal{F}(\boldsymbol{x}_2) \right\| \leq \hat{\varepsilon}\Big(d(\boldsymbol{x}_2, \mathbb{D})\Big)\|\mathcal{F}(\boldsymbol{x}_2)\| \\
&= \left( \int_0^{d(\boldsymbol{x}_2, \mathbb{D})} \hat{h}(s)\mathrm{d}s \right) \|\mathcal{F}(\boldsymbol{x}_2)\| \\
&\leq \hat{h}(d(\boldsymbol{x}_2, \mathbb{D})) \cdot d(\boldsymbol{x}_2, \mathbb{D}) \cdot \|\mathcal{F}(\boldsymbol{x}_2)\| && \text{(Monontonicity of } \hat{h}) \\
&\leq \hat{h}(d(\boldsymbol{x}_2, \mathbb{D})) \cdot d(\boldsymbol{x}_2, \mathbb{D}) \cdot h_2(d(\boldsymbol{x}_2, \mathbb{D})) && \text{(Definition of } h_2) \\
&< d(\boldsymbol{x}_2, \mathbb{D}) && (\hat{h}(r) \cdot h_2(r) < 1 \text{ as } r \geq 0) \\
&= d(\boldsymbol{x}_2, \mathbb{D}) - d(\boldsymbol{x}_1, \mathbb{D}) \leq \|\boldsymbol{x}_1 - \boldsymbol{x}_2\|
\end{aligned}
$$

Finally, we consider the last case where $\boldsymbol{x}_1, \boldsymbol{x}_2 \in \overline{\mathbb{X}} \backslash \mathbb{D}$. Without loss of generality, we assume

$$
0 < d(\boldsymbol{x}_1, \mathbb{D}) \leq d(\boldsymbol{x}_2, \mathbb{D}).
$$

Then the definition of $h_1$ and $h_2$ implies that

$$
\begin{aligned}
&\|\mathcal{F}(\boldsymbol{x}_1) - \mathcal{F}(\boldsymbol{x}_2)\| \\
&\leq \max\Big( h_1(d(\boldsymbol{x}_1, \mathbb{D})), h_1(d(\boldsymbol{x}_2, \mathbb{D})) \Big) \cdot \|\boldsymbol{x}_1 - \boldsymbol{x}_2\| \\
&= h_1(d(\boldsymbol{x}_1, \mathbb{D})) \cdot \|\boldsymbol{x}_1 - \boldsymbol{x}_2\|,
\end{aligned}
$$

and

$$
\|\mathcal{F}(\boldsymbol{x}_2)\| \leq h_2(d(\boldsymbol{x}_2, \mathbb{D})).
$$

Consequently, applying (11) and the above inequalities, we have

$$
\begin{aligned}
&\|\varepsilon(\boldsymbol{x}_1)\mathcal{F}(\boldsymbol{x}_1) - \varepsilon(\boldsymbol{x}_2)\mathcal{F}(\boldsymbol{x}_2)\| \\
&\leq \varepsilon(\boldsymbol{x}_1)\big\|\mathcal{F}(\boldsymbol{x}_1) - \mathcal{F}(\boldsymbol{x}_2)\big\| + |\varepsilon(\boldsymbol{x}_1) - \varepsilon(\boldsymbol{x}_2)| \cdot \big\|\mathcal{F}(\boldsymbol{x}_2)\big\| \\
&\leq \varepsilon(\boldsymbol{x}_1) \cdot h_1(d(\boldsymbol{x}_1, \mathbb{D})) \cdot \|\boldsymbol{x}_1 - \boldsymbol{x}_2\| + |\varepsilon(\boldsymbol{x}_1) - \varepsilon(\boldsymbol{x}_2)| \cdot h_2(d(\boldsymbol{x}_2, \mathbb{D})) \\
&\leq \hat{\varepsilon}\Big(d(\boldsymbol{x}_1, \mathbb{D})\Big) \cdot h_1(d(\boldsymbol{x}_1, \mathbb{D})) \cdot \|\boldsymbol{x}_1 - \boldsymbol{x}_2\| + \left| \hat{\varepsilon}\Big(d(\boldsymbol{x}_1, \mathbb{D})\Big) - \hat{\varepsilon}\Big(d(\boldsymbol{x}_2, \mathbb{D})\Big) \right| \cdot h_2(d(\boldsymbol{x}_2, \mathbb{D})) \\
&= \left( \int_0^{d(\boldsymbol{x}_1, \mathbb{D})} \hat{h}(s)\mathrm{d}s \right) \cdot h_1(d(\boldsymbol{x}_1, \mathbb{D})) \cdot \|\boldsymbol{x}_1 - \boldsymbol{x}_2\| + \left( \int_{d(\boldsymbol{x}_1, \mathbb{D})}^{d(\boldsymbol{x}_2, \mathbb{D})} \hat{h}(s)\mathrm{d}s \right) \cdot h_2(d(\boldsymbol{x}_2, \mathbb{D})) \\
&\leq d(\boldsymbol{x}_1, \mathbb{D}) \cdot \hat{h}(d(\boldsymbol{x}_1, \mathbb{D})) \cdot h_1(d(\boldsymbol{x}_1, \mathbb{D})) \cdot \|\boldsymbol{x}_1 - \boldsymbol{x}_2\| \\
&\quad + \left| d(\boldsymbol{x}_1, \mathbb{D}) - d(\boldsymbol{x}_2, \mathbb{D}) \right| \cdot \hat{h}(d(\boldsymbol{x}_2, \mathbb{D})) \cdot h_2(d(\boldsymbol{x}_2, \mathbb{D})) \\
&< d(\boldsymbol{x}_1, \mathbb{D}) \cdot \|\boldsymbol{x}_1 - \boldsymbol{x}_2\| + \left| d(\boldsymbol{x}_1, \mathbb{D}) - d(\boldsymbol{x}_2, \mathbb{D}) \right| \\
&\leq d(\boldsymbol{x}_1, \mathbb{D}) \cdot \|\boldsymbol{x}_1 - \boldsymbol{x}_2\| + \|\boldsymbol{x}_1 - \boldsymbol{x}_2\|
\end{aligned}
$$

The last inequality results from $\hat{h}(r) \cdot (h_1(r) + h_2(r)) < 1$ for all $r > 0$. And the above inequalities imply

$$
\|\varepsilon(\boldsymbol{x}_1)\mathcal{F}(\boldsymbol{x}_1) - \varepsilon(\boldsymbol{x}_2)\mathcal{F}(\boldsymbol{x}_2)\| \leq (\mathrm{diam}(\overline{\mathbb{X}}) + 1) \cdot \|\boldsymbol{x}_1 - \boldsymbol{x}_2\|.
$$

Combining all the results together, we have $\varepsilon\mathcal{F}$ is locally $(\mathrm{diam}(\overline{\mathbb{X}}) + 1)$-Lipschitz at any $\boldsymbol{x} \in \overline{\mathbb{X}}$. Then the compactness of $\overline{\mathbb{X}}$ concludes the global Lipschitz continuous of $\varepsilon\mathcal{F}$, which finishes the whole proof. $\qquad\square$

Follows are some lemmas (as well as their proofs) that we used in the proof of Theorem A.1.

**Lemma A.2.** *Let $\mathbb{T} \subset \mathbb{R}^d$ be closed and let $\mathcal{F} : \mathbb{T} \to \mathbb{R}^n$. Denote by $\mathbb{D}(\mathcal{F}) \subset \mathbb{T}$ the set of points at which $\mathcal{F}$ is not locally Lipschitz. Then $\mathbb{D}(\mathcal{F})$ is closed (in $\mathbb{T}$, hence in $\mathbb{R}^d$).*

*Proof.* Recall that $\mathcal{F}$ is locally Lipschitz (relative to $\mathbb{T}$) at $\boldsymbol{x} \in \mathbb{T}$ if there exist $r > 0$ and $L > 0$ such that

$$
\|\mathcal{F}(\boldsymbol{u}) - \mathcal{F}(\boldsymbol{v})\| \leq L\|\boldsymbol{u} - \boldsymbol{v}\| \qquad \text{for all } \boldsymbol{u}, \boldsymbol{v} \in \mathbb{T} \cap \mathbb{B}(\boldsymbol{x}, r).
$$

Let $\mathbb{G} := \mathbb{T} \backslash \mathbb{D}(\mathcal{F})$ be the set of points where $\mathcal{F}$ *is* locally Lipschitz. We first show that $\mathbb{G}$ is relatively open in $\mathbb{T}$. Fix $\boldsymbol{x} \in \mathbb{G}$ and choose $r, L$ as above. If $\boldsymbol{x}' \in \mathbb{T} \cap \mathbb{B}(\boldsymbol{x}, r/2)$, then $\mathbb{B}(\boldsymbol{x}', r/2) \subset$

$\mathbb{B}(\boldsymbol{x}, r)$; hence the same $L$ works on $\mathbb{T} \cap \mathbb{B}(\boldsymbol{x}', r/2)$, so $\mathcal{F}$ is locally Lipschitz at $\boldsymbol{x}'$. Therefore $\mathbb{T} \cap \mathbb{B}(\boldsymbol{x}, r/2) \subset \mathbb{G}$, proving that $\mathbb{G}$ is open in $\mathbb{T}$. Consequently, $\mathbb{D}(\mathcal{F}) = \mathbb{T} \backslash \mathbb{G}$ is closed in $\mathbb{T}$. Since $\mathbb{T}$ is closed in $\mathbb{R}^d$, every set closed in $\mathbb{T}$ is also closed in $\mathbb{R}^d$. Hence $\mathbb{D}(\mathcal{F})$ is closed in $\mathbb{R}^d$ as well. $\qquad\square$

**Lemma A.3.** *Let $\mathbb{T}$ be a compact set. If $\mathcal{F}$ is locally Lipschitz everywhere on $\mathbb{T}$, then it must be globally Lipschitz on $\mathbb{T}$.*

*Proof.* Assume, to the contrary, that $\mathcal{F}$ is not globally Lipschitz on $\mathbb{T}$. Then we can choose sequences $\{\boldsymbol{x}_k\}_{k \geq 1}, \{\boldsymbol{y}_k\}_{k \geq 1} \subset \mathbb{T}$ such that

$$\frac{\|\mathcal{F}(\boldsymbol{x}_k) - \mathcal{F}(\boldsymbol{y}_k)\|}{\|\boldsymbol{x}_k - \boldsymbol{y}_k\|} \xrightarrow{k \to \infty} \infty. \tag{12}$$

Local Lipschitzness implies continuity of $\mathcal{F}$ on $\mathbb{T}$, so by compactness $\mathcal{F}$ is bounded: there exists $C < \infty$ with $\|\mathcal{F}(\boldsymbol{z})\| \leq C$ for all $\boldsymbol{z} \in \mathbb{T}$. Consequently,

$$\|\mathcal{F}(\boldsymbol{x}_k) - \mathcal{F}(\boldsymbol{y}_k)\| \leq 2C \qquad \text{for all } k,$$

and therefore (12) forces $\|\boldsymbol{x}_k - \boldsymbol{y}_k\| \to 0$.

By sequential compactness of $\mathbb{T}$, passing to a subsequence (not relabeled) we may assume $\boldsymbol{x}_k \to \boldsymbol{x} \in \mathbb{T}$; since $\|\boldsymbol{x}_k - \boldsymbol{y}_k\| \to 0$, we also have $\boldsymbol{y}_k \to \boldsymbol{x}$. Since $\mathcal{F}$ is locally Lipschitz at $\boldsymbol{x}$, for $k$ large enough we have

$$\frac{\|\mathcal{F}(\boldsymbol{x}_k) - \mathcal{F}(\boldsymbol{y}_k)\|}{\|\boldsymbol{x}_k - \boldsymbol{y}_k\|} \leq L,$$

for some $L > 0$, which contradicts (12). Therefore $\mathcal{F}$ must be globally Lipschitz on $\mathbb{T}$. $\qquad\square$

Now we relax the condition in Theorem A.1 and extend it to unbounded domains.

**Theorem A.4.** *For any $\mathbb{X} \subset \mathbb{R}^d$ (not necessarily bounded) and any locally Lipschitz function $\mathcal{F} : \mathbb{X} \to \mathbb{R}$, there exists a function $\varepsilon : \mathbb{X} \to \mathbb{R}$ such that $0 < \varepsilon(\boldsymbol{x}) < 1$ for $\boldsymbol{x} \in \mathbb{X}$, and $\varepsilon(\boldsymbol{x})$ and $\varepsilon(\boldsymbol{x})\mathcal{F}(\boldsymbol{x})$ are both globally Lipschitz continuous on $\mathbb{X}$.*

*Proof.* We consider the following grids in $\mathbb{R}^d$:

$$\boldsymbol{k} = (2k_1, 2k_2, \ldots, 2k_d), \quad k_i \in \mathbb{Z}, \, i \in [d].$$

We define the closed d-dimensional cubic centered at $\boldsymbol{k}$ by

$$\mathcal{C}_{\boldsymbol{k}} = \left[2k_1 - \frac{3}{2}, 2k_1 + \frac{3}{2}\right] \times \left[2k_2 - \frac{3}{2}, 2k_2 + \frac{3}{2}\right] \times \cdots \times \left[2k_d - \frac{3}{2}, 2k_d + \frac{3}{2}\right].$$

According to Theorem A.1, there exists a function $\varepsilon_{\boldsymbol{k}}$ defined on $\mathcal{C}_{\boldsymbol{k}} \cap \mathbb{X}$, so that

- $0 < \varepsilon_{\boldsymbol{k}}(\boldsymbol{x}) < 1$ on $\mathcal{C}_{\boldsymbol{k}} \cap \mathbb{X}$.

- $\varepsilon_{\boldsymbol{k}}(\boldsymbol{x})$ is 1-Lipschitz on $\mathcal{C}_{\boldsymbol{k}} \cap \mathbb{X}$.

- $\varepsilon_{\boldsymbol{k}}(\boldsymbol{x})\mathcal{F}(\boldsymbol{x})$ is $(\operatorname{diam}(\mathcal{C}_{\boldsymbol{k}}) + 1)$-Lipschitz on $\mathcal{C}_{\boldsymbol{k}} \cap \mathbb{X}$.

Next, we concatenate all these $\varepsilon_{\boldsymbol{k}}$ functions and get a global $\varepsilon : \mathbb{X} \to \mathbb{R}$. We define the following concatenation function in 1-dimensional space:

$$\rho(x) = \begin{cases} x + \frac{3}{2}, & x \in \left[-\frac{3}{2}, -\frac{1}{2}\right], \\ 1, & x \in \left[-\frac{1}{2}, \frac{1}{2}\right], \\ -x + \frac{3}{2}, & x \in \left[\frac{1}{2}, \frac{3}{2}\right], \\ 0, & \text{otherwise.} \end{cases}$$

Then we define the d-dimensional concatenation function. Let $\boldsymbol{x} = (x_1, x_2, \ldots, x_d)$.

$$\rho^{(d)}(\boldsymbol{x}) = \prod_{i=1}^{d} \rho(x_i).$$

We define the shifting function $\rho_{\boldsymbol{k}}^{(d)}(\boldsymbol{x}) = \rho^{(d)}(\boldsymbol{x} - \boldsymbol{k})$. On $\mathbb{X}$, we construct

$$\varepsilon(\boldsymbol{x}) = \sum_{\boldsymbol{k} \in \{\boldsymbol{k}: \boldsymbol{x} \in \mathcal{C}_{\boldsymbol{k}}\}} \rho_{\boldsymbol{k}}^{(d)}(\boldsymbol{x}) \varepsilon_{\boldsymbol{k}}(\boldsymbol{x}).$$

Given the constructed $\varepsilon(\boldsymbol{x})$, it's enough to prove that: $0 < \varepsilon(\boldsymbol{x}) < 1$, $\varepsilon(\boldsymbol{x})$ and $\varepsilon(\boldsymbol{x})\mathcal{F}(\boldsymbol{x})$ are globally Lipschitz over $\mathbb{X}$. We will show these claims one by one and finish the proof.

First, let's show $0 < \varepsilon(\boldsymbol{x}) < 1$. As $\rho^{(d)}$ is non-negative and $0 < \varepsilon_{\boldsymbol{k}}(\boldsymbol{x}) < 1$, we have

$$0 < \varepsilon(\boldsymbol{x}) < \sum_{\boldsymbol{k}} \rho_{\boldsymbol{k}}^{(d)}(\boldsymbol{x}) = 1$$

where $\sum_{\boldsymbol{k}} \rho_{\boldsymbol{k}}^{(d)}(\boldsymbol{x}) = 1$ comes from the fact that each term $\rho(x_i - 2k_i)$ depends only on its specific index $k_i$ and not on the others, and hence we can distribute the summation as $\sum_i \sum_j a_i b_j = (\sum_i a_i)(\sum_j b_j)$. That is,

$$\sum_{\mathbf{k}} \rho_{\mathbf{k}}^{(d)}(\boldsymbol{x}) = \left( \sum_{k_1 \in \mathbb{Z}} \rho(x_1 - 2k_1) \right) \times \left( \sum_{k_2 \in \mathbb{Z}} \rho(x_2 - 2k_2) \right) \times \cdots \times \left( \sum_{k_d \in \mathbb{Z}} \rho(x_d - 2k_d) \right)$$
$$= 1 \times 1 \times \cdots \times 1 = 1.$$

Second, we prove the Lipschitz continuity of $\varepsilon(\boldsymbol{x})$. In particular, it holds that

$$|\varepsilon(\boldsymbol{x}) - \varepsilon(\hat{\boldsymbol{x}})| = \left| \sum_{\mathbf{k}} \rho_{\mathbf{k}}^{(d)}(\boldsymbol{x})\varepsilon_{\mathbf{k}}(\boldsymbol{x}) - \sum_{\mathbf{k}} \rho_{\mathbf{k}}^{(d)}(\hat{\boldsymbol{x}})\varepsilon_{\mathbf{k}}(\hat{\boldsymbol{x}}) \right|$$
$$= \left| \sum_{\mathbf{k}} \rho_{\mathbf{k}}^{(d)}(\boldsymbol{x})(\varepsilon_{\mathbf{k}}(\boldsymbol{x}) - \varepsilon_{\mathbf{k}}(\hat{\boldsymbol{x}})) + \sum_{\mathbf{k}} (\rho_{\mathbf{k}}^{(d)}(\boldsymbol{x}) - \rho_{\mathbf{k}}^{(d)}(\hat{\boldsymbol{x}}))\varepsilon_{\mathbf{k}}(\hat{\boldsymbol{x}}) \right|$$
$$\leq \sum_{\mathbf{k}} \rho_{\mathbf{k}}^{(d)}(\boldsymbol{x}) \underbrace{|\varepsilon_{\mathbf{k}}(\boldsymbol{x}) - \varepsilon_{\mathbf{k}}(\hat{\boldsymbol{x}})|}_{\leq \|\boldsymbol{x}-\hat{\boldsymbol{x}}\|} + \sum_{\boldsymbol{k} \in \{\boldsymbol{k}: \boldsymbol{x} \in \mathcal{C}_{\boldsymbol{k}}\}} \underbrace{|\rho_{\mathbf{k}}^{(d)}(\boldsymbol{x}) - \rho_{\mathbf{k}}^{(d)}(\hat{\boldsymbol{x}})|}_{\leq \sqrt{d}\|\boldsymbol{x}-\hat{\boldsymbol{x}}\|} \underbrace{|\varepsilon_{\mathbf{k}}(\hat{\boldsymbol{x}})|}_{\leq 1}$$
$$\leq \|\boldsymbol{x} - \hat{\boldsymbol{x}}\| + 2^d \sqrt{d}\|\boldsymbol{x} - \hat{\boldsymbol{x}}\| = (1 + 2^d \sqrt{d})\|\boldsymbol{x} - \hat{\boldsymbol{x}}\|$$

where $2^d$ comes from the fact at most $2^d$ grid cubes overlap at each point $\mathbf{x}$.

Third, using the same argument, we can show that $\varepsilon(\boldsymbol{x})\mathcal{F}(\boldsymbol{x})$ is globally Lipschitz over $\mathbb{X}$, and the Lipschitz constant is bounded by $\left( 3\sqrt{d} + 1 + 2^d\sqrt{d} \right)$ (Recall that $\mathrm{diam}(\mathcal{C}_{\boldsymbol{k}}) = 3\sqrt{d}$), which finishes the proof. $\qquad\square$

## A.2 PROOF OF NECESSITY

For Theorem 2.5, we adopt a similar idea: first considering a bounded domain and then extending the results to unbounded domains.

**Theorem A.5.** *Let* $\mathbb{X} \subset \mathbb{R}^d$ *be a* bounded *domain and let* $\mathcal{G} : \mathbb{R}^n \times \mathbb{X} \to \mathbb{R}^n$ *be regular. Then, for every* $\boldsymbol{x} \in \mathbb{X}$*, the map* $\boldsymbol{y} \mapsto \mathcal{G}(\boldsymbol{y}, \boldsymbol{x})$ *has a unique fixed point* $\boldsymbol{y}_*(\boldsymbol{x})$*, and the resulting fixed-point map* $\boldsymbol{y}_*(\boldsymbol{x})$ *must be locally Lipschitz on* $\mathbb{X}$*.*

*Proof.* Let $\overline{\mathbb{X}}$ be the closure of $\mathbb{X}$. In this proof, we will first extend the operator $\mathcal{G}$ to $\mathbb{R}^n \times \overline{\mathbb{X}}$, and then analyze its properties on this closed domain.

**Step 1: Extension to $\overline{\mathbb{X}}$.** For any $\boldsymbol{y} \in \mathbb{R}^n$, $\mathcal{G}(\boldsymbol{y}, \boldsymbol{x})$ is globally Lipschitz continuous on $\mathbb{X}$, hence its extension is naturally define by

$$\mathcal{G}(\boldsymbol{y}, \bar{\boldsymbol{x}}) := \lim_{\mathbb{X} \ni \boldsymbol{x} \to \bar{\boldsymbol{x}}} \mathcal{G}(\boldsymbol{y}, \boldsymbol{x}), \quad \text{for all } \bar{\boldsymbol{x}} \in \overline{\mathbb{X}} \backslash \mathbb{X}.$$

Different from the proof of Theorem A.1 where $\mathcal{F}$ might be not locally Lipschitz at $\bar{\boldsymbol{x}}$ even if it is continuous at $\bar{\boldsymbol{x}}$, here the extended $\mathcal{G}$ must be Lipschitz at $\bar{\boldsymbol{x}}$ and hence Lipschitz on the overall set $\overline{\mathbb{X}}$. This can be verified by examining the difference quotient for $\boldsymbol{x}_1 \neq \boldsymbol{x}_2$ and $\boldsymbol{y} \in \mathbb{R}^n$:

$$\Delta \mathcal{G}[\boldsymbol{y}; \boldsymbol{x}_1, \boldsymbol{x}_2] := \frac{\|\mathcal{G}(\boldsymbol{y}, \boldsymbol{x}_1) - \mathcal{G}(\boldsymbol{y}, \boldsymbol{x}_2)\|}{\|\boldsymbol{x}_1 - \boldsymbol{x}_2\|}$$

Let $\mathcal{G}(\boldsymbol{y}, \cdot)$'s Lipschitz constant on $\mathbb{X}$ be $L(\boldsymbol{y}) := \sup_{\boldsymbol{x}_1 \neq \boldsymbol{x}_2 \in \mathbb{X}} \Delta \mathcal{G}[\boldsymbol{y}; \boldsymbol{x}_1, \boldsymbol{x}_2]$. For any $\boldsymbol{x}_1 \in \mathbb{X}$ and $\bar{\boldsymbol{x}}_2 \in \overline{\mathbb{X}} \backslash \mathbb{X}$, it holds that

$$\Delta \mathcal{G}[\boldsymbol{y}; \boldsymbol{x}_1, \bar{\boldsymbol{x}}_2] = \lim_{\mathbb{X} \ni \boldsymbol{x}_2 \to \bar{\boldsymbol{x}}_2} \Delta \mathcal{G}[\boldsymbol{y}; \boldsymbol{x}_1, \boldsymbol{x}_2] \leq \sup_{\boldsymbol{x}_2 \in \mathbb{X} : \boldsymbol{x}_2 \neq \boldsymbol{x}_1} \Delta \mathcal{G}[\boldsymbol{y}; \boldsymbol{x}_1, \boldsymbol{x}_2] \leq L(\boldsymbol{y})$$

For any $\bar{\boldsymbol{x}}_1 \neq \bar{\boldsymbol{x}}_2 \in \overline{\mathbb{X}} \backslash \mathbb{X}$, we have

$$\Delta \mathcal{G}[\boldsymbol{y}; \bar{\boldsymbol{x}}_1, \bar{\boldsymbol{x}}_2] = \lim_{\mathbb{X} \ni \boldsymbol{x}_1 \to \bar{\boldsymbol{x}}_1} \lim_{\mathbb{X} \ni \boldsymbol{x}_2 \to \bar{\boldsymbol{x}}_2} \Delta \mathcal{G}[\boldsymbol{y}; \boldsymbol{x}_1, \boldsymbol{x}_2] \leq \sup_{\boldsymbol{x}_1, \boldsymbol{x}_2 \in \mathbb{X} : \boldsymbol{x}_2 \neq \boldsymbol{x}_1} \Delta \mathcal{G}[\boldsymbol{y}; \boldsymbol{x}_1, \boldsymbol{x}_2] = L(\boldsymbol{y})$$

Therefore, we obtain an upper bound for $\mathcal{G}(\boldsymbol{y}, \cdot)$'s Lipschitz constant on $\overline{\mathbb{X}}$:

$$\sup_{\boldsymbol{x}_1 \neq \boldsymbol{x}_2 \in \overline{\mathbb{X}}} \Delta \mathcal{G}[\boldsymbol{y}; \boldsymbol{x}_1, \boldsymbol{x}_2]$$
$$= \max \left( \sup_{\boldsymbol{x}_1 \neq \boldsymbol{x}_2 \in \mathbb{X}} \Delta \mathcal{G}[\boldsymbol{y}; \bar{\boldsymbol{x}}_1, \bar{\boldsymbol{x}}_2], \sup_{\boldsymbol{x}_1 \in \mathbb{X}, \boldsymbol{x}_2 \in \overline{\mathbb{X}} \backslash \mathbb{X}} \Delta \mathcal{G}[\boldsymbol{y}; \boldsymbol{x}_1, \bar{\boldsymbol{x}}_2], \sup_{\boldsymbol{x}_1 \neq \boldsymbol{x}_2 \in \overline{\mathbb{X}} \backslash \mathbb{X}} \Delta \mathcal{G}[\boldsymbol{y}; \bar{\boldsymbol{x}}_1, \bar{\boldsymbol{x}}_2] \right)$$
$$\leq \max \left( L(\boldsymbol{y}), L(\boldsymbol{y}), L(\boldsymbol{y}) \right) = L(\boldsymbol{y})$$

That is, for any $\boldsymbol{y} \in \mathbb{R}^n$, $\mathcal{G}(\boldsymbol{y}, \cdot)$ is globally Lipschitz on $\overline{\mathbb{X}}$, and the Lipschitz constant is the same with that of $\mathbb{X}$.

In the other hand, let's consider the Lipschitz constant (contraction constant) w.r.t. $\boldsymbol{y}$ when fixing $\bar{\boldsymbol{x}} \in \overline{\mathbb{X}} \backslash \mathbb{X}$:

$$\mu(\bar{\boldsymbol{x}}) = \lim_{\mathbb{X} \ni \boldsymbol{x} \to \bar{\boldsymbol{x}}} \mu(\boldsymbol{x})$$

Since $0 < \mu(\boldsymbol{x}) < 1$ for $\boldsymbol{x} \in \mathbb{X}$, by taking limit, we have $0 \leq \mu(\bar{\boldsymbol{x}}) \leq 1$. For those $\bar{\boldsymbol{x}}$ with $\mu(\bar{\boldsymbol{x}}) < 1$, the operator $\mathcal{G}(\cdot, \bar{\boldsymbol{x}})$ is still contractive. But if $\mu(\bar{\boldsymbol{x}}) = 1$, the operator $\mathcal{G}(\cdot, \bar{\boldsymbol{x}})$ is not contractive.

**Step 2: Defining $\mathbb{D}$ and $\mathbb{D}_r$.** We collect all points $\boldsymbol{x} \in \overline{\mathbb{X}}$ where the operator $\mathcal{G}(\cdot, \boldsymbol{x})$ is not contractive:

$$\mathbb{D} := \left\{ \boldsymbol{x} \in \overline{\mathbb{X}} : \mu(\boldsymbol{x}) = 1 \right\}$$

and define a "safe" set that is sufficiently far from $\mathbb{D}$:

$$\mathbb{D}_r := \left\{ \boldsymbol{x} \in \overline{\mathbb{X}} : d(\boldsymbol{x}, \mathbb{D}) \geq r \right\}.$$

Note that $\overline{\mathbb{X}} \backslash \mathbb{D} = \bigcup_{r > 0} \mathbb{D}_r$ and $\mathbb{X} \subset \overline{\mathbb{X}} \backslash \mathbb{D}$. We obtain

$$\mathbb{X} \subset \bigcup_{r > 0} \mathbb{D}_r.$$

For any $\mathbb{D}_r$ with $r > 0$, we can obtain a uniform contraction of the operator $\mathcal{G}(\cdot, \boldsymbol{x})$: There is a constant $\mu_r \in (0, 1)$ such that

$$\|\mathcal{G}(\boldsymbol{y}_1, \boldsymbol{x}) - \mathcal{G}(\boldsymbol{y}_2, \boldsymbol{x})\| \leq \mu_r \|\boldsymbol{y}_1 - \boldsymbol{y}_2\| \tag{13}$$

for all $\boldsymbol{y}_1, \boldsymbol{y}_2 \in \mathbb{R}^n$ and $\boldsymbol{x} \in \mathbb{D}_r$, which follows immediately from the continuity of $\mu(\boldsymbol{x})$ and the compactness of $\mathbb{D}_r$. By the Banach fixed-point theorem, the operator $\mathcal{G}(\cdot, \boldsymbol{x})$ must have a unique fixed point $\boldsymbol{y}_*$ for each $\boldsymbol{x} \in \mathbb{D}_r$.

To complete the proof of Theorem 2.5, thanks to the fact that $\mathbb{X} \subset \bigcup_{r > 0} \mathbb{D}_r$, it's enough to show that: For any $\mathbb{D}_r$ with $r > 0$, there is a constant $C_r$ such that

$$\|\boldsymbol{y}_*(\boldsymbol{x}_1) - \boldsymbol{y}_*(\boldsymbol{x}_2)\| \leq C_r \|\boldsymbol{x}_1 - \boldsymbol{x}_2\| \tag{14}$$

holds for all $\boldsymbol{x}_1, \boldsymbol{x}_2 \in \mathbb{D}_r$. In the following steps, we will show (14).

**Step 3: A controllable sequence.** Fix $\boldsymbol{x} \in \mathbb{D}_r$. By defining a sequence $\{\boldsymbol{y}_k(\boldsymbol{x})\}_{k \geq 0} \subset \mathbb{R}^n$:

$$\boldsymbol{y}_{k+1}(\boldsymbol{x}) = \mathcal{G}(\boldsymbol{y}_k(\boldsymbol{x}), \boldsymbol{x}), \quad \boldsymbol{y}_0 \text{ is constant for all } \boldsymbol{x},$$

we are able to estimate the upper bound of $\|\boldsymbol{y}_*(\boldsymbol{x})\|$. In particular, we decompose $\boldsymbol{y}_0 - \boldsymbol{y}_*$ by a series:

$$\boldsymbol{y}_0 - \boldsymbol{y}_* = \lim_{k \to \infty} (\boldsymbol{y}_0 - \boldsymbol{y}_k) = \sum_{k=0}^{\infty} (\boldsymbol{y}_k - \boldsymbol{y}_{k+1})$$

Thanks to (13), we have

$$\|\boldsymbol{y}_k(\boldsymbol{x}) - \boldsymbol{y}_{k+1}(\boldsymbol{x})\| \leq \mu_r \|\boldsymbol{y}_{k-1}(\boldsymbol{x}) - \boldsymbol{y}_k(\boldsymbol{x})\| \cdots \leq \mu_r^k \|\boldsymbol{y}_0 - \boldsymbol{y}_1(\boldsymbol{x})\| = \mu_r^k \|\boldsymbol{y}_0 - \mathcal{G}(\boldsymbol{y}_0, \boldsymbol{x})\|$$

for all $\boldsymbol{x} \in \mathbb{D}_r$. Therefore, it holds that

$$\begin{aligned}
\|\boldsymbol{y}_0 - \boldsymbol{y}_*(\boldsymbol{x})\| &\leq \sum_{k=0}^{\infty} \|\boldsymbol{y}_k(\boldsymbol{x}) - \boldsymbol{y}_{k+1}(\boldsymbol{x})\| \\
&\leq \left( \sum_{k=0}^{\infty} \mu_r^k \right) \|\boldsymbol{y}_0 - \mathcal{G}(\boldsymbol{y}_0, \boldsymbol{x})\| = \frac{1}{1 - \mu_r} \|\boldsymbol{y}_0 - \mathcal{G}(\boldsymbol{y}_0, \boldsymbol{x})\|
\end{aligned}$$

Now we can conclude the boundedness of $\|\boldsymbol{y}_*(\boldsymbol{x})\|$ for $\boldsymbol{x} \in \mathbb{D}_r$ by the compactness of $\mathbb{D}_r$:

$$\|\boldsymbol{y}_*(\boldsymbol{x})\| \leq \underbrace{\|\boldsymbol{y}_0\| + \frac{1}{1 - \mu_r} \sup_{\boldsymbol{x} \in \mathbb{D}_r} \|\boldsymbol{y}_0 - \mathcal{G}(\boldsymbol{y}_0, \boldsymbol{x})\|}_{\text{defined as } M_r \geq 0.}$$

With the same argument, we have $\|\boldsymbol{y}_k(\boldsymbol{x})\| \leq M_r$ for all $k \geq 0$ and $\boldsymbol{x} \in \mathbb{D}_r$. It implies that

$$L(\boldsymbol{y}_k(\boldsymbol{x})) \leq L_1 + L_2 M_r$$

for some $L_1, L_2 > 0$ as $L(\boldsymbol{y})$ grows linearly w.r.t. $\|\boldsymbol{y}\|$. Consequently, we can estimate an upper bound for the Lipschitz constant of $\boldsymbol{y}_k(\boldsymbol{x})$. In particular, for $\boldsymbol{x}_1, \boldsymbol{x}_2 \in \mathbb{D}_r$, it holds that

$$\begin{aligned}
&\|\boldsymbol{y}_{k+1}(\boldsymbol{x}_1) - \boldsymbol{y}_{k+1}(\boldsymbol{x}_2)\| \\
=& \|\mathcal{G}(\boldsymbol{y}_k(\boldsymbol{x}_1), \boldsymbol{x}_1) - \mathcal{G}(\boldsymbol{y}_k(\boldsymbol{x}_2), \boldsymbol{x}_2)\| \\
=& \|\mathcal{G}(\boldsymbol{y}_k(\boldsymbol{x}_1), \boldsymbol{x}_1) - \mathcal{G}(\boldsymbol{y}_k(\boldsymbol{x}_2), \boldsymbol{x}_1) + \mathcal{G}(\boldsymbol{y}_k(\boldsymbol{x}_2), \boldsymbol{x}_1) - \mathcal{G}(\boldsymbol{y}_k(\boldsymbol{x}_2), \boldsymbol{x}_2)\| \\
\leq& \|\mathcal{G}(\boldsymbol{y}_k(\boldsymbol{x}_1), \boldsymbol{x}_1) - \mathcal{G}(\boldsymbol{y}_k(\boldsymbol{x}_2), \boldsymbol{x}_1)\| + \|\mathcal{G}(\boldsymbol{y}_k(\boldsymbol{x}_2), \boldsymbol{x}_1) - \mathcal{G}(\boldsymbol{y}_k(\boldsymbol{x}_2), \boldsymbol{x}_2)\| \\
\leq& \mu_r \|\boldsymbol{y}_k(\boldsymbol{x}_1) - \boldsymbol{y}_k(\boldsymbol{x}_2)\| + (L_1 + L_2 M_r)\|\boldsymbol{x}_1 - \boldsymbol{x}_2\|
\end{aligned}$$

For simplicity, let $L_r := L_1 + L_2 M_r$, $a_k := \|\boldsymbol{y}_k(\boldsymbol{x}_1) - \boldsymbol{y}_k(\boldsymbol{x}_2)\|$, and $h := \|\boldsymbol{x}_1 - \boldsymbol{x}_2\|$. By recursively applying $a_{k+1} \leq \mu_r a_k + Lh$ and $a_0 = 0$, we have

$$\|\boldsymbol{y}_k(\boldsymbol{x}_1) - \boldsymbol{y}_k(\boldsymbol{x}_2)\| = a_k \leq (\mu_r)^k a_0 + (\mu_r^{k-1} + \cdots + \mu_r + 1) L_r h \leq \frac{1}{1 - \mu_r} L_r h = \frac{L_r}{1 - \mu_r} \|\boldsymbol{x}_1 - \boldsymbol{x}_2\|.$$

**Step 4: Final proof.** As $\mathcal{G}(\cdot, \boldsymbol{x})$ is a contractor w.r.t. $\boldsymbol{y}$ for any $\boldsymbol{x} \in \mathbb{D}_r$, it holds that $\boldsymbol{y}_k(\boldsymbol{x}) \to \boldsymbol{y}_*(\boldsymbol{x})$ for any $\boldsymbol{x} \in \mathbb{D}_r$. (Here, as for the "convergence," we mean the pointwise convergence, which is enough here. We don't need stronger conditions like the uniform convergence.) For the above $\boldsymbol{x}_1, \boldsymbol{x}_2$, there is a $K$ such that

$$\|\boldsymbol{y}_k(\boldsymbol{x}_1) - \boldsymbol{y}_*(\boldsymbol{x}_1)\| \leq \frac{L_r}{1 - \mu_r} \|\boldsymbol{x}_1 - \boldsymbol{x}_2\|, \quad \|\boldsymbol{y}_k(\boldsymbol{x}_2) - \boldsymbol{y}_*(\boldsymbol{x}_2)\| \leq \frac{L_r}{1 - \mu_r} \|\boldsymbol{x}_1 - \boldsymbol{x}_2\|$$

for $k \geq K$. Combining the above results, we obtain

$$\begin{aligned}
\|\boldsymbol{y}_*(\boldsymbol{x}_1) - \boldsymbol{y}_*(\boldsymbol{x}_2)\| &\leq \|\boldsymbol{y}_*(\boldsymbol{x}_1) - \boldsymbol{y}_k(\boldsymbol{x}_1)\| + \|\boldsymbol{y}_k(\boldsymbol{x}_1) - \boldsymbol{y}_k(\boldsymbol{x}_2)\| + \|\boldsymbol{y}_k(\boldsymbol{x}_2) - \boldsymbol{y}_*(\boldsymbol{x}_2)\| \\
&\leq \frac{3L_r}{1 - \mu_r} \|\boldsymbol{x}_1 - \boldsymbol{x}_2\|
\end{aligned}$$

By letting $C_r = 3L_r/(1 - \mu_r)$, we get (14), which completes the proof. $\qquad \square$

**Remark A.6.** Our result relaxes two uniformity requirements in (Dontchev & Rockafellar, 2009, Thm. 1A.4): (i) the contraction modulus $\mu(\boldsymbol{x})$ is allowed to vary with $\boldsymbol{x}$ (it only needs to be continuous in $\boldsymbol{x}$), rather than being a single global constant; and (ii) for each $\boldsymbol{y}$, the mapping $\boldsymbol{x} \mapsto \mathcal{G}(\boldsymbol{y}, \boldsymbol{x})$ is Lipschitz on $\mathbb{X}$ with a constant that may grow linearly in $\|\boldsymbol{y}\|$, instead of being uniformly bounded in $\boldsymbol{y}$. Because these bounds are not uniform, we conclude only local (as opposed to global) Lipschitz continuity of the fixed-point map $\boldsymbol{x} \mapsto \boldsymbol{y}_*(\boldsymbol{x})$ on $\mathbb{X}$.

Now we relax the condition in Theorem A.5 to unbounded domains and prove Theorem 2.5 based on Theorem A.5.

*Proof of Theorem 2.5.* We cover the domain $\mathbb{R}^d$ using the grid $\boldsymbol{k} = (2k_1, \ldots, 2k_d)$ for $k_i \in \mathbb{Z}$, defining closed cubic regions $\mathcal{C}_{\boldsymbol{k}}$ of side length 3 centered at each $\boldsymbol{k}$:

$$\mathcal{C}_{\boldsymbol{k}} = \left[2k_1 - \frac{3}{2}, 2k_1 + \frac{3}{2}\right] \times \left[2k_2 - \frac{3}{2}, 2k_2 + \frac{3}{2}\right] \times \cdots \times \left[2k_d - \frac{3}{2}, 2k_d + \frac{3}{2}\right].$$

By applying Theorem A.5 to the bounded set $\mathcal{C}_{\boldsymbol{k}} \cap \mathbb{X}$, we guarantee the existence of a unique fixed-point map $\boldsymbol{y}_{\boldsymbol{k},*} : \mathcal{C}_{\boldsymbol{k}} \cap \mathbb{X} \to \mathbb{R}^n$ which is locally Lipschitz continuous on its domain.

Consider any $\boldsymbol{x}$ in the intersection of two regions $\mathcal{C}_{\boldsymbol{k}} \cap \mathcal{C}_{\boldsymbol{k}'}$. Since $\mathcal{G}(\cdot, \boldsymbol{x})$ is a contraction, it admits a unique fixed point in $\mathbb{R}^n$. Therefore, the local solutions must coincide:

$$\boldsymbol{y}_{\boldsymbol{k},*}(\boldsymbol{x}) = \boldsymbol{y}_{\boldsymbol{k}',*}(\boldsymbol{x}).$$

This consistency allows us to define a global fixed-point map $\boldsymbol{y}_* : \mathbb{X} \to \mathbb{R}^n$ by setting $\boldsymbol{y}_*(\boldsymbol{x}) = \boldsymbol{y}_{\boldsymbol{k},*}(\boldsymbol{x})$ for any $\boldsymbol{k}$ such that $\boldsymbol{x} \in \mathcal{C}_{\boldsymbol{k}}$. Since $\boldsymbol{y}_*$ coincides with a locally Lipschitz function $\boldsymbol{y}_{\boldsymbol{k},*}$ on every compact neighborhood $\mathcal{C}_{\boldsymbol{k}}$, $\boldsymbol{y}_*$ is locally Lipschitz continuous on $\mathbb{X}$. $\quad\square$

## B  A VARIANT ARCHITECTURE

In practice, many works use a variant of the vanilla model $\boldsymbol{y}_* = \mathcal{G}(\boldsymbol{y}_*, \boldsymbol{x})$:

$$\boldsymbol{z}_* = \mathcal{G}(\boldsymbol{z}_*, \mathcal{Q}_1(\boldsymbol{x})), \quad \boldsymbol{y}_* = \mathcal{Q}_2(\boldsymbol{z}_*) \tag{15}$$

where $\mathcal{G}$ is the core implicit model, $\mathcal{Q}_1$ is a encoding network and $\mathcal{Q}_2$ is a decoding (readout).

At inference, one iterates $\boldsymbol{z}_t = \mathcal{G}(\boldsymbol{z}_{t-1}, \mathcal{Q}_1(\boldsymbol{x}))$ for $1 \leq t \leq T$ and finally $\boldsymbol{y}_T = \mathcal{Q}_2(\boldsymbol{z}_T)$. This often improves empirical performance but does not alter the expressivity in Theorems 2.4–2.5.

**Corollary B.1.** *Under Assumption 2.2, for any $\mathcal{F}$ there exists a regular implicit operator $\mathcal{G}$ and globally Lipschitz maps $\mathcal{Q}_1, \mathcal{Q}_2$ such that $\mathcal{Q}_2\left(\mathrm{Fix}\left(\mathcal{G}(\cdot, \mathcal{Q}_1(\boldsymbol{x}))\right)\right) = \mathcal{F}(\boldsymbol{x})$ for all $\boldsymbol{x} \in \mathbb{X}$. Conversely, for any regular implicit operator $\mathcal{G}$ any globally Lipschitz $\mathcal{Q}_1, \mathcal{Q}_2$, the fixed point $\boldsymbol{z}_*$ defined by (15) exists uniquely and the induced map $\boldsymbol{x} \mapsto \boldsymbol{y}_*$ must be locally Lipschitz on $\mathbb{X}$.*

*Proof.* The claim follows directly from Theorems 2.4–2.5.

*Sufficiency.* Given any locally Lipschitz target $\mathcal{F}$ on $\mathbb{X}$, Theorem 2.4 ensures the existence of a regular $\mathcal{G}$ whose fixed-point map equals $\mathcal{F}$. Taking $\mathcal{Q}_1, \mathcal{Q}_2$ as both identity maps recovers the sufficiency statement with globally Lipschitz $\mathcal{Q}_1, \mathcal{Q}_2$.

*Necessity.* Suppose $\mathcal{G}$ is regular and $\mathcal{Q}_1, \mathcal{Q}_2$ are globally Lipschitz. Then the composite update $\mathcal{G}(\boldsymbol{z}, \mathcal{Q}_1(\boldsymbol{x}))$ is still regular in $\boldsymbol{z}$ and $\boldsymbol{x}$. By Theorem 2.5, for every $\boldsymbol{x} \in \mathbb{X}$, there is a unique fixed point $\boldsymbol{z}_*(\boldsymbol{x})$ and the map $\boldsymbol{x} \mapsto \boldsymbol{z}_*(\boldsymbol{x})$ is locally Lipschitz on $\mathbb{X}$. Finally, applying the globally Lipschitz readout $\mathcal{Q}_2$ preserves local Lipschitz continuity, so $\boldsymbol{x} \mapsto \boldsymbol{y}_*$ is locally Lipschitz as claimed. The proof is finished. $\quad\square$

## C  PROOFS OF THEOREMS FOR INVERSE PROBLEMS

This section proves that the target solution mappings, $\mathcal{F}_{1a}$ and $\mathcal{F}_{1b}$, are single-valued and locally Lipschitz on their domain, as stated in Theorems 3.3 and 3.4. Before the proofs, we first provide some definitions that used in Assumption 3.1.

Given a close subset $\mathbb{M} \subset \mathbb{R}^n$, its *reach* $\tau$ is defined in Federer (1959):

$$\tau := \sup\{r > 0 : \forall \boldsymbol{y} \in \mathbb{R}^n \text{ with } \text{dist}(\boldsymbol{y}, \mathbb{M}) < r,$$
$$\text{there exists a unique } \boldsymbol{z} \in \mathbb{M} \text{ such that } \|\boldsymbol{y} - \boldsymbol{z}\| = \text{dist}(\boldsymbol{y}, \mathbb{M})\}.$$

A set with positive reach is also called a "prox-regular" set in the literature (Poliquin et al., 2000).

The Bi-Lipschitz condition refers to: for some $0 < \mu \leq L < +\infty$, it holds that

$$\mu\|\boldsymbol{y}_1 - \boldsymbol{y}_2\| \leq \|\boldsymbol{A}\boldsymbol{y}_1 - \boldsymbol{A}\boldsymbol{y}_2\| \leq L\|\boldsymbol{y}_1 - \boldsymbol{y}_2\| \quad \forall \boldsymbol{y}_1, \boldsymbol{y}_2 \in \mathbb{M}. \tag{16}$$

According to the definition, it holds that $0 < \mu \leq L \leq \sigma_{\max} < +\infty$. This condition ensures $\boldsymbol{A}$ can be viewed as an injective mapping when restricted to $\mathbb{M}$, which is important for the recovery guarantee.

*Remark for Assumption 3.1.* The assumption that data (particularly images) lies on a smooth manifold has a long and influential history (Roweis & Saul, 2000; Donoho & Grimes, 2005), and it is still widely used in recent literature. The compactness of the data manifold can be achieved by standard techniques like normalization. In addition, reach is an important concept for manifold to ensure the uniqueness of its projection (Federer, 1959; Aamari et al., 2019). The overall assumptions on manifolds, smoothness, compactness and positive reach, is typically used in recent literature regarding image and signal processing (Tang & Yang, 2024; Potaptchik et al., 2024; Azangulov et al., 2024). The on-manifold bi-Lipschitz condition does *not* require $\boldsymbol{A}$ to be globally invertible; it merely rules out ill-posedness *restricted to* $\mathbb{M}$. This is closely related to Johnson–Lindenstrauss (JL)–type embeddings in compressive sensing: e.g., Baraniuk & Wakin (2009) shows that random matrices are bi-Lipschitz on low-dimensional manifolds with high probability, and JL-style conditions are widely analyzed and used (Candes & Tao, 2006; Clarkson, 2008; Wakin, 2010; Iwen & Maggioni, 2013; Hegde & Baraniuk, 2012).

*Proof of Theorem 3.3.* For simplicity, we first denote the objective functions in (3) as $F_{1a}(\boldsymbol{y})$:

$$F_{1a}(\boldsymbol{y}) := \frac{1}{2}\|\boldsymbol{x} - \boldsymbol{A}\boldsymbol{y}\|^2 + \frac{\alpha}{2}\text{dist}^2(\boldsymbol{y}, \mathbb{M})$$

Then we introduce some definitions that will be useful in our proof:

$$\mathbb{U}_r(\mathbb{M}) := \{\boldsymbol{y} \in \mathbb{R}^n : \text{dist}(\boldsymbol{y}, \mathbb{M}) < r\}, \quad \overline{\mathbb{U}}_r(\mathbb{M}) := \{\boldsymbol{y} \in \mathbb{R}^n : \text{dist}(\boldsymbol{y}, \mathbb{M}) \leq r\}$$

Here, $\mathbb{U}_r(\mathbb{M})$ is an open tubular neighborhood of the manifold $\mathbb{M}$ and $\overline{\mathbb{U}}_r(\mathbb{M})$ is its closure. As $r = \tau$, the open set $\mathbb{U}_r(\mathbb{M})$ is named as the *reach tube* of $\mathbb{M}$, denoted as $\mathbb{U}_\tau(\mathbb{M})$. As introduced in Federer (1959), within the reach tube, some nice properties of the distance function and projection mapping can be utilized. For any $\boldsymbol{y} \in \mathbb{U}_\tau(\mathbb{M})$ or any $\boldsymbol{y} \in \overline{\mathbb{U}}_r(\mathbb{M})$ with $r < \tau$, the projection mapping

$$\boldsymbol{p}(\boldsymbol{y}) := \arg\min_{\boldsymbol{z} \in \mathbb{M}} \|\boldsymbol{z} - \boldsymbol{y}\|$$

is single valued and well defined, and $\text{dist}(\boldsymbol{y}, \mathbb{M}) = \|\boldsymbol{y} - \boldsymbol{p}(\boldsymbol{y})\|$.

**Step 1: Existence of minimizers of $F_{1a}$.** As $\boldsymbol{x} \in \mathbb{X}$, there must be an underlying $\boldsymbol{y}_* \in \mathbb{M}$ (hence $\boldsymbol{y}_* \in \overline{\mathbb{U}}_r(\mathbb{M})$) and $\boldsymbol{n}$ such that $\|\boldsymbol{x} - \boldsymbol{A}\boldsymbol{y}_*\| = \|\boldsymbol{n}\|$. Therefore, it holds that

$$F_{1a}(\boldsymbol{y}_*) = \frac{1}{2}\|\boldsymbol{x} - \boldsymbol{A}\boldsymbol{y}_*\|^2 + \frac{\alpha}{2}\text{dist}^2(\boldsymbol{y}_*, \mathbb{M}) = \frac{1}{2}\|\boldsymbol{n}\|^2 + 0 = \frac{1}{2}\|\boldsymbol{n}\|^2$$

In the other hand, for any point outside the tube: $\boldsymbol{y} \notin \overline{\mathbb{U}}_r(\mathbb{M})$, the objective value is lower bounded by:

$$F_{1a}(\boldsymbol{y}) \geq 0 + \frac{\alpha}{2}\text{dist}^2(\boldsymbol{y}, \mathbb{M}) > \frac{\alpha}{2}r^2$$

As long as we have large enough $\alpha$:

$$\alpha \geq \frac{\|\boldsymbol{n}\|^2}{r^2}, \tag{17}$$

we can ensure $F_{1a}(\boldsymbol{y}) > F_{1a}(\boldsymbol{y}_*)$ for all $\boldsymbol{y} \notin \overline{\mathbb{U}}_r(\mathbb{M})$, which implies $\inf_{\boldsymbol{y} \in \mathbb{R}^n} F_{1a}(\boldsymbol{y}) = \inf_{\boldsymbol{y} \in \overline{\mathbb{U}}_r(\mathbb{M})} F_{1a}(\boldsymbol{y})$. As $\mathbb{M}$ is compact, $\overline{\mathbb{U}}_r(\mathbb{M})$ must be compact as well. Consequently, the infimum of $F$ is attainable, which concludes the existence of the minimizer of $F_{1a}$, denoted by $\hat{\boldsymbol{y}}$, and

$\hat{\boldsymbol{y}} \in \overline{\mathbb{U}}_r(\mathbb{M})$. Finally, we have the conclusion: It holds for all $r > 0$ that, condition (17) ensures the existence of $\hat{\boldsymbol{y}}$ and $\hat{\boldsymbol{y}} \in \overline{\mathbb{U}}_r(\mathbb{M})$.

**Step 2: Bound of minimizers of $F_{1a}$.** For any $\boldsymbol{y} \in \mathbb{U}_\tau(\mathbb{M})$, the projection $\boldsymbol{p}(\boldsymbol{y})$ is uniquely defined, hence we have

$$\left\| \boldsymbol{A}\boldsymbol{y} - \boldsymbol{x} \right\| = \left\| \boldsymbol{A}\boldsymbol{y} - \boldsymbol{A}\boldsymbol{y}_* - \boldsymbol{n} \right\| = \left\| \boldsymbol{A}\boldsymbol{y} - \boldsymbol{A}\boldsymbol{p}(\boldsymbol{y}) + \boldsymbol{A}\boldsymbol{p}(\boldsymbol{y}) - \boldsymbol{A}\boldsymbol{y}_* - \boldsymbol{n} \right\|$$

$$\geq \left\| \boldsymbol{A}\boldsymbol{p}(\boldsymbol{y}) - \boldsymbol{A}\boldsymbol{y}_* \right\| - \left\| \boldsymbol{A}\boldsymbol{y} - \boldsymbol{A}\boldsymbol{p}(\boldsymbol{y}) \right\| - \|\boldsymbol{n}\|$$

$$\geq \mu \|\boldsymbol{p}(\boldsymbol{y}) - \boldsymbol{y}_*\| - \sigma_{\max} \|\boldsymbol{y} - \boldsymbol{p}(\boldsymbol{y})\| - \|\boldsymbol{n}\|$$

According to the conclusion in Step 1, as long as

$$\alpha \geq \frac{\|\boldsymbol{n}\|^2}{r^2} > \frac{\|\boldsymbol{n}\|^2}{\tau^2}, \tag{18}$$

it holds that the minimizer $\hat{\boldsymbol{y}}$ exists and $\hat{\boldsymbol{y}} \in \overline{\mathbb{U}}_r(\mathbb{M})$ for some $r < \tau$ and hence $\hat{\boldsymbol{y}} \in \mathbb{U}_\tau(\mathbb{M})$, which allows us to use the above inequalities at the beginning of Step 2. Now we aim to establish an upper bound for $\|\boldsymbol{p}(\hat{\boldsymbol{y}}) - \boldsymbol{y}_*\|$ by contradiction. Suppose

$$\mu \|\boldsymbol{p}(\hat{\boldsymbol{y}}) - \boldsymbol{y}_*\| > \sigma_{\max} \|\hat{\boldsymbol{y}} - \boldsymbol{p}(\hat{\boldsymbol{y}})\| + 2\|\boldsymbol{n}\|$$

we will obtain

$$\|\boldsymbol{A}\hat{\boldsymbol{y}} - \boldsymbol{x}\| \geq \mu \|\boldsymbol{p}(\hat{\boldsymbol{y}}) - \boldsymbol{y}_*\| - \sigma_{\max} \|\hat{\boldsymbol{y}} - \boldsymbol{p}(\hat{\boldsymbol{y}})\| - \|\boldsymbol{n}\| > \|\boldsymbol{n}\|,$$

which implies

$$F_{1a}(\hat{\boldsymbol{y}}) = \frac{1}{2}\|\boldsymbol{A}\hat{\boldsymbol{y}} - \boldsymbol{x}\|^2 + \frac{\alpha}{2}\text{dist}^2(\hat{\boldsymbol{y}}, \mathbb{M}) > \frac{1}{2}\|\boldsymbol{n}\|^2 + 0 = F_{1a}(\boldsymbol{y}_*).$$

This contradicts with the definition of $\hat{\boldsymbol{y}}$: the minimizer of function $F_{1a}$. Therefore, we obtain:

$$\mu \|\boldsymbol{p}(\hat{\boldsymbol{y}}) - \boldsymbol{y}_*\| \leq \sigma_{\max} \|\hat{\boldsymbol{y}} - \boldsymbol{p}(\hat{\boldsymbol{y}})\| + 2\|\boldsymbol{n}\| \leq \sigma_{\max} r + 2\|\boldsymbol{n}\|$$

which is equivalent to

$$\|\boldsymbol{p}(\hat{\boldsymbol{y}}) - \boldsymbol{y}_*\| \leq \frac{\sigma_{\max}}{\mu} r + \frac{2}{\mu}\|\boldsymbol{n}\|$$

and implies that

$$\|\hat{\boldsymbol{y}} - \boldsymbol{y}_*\| \leq \|\hat{\boldsymbol{y}} - \boldsymbol{p}(\hat{\boldsymbol{y}})\| + \|\boldsymbol{p}(\hat{\boldsymbol{y}}) - \boldsymbol{y}_*\| \leq \left(1 + \frac{\sigma_{\max}}{\mu}\right) r + \frac{2}{\mu}\|\boldsymbol{n}\| \tag{19}$$

holds for all $\hat{\boldsymbol{y}}$ that minimizes $F_{1a}(\boldsymbol{y})$.

**Step 3: Positive definiteness of the Hessian of $F_{1a}$.** To prove the uniqueness of the solution, we will establish the strict convexity of the objective function $F_{1a}(\boldsymbol{y})$ within a neighborhood around any point of $\mathbb{M}$. To achieve this, we establish the positive definiteness of the Hessian of $F_{1a}(\boldsymbol{y})$ in this step.

For any $\boldsymbol{y} \in \mathbb{U}_\tau(\mathbb{M})$, the projection mapping is single valued and the objective function can be written as

$$F_{1a}(\boldsymbol{y}) = \frac{1}{2}\underbrace{\|\boldsymbol{x} - \boldsymbol{A}\boldsymbol{y}\|^2}_{f(\boldsymbol{y})} + \frac{\alpha}{2}\underbrace{\|\boldsymbol{y} - \boldsymbol{p}(\boldsymbol{y})\|^2}_{g(\boldsymbol{y})}$$

The smoothness of $\mathbb{M}$ implies the smoothness of $g$ and of the projection mapping, and hence we can take first and second orders of derivatives on $g$ (Leobacher & Steinicke, 2021, Theorem 2). Thanks to (Federer, 1959, Theorem 4.8), the gradient and Hessian of $g$ are given by

$$\nabla g(\boldsymbol{y}) = 2\left(\boldsymbol{y} - \boldsymbol{p}(\boldsymbol{y})\right), \quad \nabla^2 g(\boldsymbol{y}) = 2\left(\boldsymbol{I} - D\boldsymbol{p}(\boldsymbol{y})\right),$$

where $D\boldsymbol{p}$ denotes the Jacobian of the projection mapping. The overall Hessian of $F_{1a}$ is provided by

$$\nabla^2 F_{1a}(\boldsymbol{y}) = \boldsymbol{A}^\top \boldsymbol{A} + \alpha\left(\boldsymbol{I} - D\boldsymbol{p}(\boldsymbol{y})\right). \tag{20}$$

To further present the properties of the above Hessian, we introduce a space decomposition according to $\boldsymbol{p}(\boldsymbol{y})$:

$$\mathbb{R}^n = \mathbb{T}_{\boldsymbol{p}(\boldsymbol{y})}(\mathbb{M}) \oplus \mathbb{N}_{\boldsymbol{p}(\boldsymbol{y})}(\mathbb{M})$$

where $\mathbb{T}_{\boldsymbol{p}(\boldsymbol{y})}(\mathbb{M})$ denotes the tangent space of $\mathbb{M}$ at the point $\boldsymbol{p}(\boldsymbol{y}) \in \mathbb{M}$, and $\mathbb{N}_{\boldsymbol{p}(\boldsymbol{y})}(\mathbb{M})$ represents the normal space. According to (Leobacher & Steinicke, 2021, Theorem C and Definition 7), the matrix $D\boldsymbol{p}(\boldsymbol{y})$ is actually restricted to the tangent space. In other words, for any decomposition $\boldsymbol{h}$ with $\boldsymbol{h} = \boldsymbol{h}_{\mathrm{T}} + \boldsymbol{h}_{\mathrm{N}}$ where $\boldsymbol{h}_{\mathrm{T}} \in \mathbb{T}_{\boldsymbol{p}(\boldsymbol{y})}(\mathbb{M})$ and $\boldsymbol{h}_{\mathrm{N}} \in \mathbb{N}_{\boldsymbol{p}(\boldsymbol{y})}(\mathbb{M})$, it holds that

$$D\boldsymbol{p}(\boldsymbol{y})\boldsymbol{h}_{\mathrm{N}} = \boldsymbol{0}, \quad D\boldsymbol{p}(\boldsymbol{y})\boldsymbol{h}_{\mathrm{T}} \in \mathbb{T}_{\boldsymbol{p}(\boldsymbol{y})}(\mathbb{M}). \tag{21}$$

In addition, function $g(\boldsymbol{y})$ is $(\frac{s}{\tau-s})$-weakly convex where $\tau$ is the reach of $\mathbb{M}$ and $s = \mathrm{dist}(\boldsymbol{y}, \mathbb{M})$ (Nacry & Thibault, 2022, Section 5), and hence the spectrum of $\nabla^2 g$ can be lower bounded by

$$\langle \boldsymbol{h}_{\mathrm{T}}, \nabla^2 g(\boldsymbol{y})\boldsymbol{h}_{\mathrm{T}} \rangle \geq -\frac{2s}{\tau-s}\|\boldsymbol{h}_{\mathrm{T}}\|^2, \tag{22}$$

Now, let's turn to the first term in the Hessian: $\boldsymbol{A}^\top \boldsymbol{A}$. It can be shown using the JL condition (16) that, the spectrum of $\boldsymbol{A}^\top \boldsymbol{A}$ restricted to the tangent space can also be lower bounded. In particular, we pick an arbitrary tangent vector $\boldsymbol{h}_{\mathrm{T}} \in \mathbb{T}_{\boldsymbol{p}(\boldsymbol{y})}(\mathbb{M})$. According to the definition of tangent space, there must be a curve $\gamma : (-\delta, \delta) \to \mathbb{M}$ with $\delta > 0$, $\gamma(0) = \boldsymbol{p}(\boldsymbol{y})$, and $\gamma'(0) = \boldsymbol{h}_{\mathrm{T}}$. For any $0 \leq t < \delta$, $\gamma(t) \in \mathbb{M}$. By applying condition (16) with the pair $(\gamma(t), \gamma(0))$ and divide by $t^2$, we have

$$\mu^2 \frac{\|\gamma(t) - \gamma(0)\|^2}{t^2} \leq \frac{\|\boldsymbol{A}\gamma(t) - \boldsymbol{A}\gamma(0)\|^2}{t^2} \leq L^2 \frac{\|\gamma(t) - \gamma(0)\|^2}{t^2}$$

By differentiability and the continuity of the operator $\boldsymbol{A}$, it holds that

$$\lim_{t \to 0} \frac{\gamma(t) - \gamma(0)}{t} = \boldsymbol{h}_{\mathrm{T}}, \quad \lim_{t \to 0} \frac{\boldsymbol{A}\gamma(t) - \boldsymbol{A}\gamma(0)}{t} = \boldsymbol{A}\boldsymbol{h}_{\mathrm{T}}$$

which implies

$$\mu^2 \|\boldsymbol{h}_{\mathrm{T}}\|^2 \leq \|\boldsymbol{A}\boldsymbol{h}_{\mathrm{T}}\|^2 \leq L^2 \|\boldsymbol{h}_{\mathrm{T}}\|^2. \tag{23}$$

Combining (20), (21), (22), and (23), we have

$$\langle \boldsymbol{h}, \nabla^2 F_{1a}(\boldsymbol{y})\boldsymbol{h} \rangle$$
$$= \underbrace{\langle \boldsymbol{h}_{\mathrm{T}}, \boldsymbol{A}^\top \boldsymbol{A}\boldsymbol{h}_{\mathrm{T}} \rangle}_{\geq \mu^2 \|\boldsymbol{h}_{\mathrm{T}}\|^2} + 2\langle \boldsymbol{h}_{\mathrm{T}}, \boldsymbol{A}^\top \boldsymbol{A}\boldsymbol{h}_{\mathrm{N}} \rangle + \underbrace{\langle \boldsymbol{h}_{\mathrm{N}}, \boldsymbol{A}^\top \boldsymbol{A}\boldsymbol{h}_{\mathrm{N}} \rangle}_{\geq 0}$$
$$+ \alpha \underbrace{\langle \boldsymbol{h}_{\mathrm{T}}, (\boldsymbol{I} - D\boldsymbol{p}(\boldsymbol{y}))\,\boldsymbol{h}_{\mathrm{T}} \rangle}_{\geq -\frac{s}{\tau-s}\|\boldsymbol{h}_{\mathrm{T}}\|^2} + 2\alpha \underbrace{\langle \boldsymbol{h}_{\mathrm{T}}, (\boldsymbol{I} - D\boldsymbol{p}(\boldsymbol{y}))\,\boldsymbol{h}_{\mathrm{N}} \rangle}_{= \langle \boldsymbol{h}_{\mathrm{T}}, \boldsymbol{h}_{\mathrm{N}} \rangle = 0} + \alpha \underbrace{\langle \boldsymbol{h}_{\mathrm{N}}, (\boldsymbol{I} - D\boldsymbol{p}(\boldsymbol{y}))\,\boldsymbol{h}_{\mathrm{N}} \rangle}_{= \|\boldsymbol{h}_{\mathrm{N}}\|^2}$$
$$\geq \left( \mu^2 - \alpha \frac{s}{\tau-s} \right) \|\boldsymbol{h}_{\mathrm{T}}\|^2 + \alpha\|\boldsymbol{h}_{\mathrm{N}}\|^2 - 2\|\boldsymbol{A}\boldsymbol{h}_{\mathrm{T}}\| \cdot \|\boldsymbol{A}\boldsymbol{h}_{\mathrm{N}}\|$$
$$\geq \left( \mu^2 - \alpha \frac{s}{\tau-s} \right) \|\boldsymbol{h}_{\mathrm{T}}\|^2 + \alpha\|\boldsymbol{h}_{\mathrm{N}}\|^2 - 2L\|\boldsymbol{h}_{\mathrm{T}}\| \cdot \sigma_{\max}\|\boldsymbol{h}_{\mathrm{N}}\|$$
$$= [\|\boldsymbol{h}_{\mathrm{T}}\| \quad \|\boldsymbol{h}_{\mathrm{N}}\|] \begin{bmatrix} \mu^2 - \alpha\frac{s}{\tau-s} & -\sigma_{\max}L \\ -\sigma_{\max}L & \alpha \end{bmatrix} \begin{bmatrix} \|\boldsymbol{h}_{\mathrm{T}}\| \\ \|\boldsymbol{h}_{\mathrm{N}}\| \end{bmatrix}$$

Therefore, to ensure $\langle \boldsymbol{h}, \nabla^2 F_{1a}(\boldsymbol{y})\boldsymbol{h} \rangle > 0$ for any $\boldsymbol{h} \neq \boldsymbol{0}$, it's enough to ensure the $2 \times 2$ matrix to be positive definite:

$$\mu^2 - \alpha\frac{s}{\tau-s} > 0 \quad \text{and} \quad \alpha\left( \mu^2 - \alpha\frac{s}{\tau-s} \right) - \sigma_{\max}^2 L^2 > 0. \tag{24}$$

In other words, (24) will guarantee the positive definiteness of $\nabla^2 F_{1a}(\boldsymbol{y})$ for all $\boldsymbol{y} \in \overline{\mathbb{U}}_s(\mathbb{M})$ and any $s < \tau$.

**Step 4: Uniqueness of minimizers of $F_{1a}$.** In this step, we will combine the results from Steps 2 and 3. Then we are able to prove that the objective function $F_{1a}(\boldsymbol{y})$ is strictly convex in a neighborhood of its minimizers, which implies the uniqueness of the minimizer. To achieve this, it's enough to ensure

$$\|\hat{\boldsymbol{y}} - \boldsymbol{y}_*\| \leq s \tag{25}$$

for all $\hat{\boldsymbol{y}} \in \arg\min_{\boldsymbol{y}} F_{1a}(\boldsymbol{y})$, where $s$ satisfies (24). With this condition (25), it holds that

$$\hat{\boldsymbol{y}} \in \mathbb{B}(\boldsymbol{y}_*, s) \subset \overline{\mathbb{U}}_s(\mathbb{M}).$$

Along with the fact that $\mathbb{B}(\boldsymbol{y}_*, s)$ is convex and that $\nabla^2 F_{1a}(\boldsymbol{y})$ is positive definite for all $\boldsymbol{y} \in \overline{\mathbb{U}}_s(\mathbb{M})$, $F_{1a}$ is strictly convex within $\mathbb{B}(\boldsymbol{y}_*, s)$ (Boyd & Vandenberghe, 2004, Section 3.1.4). As all minimizers of the strict convex function belong to this convex set, $\mathbb{B}(\boldsymbol{y}_*, s)$, the minimizer $\hat{\boldsymbol{y}}$ must be unique.

Now the question is: How to guarantee (25)? According to (19), Condition (18) along with

$$\left(1 + \frac{\sigma_{\max}}{\mu}\right) r + \frac{2}{\mu}\|\boldsymbol{n}\| \leq s \tag{26}$$

can guarantee (25). Finally, it's enough to choose $\alpha$, $s$, and $r$ such that (18), (24), and (26) are satisfied together. In particular, we choose

$$s = \frac{4}{\mu}\|\boldsymbol{n}\|, \quad r = \frac{1}{\sigma_{\max}}\|\boldsymbol{n}\|, \quad \alpha = \frac{2\sigma_{\max}^2 L^2}{\mu^2}$$

where $\alpha$ merely depends on $\boldsymbol{A}$ and $\mathbb{M}$ but is independent of $\boldsymbol{x}$. Such a parameter choice implies (26):

$$\left(1 + \frac{\sigma_{\max}}{\mu}\right) r + \frac{2}{\mu}\|\boldsymbol{n}\| \leq 2\frac{\sigma_{\max}}{\mu}r + \frac{2}{\mu}\|\boldsymbol{n}\| = \frac{2}{\mu}\|\boldsymbol{n}\| + \frac{2}{\mu}\|\boldsymbol{n}\| = s.$$

As $\|\boldsymbol{n}\| < \frac{1}{20}\frac{\mu^5}{\sigma_{\max}^2 L^2}\tau$, it holds that

$$s = \frac{4}{\mu}\|\boldsymbol{n}\| < \frac{\mu^4}{5\sigma_{\max}^2 L^2}\tau \implies \frac{s}{\tau - s} < \frac{\frac{\mu^4}{5\sigma_{\max}^2 L^2}\tau}{\tau - \frac{\mu^4}{5\sigma_{\max}^2 L^2}\tau} \leq \frac{\frac{\mu^4}{5\sigma_{\max}^2 L^2}\tau}{\tau - \frac{1}{5}\tau} = \frac{\mu^4}{4\sigma_{\max}^2 L^2}$$

and therefore (24) is satisfied:

$$\mu^2 - \alpha\frac{s}{\tau - s} > \mu^2 - \frac{2\sigma_{\max}^2 L^2}{\mu^2}\frac{\mu^4}{4\sigma_{\max}^2 L^2} = \frac{1}{2}\mu^2 > 0$$

and

$$\alpha\left(\mu^2 - \alpha\frac{s}{\tau - s}\right) > \frac{2\sigma_{\max}^2 L^2}{\mu^2} \cdot \frac{1}{2}\mu^2 = \sigma_{\max}^2 L^2.$$

Finally, by choosing $\alpha$ as before, condition (18) is satisfied:

$$\alpha = 2\sigma_{\max}^2 \cdot \frac{L^2}{\mu^2} \geq 2\sigma_{\max}^2 = \frac{2\|\boldsymbol{n}\|^2}{r^2}, \quad r = \frac{1}{\sigma_{\max}}\|\boldsymbol{n}\| < \frac{1}{\sigma_{\max}} \cdot \frac{1}{20}\frac{\mu^5}{\sigma_{\max}^2 L^2}\tau < \tau,$$

which finishes the proof of the uniqueness of minimizers of $F_{1a}$.

**Step 5: Local Lipschitz continuity of $\mathcal{F}_{1a}$.** Previous results from Steps 1-4 indicate that, for any $\boldsymbol{x} \in \mathbb{X}$, there is a unique $\hat{\boldsymbol{y}}(\boldsymbol{x})$ that minimizes $F_{1a}$, but the continuity of $\hat{\boldsymbol{y}}$ w.r.t. $\boldsymbol{x}$ has not been established. In this step, we will show this continuity via the implicit function theorem. Firstly, as $\hat{\boldsymbol{y}}$ minimizes $F_{1a}$, by first-order optimality conditions for smooth minimization, it holds that

$$\nabla F_{1a}(\hat{\boldsymbol{y}}) = \underbrace{\boldsymbol{A}^\top(\boldsymbol{A}\hat{\boldsymbol{y}} - \boldsymbol{x}) + \alpha(\hat{\boldsymbol{y}} - \boldsymbol{p}(\hat{\boldsymbol{y}}))}_{=:\mathcal{H}(\boldsymbol{x},\hat{\boldsymbol{y}})} = \boldsymbol{0}$$

Now, let's pick a point $\boldsymbol{x}_0$ from $\mathbb{X}$. Previous results from Steps 1-4 indicate that, operator $\mathcal{H}(\boldsymbol{x}, \boldsymbol{y})$ is continuously differentiable within a neighborhood of $(\boldsymbol{x}_0, \hat{\boldsymbol{y}}(\boldsymbol{x}_0))$, and its Jacobian matrix w.r.t. $\boldsymbol{y}$

$$D_{\boldsymbol{y}}\mathcal{H}(\boldsymbol{x}, \boldsymbol{y}) = \nabla^2 F_{1a}(\boldsymbol{y})$$

is positive definite within that neighborhood of $(\boldsymbol{x}_0, \hat{\boldsymbol{y}}(\boldsymbol{x}_0))$. Therefore, we are able to apply the implicit function theorem (Folland, 2023, Theorem 3.9) and conclude that $\hat{\boldsymbol{y}}(\boldsymbol{x})$ is Lipschitz continuous within a neighborhood of $\boldsymbol{x}_0$. This argument applies for any points $\boldsymbol{x}_0$ in $\mathbb{X}$. Therefore, $\hat{\boldsymbol{y}} = \mathcal{F}_{1a}(\boldsymbol{x})$ is locally Lipschitz continuous on $\mathbb{X}$. $\qquad\square$

The proof line of Theorem 3.4 largely follows the proof of Theorem 3.3. Here we will highlight the difference of proofs between the two theorems, so that Theorem 3.4 will be rigorously proved without too much redundancy.

*Proof of Theorem 3.4.* For simplicity, we denote the objective function in (4) as $F_{1b}(\boldsymbol{y}, \boldsymbol{z})$:

$$F_{1b}(\boldsymbol{y}, \boldsymbol{z}) := \frac{1}{2}\|\boldsymbol{x} - \boldsymbol{A}\boldsymbol{y}\|^2 + \frac{\alpha}{2}\text{dist}^2(\boldsymbol{z}, \mathbb{M}) + \frac{\beta}{2}\|\boldsymbol{z} - \boldsymbol{y}\|^2,$$

and we will study its properties analogously to $F_{1a}$.

**Step 1: Existence of minimizers of $F_{1b}$.** For any $r > 0$, as

$$\alpha \geq \frac{\|\boldsymbol{n}\|^2}{r^2}, \quad \beta \geq \frac{\|\boldsymbol{n}\|^2}{r^2},$$

it holds that

$$\inf_{\boldsymbol{y},\boldsymbol{z}} F_{1b}(\boldsymbol{y}, \boldsymbol{z}) = \inf_{(\boldsymbol{y},\boldsymbol{z}): \text{ dist}(\boldsymbol{z},\mathbb{M}) \leq r \text{ and } \|\boldsymbol{z}-\boldsymbol{y}\| \leq r} F_{1b}(\boldsymbol{y}, \boldsymbol{z}). \quad (27)$$

This can be proved by contradiction: (I) Suppose $F_{1b}(\hat{\boldsymbol{y}}, \hat{\boldsymbol{z}})$ is lower than the right-hand-side of (27) and $\text{dist}(\hat{\boldsymbol{z}}, \mathbb{M}) > r$, we have

$$F_{1b}(\hat{\boldsymbol{y}}, \hat{\boldsymbol{z}}) \geq 0 + \frac{\|\boldsymbol{n}\|^2}{2r^2}\text{dist}^2(\hat{\boldsymbol{z}}, \mathbb{M}) + 0 > \frac{1}{2}\|\boldsymbol{n}\|^2 = F_{1b}(\boldsymbol{y}_*, \boldsymbol{y}_*)$$

which contradicts with the hypothesis regarding $(\hat{\boldsymbol{y}}, \hat{\boldsymbol{z}})$. (II) Suppose $F_{1b}(\hat{\boldsymbol{y}}, \hat{\boldsymbol{z}})$ is lower than the right-hand-side of (27) and $\|\hat{\boldsymbol{z}} - \hat{\boldsymbol{y}}\| > r$, we have

$$F_{1b}(\hat{\boldsymbol{y}}, \hat{\boldsymbol{z}}) \geq 0 + 0 + \frac{\|\boldsymbol{n}\|^2}{2r^2}\|\hat{\boldsymbol{z}} - \hat{\boldsymbol{y}}\|^2 > \frac{1}{2}\|\boldsymbol{n}\|^2 = F_{1b}(\boldsymbol{y}_*, \boldsymbol{y}_*)$$

which also derives a contradiction. Arguments in (I) and (II) together prove (27). Similar to the proof of Theorem 3.3, (27) implies the existence of minimizers of $F_{1b}$ (i.e., minimizers are attainable.)

**Step 2: Bound of minimizers of $F_{1b}$.** To extend the proof regarding $F_{1a}$ to $F_{1b}$, we consider the following inequality that holds for all $\boldsymbol{y}, \boldsymbol{z} \in \mathbb{U}_\tau(\mathbb{M})$

$$\|\boldsymbol{y} - \boldsymbol{p}(\boldsymbol{y})\| \leq \|\boldsymbol{y} - \boldsymbol{p}(\boldsymbol{z})\| \leq \|\boldsymbol{y} - \boldsymbol{z}\| + \|\boldsymbol{z} - \boldsymbol{p}(\boldsymbol{z})\| = \|\boldsymbol{y} - \boldsymbol{z}\| + \text{dist}(\boldsymbol{z}, \mathbb{M}) \leq 2r.$$

Therefore, we need $2r < \tau$ and

$$\alpha \geq \frac{\|\boldsymbol{n}\|^2}{r^2} > \frac{4\|\boldsymbol{n}\|^2}{\tau^2}, \quad \beta \geq \frac{\|\boldsymbol{n}\|^2}{r^2} > \frac{4\|\boldsymbol{n}\|^2}{\tau^2} \quad (28)$$

to ensure $\hat{\boldsymbol{y}}, \hat{\boldsymbol{z}} \in \mathbb{U}_\tau(\mathbb{M})$. Following the same argument as the proof of Theorem 3.3, the above condition (28) implies

$$\|\boldsymbol{p}(\hat{\boldsymbol{y}}) - \boldsymbol{y}_*\| \leq \frac{\sigma_{\max}}{\mu}(2r) + \frac{2}{\mu}\|\boldsymbol{n}\|$$

and hence

$$\|\hat{\boldsymbol{y}} - \boldsymbol{y}_*\| \leq \|\hat{\boldsymbol{y}} - \boldsymbol{p}(\hat{\boldsymbol{y}})\| + \|\boldsymbol{p}(\hat{\boldsymbol{y}}) - \boldsymbol{y}_*\| \leq 2\left(1 + \frac{\sigma_{\max}}{\mu}\right)r + \frac{2}{\mu}\|\boldsymbol{n}\| \quad (29)$$

and

$$\|\hat{\boldsymbol{z}} - \boldsymbol{y}_*\| \leq \|\hat{\boldsymbol{z}} - \hat{\boldsymbol{y}}\| + \|\hat{\boldsymbol{y}} - \boldsymbol{y}_*\| \leq \left(3 + 2\frac{\sigma_{\max}}{\mu}\right)r + \frac{2}{\mu}\|\boldsymbol{n}\| \quad (30)$$

holds for all $(\hat{\boldsymbol{y}}, \hat{\boldsymbol{z}})$ that minimizes $F_{1b}(\boldsymbol{y}, \boldsymbol{z})$.

**Step 3: Positive definiteness of the Hessian of $F_{1b}$.** Function $F_{1b}(\boldsymbol{y}, \boldsymbol{z})$'s Hessian matrix is of size $2n \times 2n$ and can be written as a $2 \times 2$ block w.r.t. $\boldsymbol{y}$ and $\boldsymbol{z}$:

$$\nabla^2 F_{1b}(\boldsymbol{y}, \boldsymbol{z}) = \begin{bmatrix} \boldsymbol{A}^\top \boldsymbol{A} & \boldsymbol{0} \\ \boldsymbol{0} & \boldsymbol{0} \end{bmatrix} + \alpha \begin{bmatrix} \boldsymbol{0} & \boldsymbol{0} \\ \boldsymbol{0} & \boldsymbol{I} - D\boldsymbol{p}(\boldsymbol{z}) \end{bmatrix} + \beta \begin{bmatrix} \boldsymbol{I} & -\boldsymbol{I} \\ -\boldsymbol{I} & \boldsymbol{I} \end{bmatrix}$$

For any $\boldsymbol{h} = [\boldsymbol{u}^\top \ \boldsymbol{v}^\top]^\top \in \mathbb{R}^{2n}$, the quadratic form $\langle \boldsymbol{h}, \nabla^2 F_{1b}(\boldsymbol{y}, \boldsymbol{z})\boldsymbol{h} \rangle$ can be calculated through:

$$\langle \boldsymbol{h}, \nabla^2 F_{1b}(\boldsymbol{y}, \boldsymbol{z})\boldsymbol{h} \rangle = \boldsymbol{u}^\top \boldsymbol{A}^\top \boldsymbol{A}\boldsymbol{u} + \alpha \boldsymbol{v}^\top (\boldsymbol{I} - D\boldsymbol{p}(\boldsymbol{z}))\boldsymbol{v} + \beta\|\boldsymbol{u} - \boldsymbol{v}\|^2$$

Decompose $\boldsymbol{u} = \boldsymbol{u}_{\mathrm{T}} + \boldsymbol{u}_{\mathrm{N}}$ and $\boldsymbol{v} = \boldsymbol{v}_{\mathrm{T}} + \boldsymbol{v}_{\mathrm{N}}$ in $\mathbb{T}_{\boldsymbol{p}(\boldsymbol{z})}(\mathbb{M}) \oplus \mathbb{N}_{\boldsymbol{p}(\boldsymbol{z})}(\mathbb{M})$. Using the same argument as the proof of Theorem 3.3, we have

$$
\begin{aligned}
&\langle \boldsymbol{h}, \nabla^2 F_{1b}(\boldsymbol{y}, \boldsymbol{z})\boldsymbol{h}\rangle \\
&\geq \left( \mu^2 \|\boldsymbol{u}_{\mathrm{T}}\|^2 - 2\sigma_{\max} L\|\boldsymbol{u}_{\mathrm{T}}\|\|\boldsymbol{u}_{\mathrm{N}}\| \right) + \alpha\left( -\frac{s}{\tau - s}\|\boldsymbol{v}_{\mathrm{T}}\|^2 + \|\boldsymbol{v}_{\mathrm{N}}\|^2 \right) + \beta\|\boldsymbol{u} - \boldsymbol{v}\|^2
\end{aligned}
$$

which implies

$$
\begin{aligned}
&\langle \boldsymbol{h}, \nabla^2 F_{1b}(\boldsymbol{y}, \boldsymbol{z})\boldsymbol{h}\rangle \\
&\geq \left( \mu^2\|\boldsymbol{u}_{\mathrm{T}}\|^2 - 2\sigma_{\max} L\|\boldsymbol{u}_{\mathrm{T}}\|\|\boldsymbol{u}_{\mathrm{N}}\| \right) \\
&\quad + \alpha\left( -\frac{s}{\tau-s}\|\boldsymbol{v}_{\mathrm{T}}\|^2 + \|\boldsymbol{v}_{\mathrm{N}}\|^2 \right) + \beta\left( \|\boldsymbol{u}_{\mathrm{T}} - \boldsymbol{v}_{\mathrm{T}}\|^2 + \|\boldsymbol{u}_{\mathrm{N}} - \boldsymbol{v}_{\mathrm{N}}\|^2 \right) \\
&\geq \left( \mu^2\|\boldsymbol{u}_{\mathrm{T}}\|^2 - 2\sigma_{\max} L\|\boldsymbol{u}_{\mathrm{T}}\|\|\boldsymbol{u}_{\mathrm{N}}\| \right) \\
&\quad + \alpha\left( -\frac{s}{\tau-s}\|\boldsymbol{v}_{\mathrm{T}}\|^2 + \|\boldsymbol{v}_{\mathrm{N}}\|^2 \right) + \beta\left( (\|\boldsymbol{u}_{\mathrm{T}}\| - \|\boldsymbol{v}_{\mathrm{T}}\|)^2 + (\|\boldsymbol{u}_{\mathrm{N}}\| - \|\boldsymbol{v}_{\mathrm{N}}\|)^2 \right) \\
&= [\|\boldsymbol{u}_{\mathrm{T}}\| \quad \|\boldsymbol{u}_{\mathrm{N}}\| \quad \|\boldsymbol{v}_{\mathrm{T}}\| \quad \|\boldsymbol{v}_{\mathrm{N}}\|] \underbrace{\begin{bmatrix} \mu^2 + \beta & -\sigma_{\max}L & -\beta & \\ -\sigma_{\max}L & \beta & & -\beta \\ -\beta & & \beta - \alpha\frac{s}{\tau-s} & \\ & -\beta & & \alpha + \beta \end{bmatrix}}_{=:\boldsymbol{B}} \begin{bmatrix} \|\boldsymbol{u}_{\mathrm{T}}\| \\ \|\boldsymbol{u}_{\mathrm{N}}\| \\ \|\boldsymbol{v}_{\mathrm{T}}\| \\ \|\boldsymbol{v}_{\mathrm{N}}\| \end{bmatrix}
\end{aligned}
$$

To ensure the positive definiteness of $\nabla^2 F_{1b}(\boldsymbol{y}, \boldsymbol{z})$, it's enough to ensure $\boldsymbol{B} \succ \boldsymbol{0}$. For simplicity, we define

$$
\theta := \alpha\frac{s}{\tau - s}, \quad \boldsymbol{B}_1 := \begin{bmatrix} \mu^2 + \beta & -\sigma_{\max}L \\ -\sigma_{\max}L & \beta \end{bmatrix} \quad \boldsymbol{B}_2 := \begin{bmatrix} -\beta & \\ & -\beta \end{bmatrix} \quad \boldsymbol{B}_3 := \begin{bmatrix} \beta - \theta & \\ & \alpha + \beta \end{bmatrix}
$$

Then $\boldsymbol{B} = \begin{bmatrix} \boldsymbol{B}_1 & \boldsymbol{B}_2 \\ \boldsymbol{B}_2^\top & \boldsymbol{B}_3 \end{bmatrix}$ is positive definite if and only if $\boldsymbol{B}_3$ and its Schur complement $\boldsymbol{S}$ are both positive definite:

$$
\boldsymbol{B}_3 \succ \boldsymbol{0}, \quad \boldsymbol{S} = \boldsymbol{B}_1 - \boldsymbol{B}_2 \boldsymbol{B}_3^{-1} \boldsymbol{B}_2^\top \succ \boldsymbol{0}
$$

As $\boldsymbol{B}_2$ and $\boldsymbol{B}_3$ are both diagonal, so $\boldsymbol{B}_2 \boldsymbol{B}_3^{-1} \boldsymbol{B}_2^\top$ is straight forward to calculate: $\boldsymbol{B}_2 \boldsymbol{B}_3^{-1} \boldsymbol{B}_2^\top = \mathrm{diag}\left( \frac{\beta^2}{\beta - \theta}, \frac{\beta^2}{\alpha + \beta} \right)$. Then the Schur complement can be calculated:

$$
\boldsymbol{S} = \begin{bmatrix} \mu^2 + \beta - \frac{\beta^2}{\beta - \theta} & -\sigma_{\max}L \\ -\sigma_{\max}L & \beta - \frac{\beta^2}{\alpha + \beta} \end{bmatrix} = \begin{bmatrix} \mu^2 - \frac{\beta\theta}{\beta - \theta} & -\sigma_{\max}L \\ -\sigma_{\max}L & \frac{\alpha\beta}{\alpha + \beta} \end{bmatrix}
$$

Note that $\boldsymbol{B}_3 \succ \boldsymbol{0}$ if.f $\beta > \theta$. Therefore, $\boldsymbol{B} \succ \boldsymbol{0}$ if.f.

$$
\beta > \theta, \quad \mu^2 > \frac{\beta\theta}{\beta - \theta}, \quad \left( \mu^2 - \frac{\beta\theta}{\beta - \theta} \right) \frac{\alpha\beta}{\alpha + \beta} > \sigma_{\max}^2 L^2, \tag{31}
$$

where $\theta = \alpha\frac{s}{\tau - s}$. Finally, we obtain that (31) ensures $\nabla^2 F_{1b}(\boldsymbol{y}, \boldsymbol{z}) \succ \boldsymbol{0}$ for all $\boldsymbol{y} \in \mathbb{R}^n$ and all $\boldsymbol{z} \in \overline{\mathbb{U}}_s(\mathbb{M})$ with $s < \tau$.

**Step 4: Uniqueness of minimizers of $F_{1b}$.** Comparable to the Step 4 in Theorem 3.3, we need $\|\hat{\boldsymbol{z}} - \boldsymbol{y}_*\| \leq s$ for all $(\hat{\boldsymbol{y}}, \hat{\boldsymbol{z}}) \in \arg\min F_{1b}(\boldsymbol{y}, \boldsymbol{z})$. Based on (30), it's enough to guarantee

$$
\left( 3 + 2\frac{\sigma_{\max}}{\mu} \right) r + \frac{2}{\mu}\|\boldsymbol{n}\| \leq s \tag{32}
$$

Now we choose

$$
s = \frac{4}{\mu}\|\boldsymbol{n}\|, \quad r = \frac{2}{5\sigma_{\max}}\|\boldsymbol{n}\|
$$

which directly satisfies (32). As $\|\boldsymbol{n}\| < \frac{1}{76}\frac{\mu^5}{\sigma_{\max}^2 L^2}\tau$, we have

$$s = \frac{4}{\mu}\|\boldsymbol{n}\| < \frac{1}{19}\frac{\mu^4}{\sigma_{\max}^2 L^2}\tau, \quad \frac{s}{\tau - s} < \frac{\frac{1}{19}\frac{\mu^4}{\sigma_{\max}^2 L^2}\tau}{\tau - \frac{1}{19}\frac{\mu^4}{\sigma_{\max}^2 L^2}\tau} \leq \frac{\frac{1}{19}\frac{\mu^4}{\sigma_{\max}^2 L^2}\tau}{\tau - \frac{1}{19}\tau} = \frac{1}{18}\frac{\mu^4}{\sigma_{\max}^2 L^2}$$

As long as we take

$$\alpha = \frac{9\sigma_{\max}^2 L^2}{\mu^2}, \quad \beta \geq \max\left(\alpha, \frac{3}{2}\mu^2\right)$$

it holds that

$$\theta = \alpha\frac{s}{\tau - s} < \frac{9\sigma_{\max}^2 L^2}{\mu^2}\frac{1}{18}\frac{\mu^4}{\sigma_{\max}^2 L^2} = \frac{1}{2}\mu^2$$

which implies $\beta > 3\theta$ and hence $\beta > \theta$. Moreover, we can verify the remaining part of (31):

$$\frac{\beta\theta}{\beta - \theta} < \frac{\beta\theta}{\beta - \beta/3} = \frac{3}{2}\theta < \frac{3}{4}\mu^2 < \mu^2,$$

$$\left(\mu^2 - \frac{\beta\theta}{\beta - \theta}\right)\frac{\alpha\beta}{\alpha + \beta} > \left(\mu^2 - \frac{3}{4}\mu^2\right)\frac{\alpha\beta}{\beta + \beta} = \frac{1}{8}\mu^2\alpha = \frac{1}{8}\mu^2 \cdot \frac{9\sigma_{\max}^2 L^2}{\mu^2} > \sigma_{\max}^2 L^2.$$

which finishes the proof of (31). Finally, it's enough to verify (28):

$$2r \leq \frac{\|\boldsymbol{n}\|}{\sigma_{\max}} \leq \frac{1}{76}\frac{\mu^5}{\sigma_{\max}^3 L^2}\tau < \tau, \quad \frac{\|\boldsymbol{n}\|^2}{r^2} = \frac{25}{4}\sigma_{\max}^2 \leq \alpha \leq \beta,$$

which finishes Step 4, and concludes the uniqueness of $(\hat{\boldsymbol{y}}, \hat{\boldsymbol{z}})$.

**Step 5: Local Lipschitz continuity of $\mathcal{F}_{1b}$.** By largely following Step 5 in the proof of Theorem 3.3 and changing $\nabla^2 F_{1a}(\boldsymbol{y})$ to $\nabla^2 F_{1b}(\boldsymbol{y}, \boldsymbol{z})$, one can directly conclude that the mapping $\mathcal{F}_{1b}$ is locally Lipschitz continuous on $\mathbb{X}$. $\square$

## C.1 PROXIMAL OPERATOR NEAR A MANIFOLD

We collect here the definition and basic properties of the proximal map used in the main text and relate them to the convergence condition proposed in Ryu et al. (2019).

**Theorem C.1** (Contractivity of the proximal residual near a $\mathcal{C}^2$ manifold). *Let $\mathbb{M} \subset \mathbb{R}^n$ be a compact $\mathcal{C}^2$ embedded submanifold with reach $\tau > 0$. For $\sigma > 0$ define, for each $\boldsymbol{z} \in \mathbb{U}_\tau(\mathbb{M})$,*

$$\phi_\sigma(\boldsymbol{y}, \boldsymbol{z}) := \frac{\sigma}{2}\text{dist}^2(\boldsymbol{y}, \mathbb{M}) + \frac{1}{2}\|\boldsymbol{y} - \boldsymbol{z}\|^2.$$

*Then $\phi_\sigma$ must yield a unique minimizer, and hence we are able to define*

$$\text{prox}_\sigma(\boldsymbol{z}) := \arg\min_{\boldsymbol{y}} \phi_\sigma(\boldsymbol{y}, \boldsymbol{z}), \qquad \mathcal{S}_\sigma(\boldsymbol{z}) := \text{prox}_\sigma(\boldsymbol{z}) - \boldsymbol{z}.$$

*Then $\mathcal{S}_\sigma$ is a contractive operator within a tubular neighborhood of $\mathbb{M}$. In particular, it holds that*

$$\|\mathcal{S}_\sigma(\boldsymbol{z}) - \mathcal{S}_\sigma(\boldsymbol{z}')\| \leq \frac{\sigma}{1 + \sigma}\|\boldsymbol{z} - \boldsymbol{z}'\| \tag{33}$$

*for all $\boldsymbol{z}, \boldsymbol{z}' \in \mathbb{U}_r(\mathbb{M})$ where $r \leq \tau/4$ and $\|\boldsymbol{z} - \boldsymbol{z}'\| \leq \tau/4$.*

Relation to plug-and-play (PnP): Condition (A) of Ryu et al. (2019) assumes a (nearly) contractive *denoiser residual*—precisely the kind of property (33) guarantees for the proximal residual $\text{prox}_\sigma - \boldsymbol{I}$ on a neighborhood of $\mathbb{M}$. In practice, $\mathbb{M}$ is unknown; one therefore learns a parameterized operator (e.g., a neural network) whose residual is constrained to be (nearly) $\sigma$-contractive and plugs it into PGD/HQS in place of the exact proximal map. Whereas Ryu et al. (2019) posits Condition (A) to ensure convergence, Theorem C.1 shows this condition arises naturally when the prior corresponds to the manifold-penalty $\frac{\sigma}{2}\text{dist}^2(\cdot, \mathbb{M})$.

*Proof of Theorem C.1.* We first note that, for any $\boldsymbol{y}$, if $\|\boldsymbol{y} - \boldsymbol{z}\| > \|\boldsymbol{z} - \boldsymbol{p}(\boldsymbol{z})\|$, then it holds that

$$\phi_\sigma(\boldsymbol{p}(\boldsymbol{z}), \boldsymbol{z}) = 0 + \frac{1}{2}\|\boldsymbol{z} - \boldsymbol{p}(\boldsymbol{z})\|^2 < \frac{\sigma}{2}\text{dist}^2(\boldsymbol{y}, \mathbb{M}) + \frac{1}{2}\|\boldsymbol{y} - \boldsymbol{z}\|^2 = \phi_\sigma(\boldsymbol{y}, \boldsymbol{z})$$

which implies

$$\inf_{\boldsymbol{y}} \phi_\sigma(\boldsymbol{y}, \boldsymbol{z}) = \inf_{\boldsymbol{y}:\|\boldsymbol{y}-\boldsymbol{z}\|\leq\|\boldsymbol{z}-\boldsymbol{p}(\boldsymbol{z})\|} \phi_\sigma(\boldsymbol{y}, \boldsymbol{z})$$

Let $r = \|\boldsymbol{z} - \boldsymbol{p}(\boldsymbol{z})\|$. We further notice that, for any $\boldsymbol{y}$ with $\|\boldsymbol{y} - \boldsymbol{z}\| = s \leq r$, we are able to define $\tilde{\boldsymbol{y}}$

$$\tilde{\boldsymbol{y}} := \frac{r-s}{r}\boldsymbol{z} + \frac{s}{r}\boldsymbol{p}(\boldsymbol{z})$$

which satisfies $\boldsymbol{p}(\tilde{\boldsymbol{y}}) = \boldsymbol{p}(\boldsymbol{z})$ and hence it holds that

$$\begin{aligned}
\text{dist}(\tilde{\boldsymbol{y}}, \mathbb{M}) = \|\tilde{\boldsymbol{y}} - \boldsymbol{p}(\boldsymbol{z})\| =& \|\boldsymbol{z} - \boldsymbol{p}(\boldsymbol{z})\| - \|\tilde{\boldsymbol{y}} - \boldsymbol{z}\| \\
<& \|\boldsymbol{z} - \boldsymbol{p}(\boldsymbol{y})\| - \|\tilde{\boldsymbol{y}} - \boldsymbol{z}\| \\
\leq& \|\boldsymbol{z} - \boldsymbol{y}\| + \|\boldsymbol{y} - \boldsymbol{p}(\boldsymbol{y})\| - \|\tilde{\boldsymbol{y}} - \boldsymbol{z}\| \\
=& s + \|\boldsymbol{y} - \boldsymbol{p}(\boldsymbol{y})\| - s = \|\boldsymbol{y} - \boldsymbol{p}(\boldsymbol{y})\| = \text{dist}(\boldsymbol{y}, \mathbb{M})
\end{aligned}$$

which implies

$$\phi_\sigma(\tilde{\boldsymbol{y}}, \boldsymbol{z}) = \frac{\sigma}{2}\text{dist}^2(\tilde{\boldsymbol{y}}, \mathbb{M}) + \frac{1}{2}\|\tilde{\boldsymbol{y}} - \boldsymbol{z}\|^2 < \frac{\sigma}{2}\text{dist}^2(\boldsymbol{y}, \mathbb{M}) + \frac{1}{2}\|\boldsymbol{y} - \boldsymbol{z}\|^2 = \phi_\sigma(\boldsymbol{y}, \boldsymbol{z})$$

Consequently, we conclude that minimizing $\phi_\sigma$ is equal to minimizing it over the line segment between $\boldsymbol{z}$ and its projection $\boldsymbol{p}(\boldsymbol{z})$:

$$\inf_{\boldsymbol{y}} \phi_\sigma(\boldsymbol{y}, \boldsymbol{z}) = \inf_{\xi \in [0,1]} \phi_\sigma(\xi\boldsymbol{z} + (1-\xi)\boldsymbol{p}(\boldsymbol{z}), \boldsymbol{z}).$$

Now define $\psi(\xi) = \phi_\sigma(\xi\boldsymbol{z} + (1-\xi)\boldsymbol{p}(\boldsymbol{z}), \boldsymbol{z})$. We have

$$\begin{aligned}
\psi(\xi) =& \frac{\sigma}{2}\left\|\left(\xi\boldsymbol{z} + (1-\xi)\boldsymbol{p}(\boldsymbol{z})\right) - \boldsymbol{p}(\boldsymbol{z})\right\|^2 + \frac{1}{2}\left\|\left(\xi\boldsymbol{z} + (1-\xi)\boldsymbol{p}(\boldsymbol{z})\right) - \boldsymbol{z}\right\|^2 \\
=& \frac{\sigma}{2}\xi^2\|\boldsymbol{z} - \boldsymbol{p}(\boldsymbol{z})\|^2 + \frac{1}{2}(1-\xi)^2\|\boldsymbol{z} - \boldsymbol{p}(\boldsymbol{z})\|^2 \\
=& \left(\sigma\xi^2 + (1-\xi)^2\right) \cdot \frac{1}{2}\|\boldsymbol{z} - \boldsymbol{p}(\boldsymbol{z})\|^2
\end{aligned}$$

Therefore, $\inf_{\xi \in [0,1]} \psi(\xi)$ is attainable, and the minimizer is $\xi_* = \frac{1}{1+\sigma}$, which implies $\phi_\sigma$ must yield a unique minimizer at

$$\boldsymbol{y}_* = \frac{\boldsymbol{z} + \sigma\boldsymbol{p}(\boldsymbol{z})}{1 + \sigma}.$$

Consequently, we have

$$\mathcal{S}_\sigma(\boldsymbol{z}) = \boldsymbol{y}_* - \boldsymbol{z} = \frac{\sigma}{1+\sigma}(\boldsymbol{p}(\boldsymbol{z}) - \boldsymbol{z})$$

and hence

$$D\mathcal{S}_\sigma(\boldsymbol{z}) = \frac{\sigma}{1+\sigma}(D\boldsymbol{p}(\boldsymbol{z}) - \boldsymbol{I}).$$

According to (Leobacher & Steinicke, 2021, Theorem C), $D\boldsymbol{p}(\boldsymbol{z})$ is actually restricted to the tangent space $\mathbb{T}_{\boldsymbol{p}(\boldsymbol{z})(\mathbb{M})}$:

$$D\boldsymbol{p}(\boldsymbol{z}) = \left(\boldsymbol{I}_{\mathbb{T}_{\boldsymbol{p}(\boldsymbol{z})(\mathbb{M})}} - r\mathcal{L}_{\boldsymbol{p}(\boldsymbol{z}),\boldsymbol{v}}\right)^{-1} P_{\mathbb{T}_{\boldsymbol{p}(\boldsymbol{z})(\mathbb{M})}}$$

where $r = \|\boldsymbol{p}(\boldsymbol{z}) - \boldsymbol{z}\|$, $\boldsymbol{v} = (\boldsymbol{p}(\boldsymbol{z}) - \boldsymbol{z})/r$, and $\mathcal{L}_{\boldsymbol{p}(\boldsymbol{z}),\boldsymbol{v}}$ is the shape operator in direction $\boldsymbol{v}$ at $\boldsymbol{p}(\boldsymbol{z})$. The shape operator's eigenvalues $\kappa_1, \cdots, \kappa_d$ (In this context, $d$ means the dimension of the tangent space) are the principal curvatures of $\mathbb{M}$ (Do Carmo, 2016), which implies the eigenvalues of $D\boldsymbol{p}(\boldsymbol{z})$, when restricted to the tangent space, are

$$\frac{1}{1 - r\kappa_1}, \cdots, \frac{1}{1 - r\kappa_d}.$$

All the curvatures are bounded by the reciprocal of the reach: $|\kappa_i| \leq 1/\tau$ (Aamari et al., 2019). Therefore, it holds that

$$\frac{\tau}{\tau + r}\boldsymbol{I}\Big|_{\mathbb{T}_{\boldsymbol{p}(\boldsymbol{z})(\mathbb{M})}} \preceq D\boldsymbol{p}(\boldsymbol{z})\Big|_{\mathbb{T}_{\boldsymbol{p}(\boldsymbol{z})(\mathbb{M})}} \preceq \frac{\tau}{\tau - r}\boldsymbol{I}\Big|_{\mathbb{T}_{\boldsymbol{p}(\boldsymbol{z})(\mathbb{M})}}.$$

Moreover, as $D\boldsymbol{p}(\boldsymbol{z})$ is restricted to and acts only on the tangent space $\mathbb{T}_{\boldsymbol{p}(\boldsymbol{z})(\mathbb{M})}$, we have $\boldsymbol{0} \preceq D\boldsymbol{p}(\boldsymbol{z}) \preceq \frac{\tau}{\tau - r}\boldsymbol{I}$, which implies

$$-\boldsymbol{I} \preceq D\boldsymbol{p}(\boldsymbol{z}) - \boldsymbol{I} \preceq \frac{r}{\tau - r}\boldsymbol{I}.$$

For $r \leq \tau/2$, we have $\frac{r}{\tau - r} \leq 1$ and hence $\|D\mathcal{S}_\sigma(\boldsymbol{z})\| \leq \frac{\sigma}{1+\sigma}$. As long as $\boldsymbol{z}, \boldsymbol{z}' \in \mathbb{U}_r(\mathbb{M})$ where $r \leq \tau/4$ and $\|\boldsymbol{z} - \boldsymbol{z}'\| \leq \tau/4$, the two points $\boldsymbol{z}, \boldsymbol{z}'$ can be included in a convex subset (actually a ball) of $\mathbb{U}_r(\mathbb{M})$ with $r = \tau/2$. By the mean value theorem, we finish the proof of (33). □

## C.2 DISCUSSIONS REGARDING PNP

**Derivation of HQS.** Consider (4):

$$\min_{\boldsymbol{y}, \boldsymbol{z} \in \mathbb{R}^n} \frac{1}{2}\|\boldsymbol{x} - \boldsymbol{A}\boldsymbol{y}\|^2 + \frac{\alpha}{2}\operatorname{dist}^2(\boldsymbol{z}, \mathbb{M}) + \frac{\beta}{2}\|\boldsymbol{y} - \boldsymbol{z}\|^2.$$

A typically method to solve it is applying block coordinate descent on it, which is also named "Half-quadratic-splitting (HQS)" in the literature (Yang, 1995):

$$\boldsymbol{y}_{t+1} = \arg\min_{\boldsymbol{y} \in \mathbb{R}^n} \frac{1}{2}\|\boldsymbol{A}\boldsymbol{y} - \boldsymbol{x}\|^2 + \frac{\beta}{2}\|\boldsymbol{y} - \boldsymbol{z}_t\|^2 = \left(\boldsymbol{A}^\top\boldsymbol{A} + \beta\boldsymbol{I}\right)^{-1}\left(\boldsymbol{A}^\top\boldsymbol{x} + \beta\boldsymbol{z}_t\right)$$

$$\boldsymbol{z}_{t+1} = \arg\min_{\boldsymbol{z} \in \mathbb{R}^n} \frac{\alpha}{2}\operatorname{dist}^2(\boldsymbol{z}, \mathbb{M}) + \frac{\beta}{2}\|\boldsymbol{z} - \boldsymbol{y}_{t+1}\|^2 = \operatorname{prox}_\sigma(\boldsymbol{y}_{t+1}) \quad (\text{let } \sigma = \alpha/\beta)$$

Similarly, we can parameterize $\operatorname{prox}_\sigma$ as a neural network $\mathcal{H}_{\theta,\sigma}$. Therefore, HQS suggests an implicit model

$$\mathcal{G}_\Theta(\boldsymbol{z}, \boldsymbol{x}) = \mathcal{H}_{\theta,\sigma}\left(\left(\boldsymbol{A}^\top\boldsymbol{A} + \beta\boldsymbol{I}\right)^{-1}\left(\boldsymbol{A}^\top\boldsymbol{x} + \beta\boldsymbol{z}\right)\right)$$

where $\Theta = \{\theta, \sigma, \beta\}$ includes all trainable parameters, which derives (6).

**Bibliographical notes.** Here we adopt the long-standing "plug-in denoiser" idea. It originated with Plug-and-Play (PnP) ADMM, which replaces a proximal operator with an off-the-shelf denoiser inside ADMM (Venkatakrishnan et al., 2013). The framework has since been developed and analyzed extensively—see, e.g., (Chan et al., 2016; Kamilov et al., 2017; Buzzard et al., 2018; Sun et al., 2019) and the recent survey (Kamilov et al., 2023). In the PGD setting, one pretrains $\mathcal{H}$ for Gaussian denoising and plugs it into (5) (Ryu et al., 2019; Gavaskar & Chaudhury, 2020; Liu et al., 2021; Hurault et al., 2022b). The same plug-in idea applies to HQS via (6) (Zhang et al., 2021; Hurault et al., 2022a; Rasti-Meymandi et al., 2023). In contrast to training a denoiser off-the-shelf and plugging it in, one can train the *entire* $\mathcal{G}_\Theta$ via deep equilibrium methods for the target task (the approach closest to this paper) in both PGD-style (Gilton et al., 2021; Winston & Kolter, 2020; Zou et al., 2023; Yu & Dansereau, 2024; Daniele et al., 2025; Shenoy et al., 2025) and HQS-style (Gkillas et al., 2023).

## D PROOFS REGARDING NS EQUATIONS

To rigorously state and prove the theorems, we present some definitions here. First, We denote by $H^m(\Omega)$ the Sobolev space of functions which are in $L^2(\Omega)$ together with all their derivatives of order $\leq m$. Then $H_{\mathrm{p}}^m(\Omega) \subset H^m(\Omega)$ is the collection of functions in $H^m(\Omega)$ that satisfies the periodic boundary condition on $\Omega$ with zero mean (ref. to (Temam, 1995, Remark 1.1)). Then, we can define the spaces considered in this paper:

$$\mathbb{H} := \left\{u \in \left\{H_{\mathrm{p}}^0(\Omega)\right\}^2 : \nabla \cdot u = 0\right\}, \quad \mathbb{V} := \left\{u \in \left\{H_{\mathrm{p}}^1(\Omega)\right\}^2 : \nabla \cdot u = 0\right\}$$

For the NS equation (7), we consider $f \in \mathbb{H}$ and $u \in \mathbb{V}$. Moreover, we denote $\mathbb{V}'$ as the dual space of $\mathbb{V}$ and have

$$\mathbb{V} \subset \mathbb{H} \subset \mathbb{V}'.$$

We then equip $\mathbb{H}$ with the standard $L^2$ inner product and norm for vector fields:

$$\langle u, v \rangle_{\mathbb{H}} := \int_{\Omega} \langle u(\xi), v(\xi) \rangle \mathrm{d}\xi, \quad \|u\|_{\mathbb{H}} := \sqrt{\langle u, u \rangle_{\mathbb{H}}} = \left( \int_{\Omega} \|u(\xi)\|^2 \mathrm{d}\xi \right)^{1/2} = \|u\|_{L^2(\Omega)}$$

The space $\mathbb{V}$ is equipped with the $L^2$ norm on the first-order derivatives of $u$. In particular,

$$\langle u, v \rangle_{\mathbb{V}} := \sum_{i=1}^{2} \int_{\Omega} \left\langle \frac{\partial u}{\partial \xi_i}(\xi), \frac{\partial v}{\partial \xi_i}(\xi) \right\rangle \mathrm{d}\xi$$

$$\|u\|_{\mathbb{V}} := \sqrt{\langle u, u \rangle_{\mathbb{V}}} = \left( \sum_{i=1}^{2} \int_{\Omega} \left\| \frac{\partial u}{\partial \xi_i}(x) \right\|^2 \mathrm{d}\xi \right)^{1/2} = \|\nabla u\|_{L^2(\Omega)}$$

and $\|\cdot\|_{\mathbb{V}'}$ is defined as the dual norm of $\|\cdot\|_{\mathbb{V}}$. By Poincare and Cauchy-Shwartz inequalities, we have

$$\|v\|_{\mathbb{H}} \leq c_1 \|v\|_{\mathbb{V}}, \quad \forall v \in \mathbb{V}$$

and

$$\|v\|_{\mathbb{V}'} \leq c_2 \|v\|_{\mathbb{H}}, \quad \forall v \in \mathbb{H}$$

where $c_1, c_2$ are constants depending on the domain $\Omega$. The above definitions and results are standard in the literature and we largely follow the notation in (Temam, 1995, Section 2).

*Proof of Theorem 3.6.* (Temam, 1995, Theorem 10.1) states that, for any $f \in \mathbb{V}'$, if $\|f\|_{\mathbb{V}'} \leq c_0 \nu^2$ (with $c_0 > 0$ depending only on $\Omega$), then the steady NS problem (7) has a unique solution $u_*$. Since $\mathbb{H} \subset \mathbb{V}'$ and $\|f\|_{\mathbb{V}'} \leq c_2 \|f\|_{\mathbb{H}}$, this yields uniqueness on

$$\mathbb{H}_{\nu}^{(1)} := \left\{ f \in \mathbb{H} : \|f\|_{\mathbb{H}} \leq \frac{c_0}{c_2} \nu^2 \right\}.$$

Moreover, by (Temam, 1995, Theorem 10.4), there exists an open dense set $\mathbb{H}_{\nu}^{(2)} \subset \mathbb{H}$ such that, on each connected component of $\mathbb{H}_{\nu}^{(2)}$, the solution $u_*$ depends $C^{\infty}$ on $f$; in particular, $f \mapsto u_*$ is locally Lipschitz there. Define $\mathbb{H}_{\nu} := \mathbb{H}_{\nu}^{(1)} \cap \mathbb{H}_{\nu}^{(2)}$. Since $\mathbb{H}_{\nu}^{(2)}$ is open and dense in $\mathbb{H}$, the set $\mathbb{H}_{\nu}$ is dense in $\mathbb{H}_{\nu}^{(1)}$. On $\mathbb{H}_{\nu}$ the solution is unique and the map $f \mapsto u_*$ is locally Lipschitz. This completes the proof. $\qquad\square$

Before moving to Corollary 3.7, let's reclarify lifting and projection operators: Let the lifting (or extension) operator $\mathcal{E}_h : \mathbb{R}^{N_h \times 2} \to \{L^2(\Omega)\}^2$ be the piecewise–constant reconstruction $\mathcal{E}_h(\boldsymbol{x}) := \sum_{C \in \Omega_h} x_C \mathbf{1}_C$, and let $\mathcal{P} : \{L^2(\Omega)\}^2 \to \mathbb{H}$ be the orthogonal projection onto divergence–free, zero–mean fields. Then we move on to Corollary 3.7.

*Proof of Corollary 3.7.* The mapping $\mathcal{F}_2 : \boldsymbol{x} \mapsto \boldsymbol{y}_*$ can be viewed as a composition of multiple mappings: We first map $\boldsymbol{x} \in \mathbb{R}^{N_h \times 2}$ to a continuous version $f \in \mathbb{H}$ by $\mathcal{P} \circ \mathcal{E}_h$, then $f$ can be mapped to its corresponding solution $u_*$ by a Locally Lipschitz operator as stated in Theorem 3.6. Here we denote this mapping by $\mathcal{S} : f \mapsto u_*$. Then $u_*$ is mapped to $\omega_*$ by vorticity: $\nabla \times u_*$, and finally $\omega_*$ can be mapped to $\boldsymbol{y}_*$ by a restriction operator $\mathcal{R}_h$:

$$\mathcal{F}_2 = \mathcal{R}_h \circ (\nabla \times) \circ \mathcal{S} \circ \mathcal{P} \circ \mathcal{E}_h.$$

Then let's analyze the norm of the above operators one by one. Firstly, the restriction operator $\mathcal{R}_h$ has a norm no greater than 1 as:

$$\|\mathcal{R}_h(\omega)\|_{\ell_h^2}^2 = \sum_{C \in \Omega_h} |C| \left| \frac{1}{|C|} \int_C \omega(\xi) \mathrm{d}\xi \right|^2$$

$$\leq \sum_{C \in \Omega_h} \frac{1}{|C|} \left( \int_C |\omega(\xi)| \mathrm{d}\xi \right)^2 \leq \sum_{C \in \Omega_h} \int_C |\omega(\xi)|^2 \mathrm{d}\xi = \|\omega\|_{L^2(\Omega)}^2$$

Note that $\mathcal{R}_h$ is a linear operator, hence its bounded norm immediately leads to its bounded Lipschitz constant:

$$\|\mathcal{R}_h(\omega) - \mathcal{R}_h(\omega')\|_{\ell_h^2}^2 = \|\mathcal{R}_h(\omega - \omega')\|_{\ell_h^2}^2 \leq \|\omega - \omega'\|_{L^2(\Omega)}^2.$$

Second, the curl operator $\nabla \times$ must be a bounded linear operator because the solution $u_* \in \mathbb{V}$, where first-order derivatives must be $L^2$. Third, the solution mapping $\mathcal{S}$ has been discussed in Theorem 3.6, it is a nonlinear operator, but it is locally Lipschitz continuous. Fourth, the projection operator $\mathcal{P}$ must be linear and have a norm no greater than 1. Finally, the lifting operator is linear and has a bounded norm as:

$$\|\mathcal{E}_h(\boldsymbol{x})\|_{L^2(\Omega)}^2 = \sum_{C \in \Omega_h} |C| \, |x_C|^2 = \|\boldsymbol{x}\|_{\ell_h^2}^2$$

Therefore, except for the nonlinear operator $\mathcal{S}$, the other four operators are all linear and bounded and hence are globally Lipschitz continuous. As long as we can show that the input of $\mathcal{S}$ must be taken from the unique solution regime $\mathbb{H}_\nu$, we will complete the proof that $\mathcal{F}_2$ is locally Lipschitz everywhere on $\mathbb{X}_{\nu,h}$. This can be proved because $\boldsymbol{x} \in \mathbb{X}_{\nu,h}$ implies $\mathcal{P}(\mathcal{E}_h(\boldsymbol{x})) \in \mathbb{H}_\nu$. Finally, by applying Theorem 2.4, we conclude the existence of $\mathcal{G}$ described in Corollary 3.7, which finishes the entire proof. $\qquad\square$

## E  PROOFS REGARDING LINEAR PROGRAMMING

Although Lipschitz continuity of LP solution maps has been studied (e.g., (Mangasarian & Shiau, 1987; Dontchev & Rockafellar, 2009)), we are not aware of a reference that states Theorem 3.8 in the precise form needed here—particularly allowing perturbations of $\boldsymbol{A}$ (rather than treating $\boldsymbol{A}$ as fixed). For completeness, we therefore include a self-contained discussion and proof.

To work with a standard form, we rewrite the general-form problem (8) in standard form. Suppose there are $p$ equality constraints and $q$ inequality constraints. Without loss of generality, we assume $\circ_i$ equals to "$=$" for $1 \leq i \leq p$ and $\circ_i$ equals to "$\leq$" for $p+1 \leq i \leq m$. Then we denote $\boldsymbol{A}_p$ as the first $p$ rows of matrix $\boldsymbol{A}$ and $\boldsymbol{A}_q$ as the remaining part:

$$\boldsymbol{A}_p := \boldsymbol{A}[1:p, :], \qquad \boldsymbol{A}_q := \boldsymbol{A}[p+1:m, :]$$

And therefore the general form LP (8) can be written as

$$\min_{\boldsymbol{y} \in \mathbb{R}^n} \boldsymbol{c}^\top \boldsymbol{y}, \quad \text{s.t. } \boldsymbol{A}_p \boldsymbol{y} = \boldsymbol{b}_p, \;\; \boldsymbol{A}_q \boldsymbol{y} \leq \boldsymbol{b}_q, \;\; \boldsymbol{l} \leq \boldsymbol{y} \leq \boldsymbol{u}.$$

Let $\hat{\boldsymbol{y}} := \boldsymbol{y} - \boldsymbol{l}$, $\boldsymbol{s} := \boldsymbol{b}_q - \boldsymbol{A}_q \boldsymbol{y}$, and $\boldsymbol{t} := \boldsymbol{u} - \boldsymbol{y}$, the above problem can be transformed to

$$\min_{\boldsymbol{y} \in \mathbb{R}^n} \boldsymbol{c}^\top \hat{\boldsymbol{y}}, \quad \text{s.t. } \begin{bmatrix} \boldsymbol{A}_p & & \\ \boldsymbol{A}_q & \boldsymbol{I} & \\ \boldsymbol{I} & & \boldsymbol{I} \end{bmatrix} \begin{bmatrix} \hat{\boldsymbol{y}} \\ \boldsymbol{s} \\ \boldsymbol{t} \end{bmatrix} = \begin{bmatrix} \boldsymbol{b}_p - \boldsymbol{A}_p \boldsymbol{l} \\ \boldsymbol{b}_q - \boldsymbol{A}_q \boldsymbol{l} \\ \boldsymbol{u} - \boldsymbol{l} \end{bmatrix}, \;\; \hat{\boldsymbol{y}} \geq \boldsymbol{0}, \boldsymbol{s} \geq \boldsymbol{0}, \boldsymbol{t} \geq \boldsymbol{0}$$

By letting

$$\tilde{\boldsymbol{c}} := \begin{bmatrix} \boldsymbol{c} \\ \boldsymbol{0} \\ \boldsymbol{0} \end{bmatrix}, \;\; \tilde{\boldsymbol{A}} := \begin{bmatrix} \boldsymbol{b}_p - \boldsymbol{A}_p \boldsymbol{l} \\ \boldsymbol{b}_q - \boldsymbol{A}_q \boldsymbol{l} \\ \boldsymbol{u} - \boldsymbol{l} \end{bmatrix}, \;\; \tilde{\boldsymbol{b}} := \begin{bmatrix} \boldsymbol{b}_p - \boldsymbol{A}_p \boldsymbol{l} \\ \boldsymbol{b}_q - \boldsymbol{A}_q \boldsymbol{l} \\ \boldsymbol{u} - \boldsymbol{l} \end{bmatrix}, \;\; \tilde{\boldsymbol{y}} := \begin{bmatrix} \hat{\boldsymbol{y}} \\ \boldsymbol{s} \\ \boldsymbol{t} \end{bmatrix}$$

The problem is equivalently expressed in standard form as

$$\min_{\tilde{\boldsymbol{y}}} \tilde{\boldsymbol{c}}^\top \tilde{\boldsymbol{y}}, \;\; \text{s.t. } \tilde{\boldsymbol{A}} \tilde{\boldsymbol{y}} = \tilde{\boldsymbol{b}}, \;\; \tilde{\boldsymbol{y}} \geq \boldsymbol{0}.$$

In fact, every LP can be rewritten in an equivalent standard form. While concepts such as basic feasible solutions, degeneracy, and complementary slackness are most naturally and cleanly stated in standard form, each admits a closely related analogue (with minor adjustments) for the general form. Accordingly—without loss of generality and to keep the focus on core ideas—we carry out the proof in the standard-form setting:

$$\min_{\boldsymbol{y}} \boldsymbol{c}^\top \boldsymbol{y}, \;\; \text{s.t. } \boldsymbol{A} \boldsymbol{y} = \boldsymbol{b}, \;\; \boldsymbol{y} \geq \boldsymbol{0},$$

with dual

$$\min_{\boldsymbol{z}} \boldsymbol{b}^\top \boldsymbol{z}, \;\; \text{s.t. } \boldsymbol{A}^\top \boldsymbol{z} \leq \boldsymbol{c}.$$

Here, we follow the standard settings in the literature: $\boldsymbol{y}, \boldsymbol{c} \in \mathbb{R}^n$, $\boldsymbol{z}, \boldsymbol{b} \in \mathbb{R}^m$, $\boldsymbol{A} \in \mathbb{R}^{m \times n}$, $\text{rank}(\boldsymbol{A}) = m$ (ensured by preprocessing with removing redundant equalities), and $m \leq n$. In this context, we define the domain of LP that we work on:

$$\mathbb{X} := \{(\boldsymbol{A}, \boldsymbol{b}, \boldsymbol{c}) : \text{The resulting standard LP is feasible and bounded}\}$$

Note that, to match the rest of the paper, we reserve $\boldsymbol{x}$ for machine learning model inputs (in this context, it is $\boldsymbol{x} = (\boldsymbol{A}, \boldsymbol{b}, \boldsymbol{c})$) and hence write the primal LP variable as $\boldsymbol{y}$ and the dual LP variable as $\boldsymbol{z}$. This departs from the common $(\boldsymbol{x}, \boldsymbol{y})$ convention. Note also that in the main text the symbol $\boldsymbol{z}$ denotes a latent variable; here, in the appendix regarding LP's technical details, it denotes the dual variable. These meanings are unrelated and should be clear from context.

Now let's present some definitions used in this appendix. Fix a *basis* by selecting an index set $B \subset \{1, 2, \cdots, n\}$ with $|B| = m$ such that the $m \times m$ submatrix $\boldsymbol{B} := \boldsymbol{A}[:, B]$ is *nonsingular*. Let $N = \{1, 2, \cdots, n\} \backslash B$ be the complement of the basis and let $\boldsymbol{N} := \boldsymbol{A}[:, N]$. Then the equality constraints read

$$\boldsymbol{B}\boldsymbol{y}_B + \boldsymbol{N}\boldsymbol{y}_N = \boldsymbol{b}$$

Setting $\boldsymbol{y}_N = \boldsymbol{0}$ yields $\boldsymbol{y}_B = \boldsymbol{B}^{-1}\boldsymbol{b}$. Such a $\boldsymbol{y} = [\boldsymbol{y}_B, \boldsymbol{0}]$ is called a *basic solution*. If additionally $\boldsymbol{y}_B \geq \boldsymbol{0}$, this basic solution is feasible, then it is called a *basic feasible solution (BFS)*. On the dual side, we define the slack variable $\boldsymbol{s}$ and its sub-vector restricted to $B$ and $N$:

$$\boldsymbol{s} := \boldsymbol{c} - \boldsymbol{A}^\top \boldsymbol{z}, \quad \boldsymbol{s}_B := \boldsymbol{c}_B - \boldsymbol{B}^\top \boldsymbol{z}, \quad \boldsymbol{s}_N := \boldsymbol{c}_N - \boldsymbol{N}^\top \boldsymbol{z}.$$

A pair $(\boldsymbol{y}, \boldsymbol{z})$ is primal–dual optimal (i.e., satisfies KKT for LP) iff

$$\boldsymbol{A}\boldsymbol{y} = \boldsymbol{b}, \quad \boldsymbol{c} = \boldsymbol{A}^\top \boldsymbol{z} + \boldsymbol{s}, \quad \boldsymbol{y} \odot \boldsymbol{s} = \boldsymbol{0}, \quad \boldsymbol{y} \geq \boldsymbol{0}, \quad \boldsymbol{s} \geq \boldsymbol{0} \tag{34}$$

for some $\boldsymbol{s} \in \mathbb{R}^n$. If, in addition, there exists a basis $B$ such that

$$\boldsymbol{y}_B \geq \boldsymbol{0}, \quad \boldsymbol{y}_N = \boldsymbol{0}, \quad \boldsymbol{s}_B = \boldsymbol{0}, \quad \boldsymbol{s}_N \geq \boldsymbol{0}, \tag{35}$$

then the tuple $(\boldsymbol{y}, \boldsymbol{z}, \boldsymbol{s})$ is called an optimal BFS with a complementary dual. By the fundamental theorem of linear programming, any feasible instance with finite optimal value $(\boldsymbol{A}, \boldsymbol{b}, \boldsymbol{c}) \in \mathbb{X}$ admits an optimal BFS with a complementary dual satisfying (34) and (35) together (Bertsimas & Tsitsiklis, 1997).

While conditions (34) and (35) are enough to ensure the existence of the optimal basic solutions, they are not enough to ensure that the optimal solution is unique and local Lipschitz continuous w.r.t. the inputs $(\boldsymbol{A}, \boldsymbol{b}, \boldsymbol{c})$. To ensure these points, we present two additional conditions based on (34) and (35):

$$\boldsymbol{y}_B > \boldsymbol{0} \quad \text{(Non-degeneracy)} \tag{36}$$
$$\boldsymbol{s}_N > \boldsymbol{0} \quad \text{(Strict complementary slackness)} \tag{37}$$

All the conditions together are enough to the uniquenss and local Lischitz continuity. Let's introduce a set consisting of all "good" LP instances:

$$\mathbb{X}_{\text{sub}} := \{(\boldsymbol{A}, \boldsymbol{b}, \boldsymbol{c}) \in \mathbb{X} : \text{The LP yields a tuple } (\boldsymbol{y}, \boldsymbol{z}, \boldsymbol{s}) \text{ satisfying (34), (35), (36) and (37).}\}$$

With all the preparations, we can prove Theorem 3.8 now. Actually, proving Theorem 3.8 in the context of standard-form LP is equivalent to proving the following two theorems.

**Theorem E.1.** *For any LP $(\boldsymbol{A}, \boldsymbol{b}, \boldsymbol{c}) \in \mathbb{X}_{\text{sub}}$, it must yield a unique optimal solution $\boldsymbol{y}_*$, and the solution mapping $(\boldsymbol{A}, \boldsymbol{b}, \boldsymbol{c}) \mapsto \boldsymbol{y}_*$ is locally Lipschitz continuous everywhere on $\mathbb{X}_{\text{sub}}$.*

**Theorem E.2.** $\mathbb{X}_{\text{sub}}$ *is a dense subset of* $\mathbb{X}$.

Theorem E.1 follows from Dontchev & Rockafellar (1996), which develops Robinson's notion of strong regularity (Robinson, 1980) for nonlinear programs. For completeness—and to keep notation consistent with linear programming—we restate the relevant lemma in an LP-adapted form and then verify its hypotheses for LP. We begin by quoting the result from Dontchev & Rockafellar (1996).

**Lemma E.3** (Dontchev & Rockafellar (1996)). *Consider a parameteric nonlinear program:*

$$\min_{\boldsymbol{y} \in \mathbb{R}^n} \boldsymbol{c}^\top \boldsymbol{y} + g_0(\boldsymbol{w}, \boldsymbol{y})$$

$$\text{s.t. } g_i(\boldsymbol{w}, \boldsymbol{y}) = u_i, \ \ 1 \leq i \leq r$$
$$g_i(\boldsymbol{w}, \boldsymbol{y}) \leq u_i, \ \ r + 1 \leq i \leq d$$

*where $g_i(0 \leq i \leq d)$ are all $\mathcal{C}^2$ functions, and $\boldsymbol{c}, \boldsymbol{w}$ and $\boldsymbol{u} = [u_1, \cdots, u_d]^\top$ are parameters to describe the program, and consider its Lagrangian with multipliers $\lambda = [\lambda_1, \cdots, \lambda_d] \in \mathbb{R}^d$ given by*

$$L(\boldsymbol{w}, \boldsymbol{y}, \lambda) = g_0(\boldsymbol{w}, \boldsymbol{y}) + \sum_{i=1}^{d} \lambda_i g_i(\boldsymbol{w}, \boldsymbol{y}).$$

*Let $(\bar{\boldsymbol{y}}, \bar{\lambda})$ be a KKT point at $(\bar{\boldsymbol{c}}, \bar{\boldsymbol{w}}, \bar{\boldsymbol{u}})$, and define the index sets at $(\bar{\boldsymbol{y}}, \bar{\lambda})$*

$$I_1 = \Big\{ r + 1 \leq i \leq d : g_i(\bar{\boldsymbol{w}}, \bar{\boldsymbol{y}}) = u_i, \bar{\lambda}_i > 0 \Big\} \cup \Big\{ 1, \cdots, r \Big\},$$
$$I_2 = \Big\{ r + 1 \leq i \leq d : g_i(\bar{\boldsymbol{w}}, \bar{\boldsymbol{y}}) = u_i, \bar{\lambda}_i = 0 \Big\},$$
$$I_3 = \Big\{ r + 1 \leq i \leq d : g_i(\bar{\boldsymbol{w}}, \bar{\boldsymbol{y}}) < u_i, \bar{\lambda}_i = 0 \Big\}.$$

*If the following conditions hold:*

- *The constraint gradients $\nabla_{\boldsymbol{y}} g_i(\bar{\boldsymbol{w}}, \bar{\boldsymbol{y}})$ for $i \in I_1 \cup I_2$ are linearly independent; and*

- *It holds that*
$$\langle \boldsymbol{y}', \nabla_{\boldsymbol{y}\boldsymbol{y}}^2 L(\bar{\boldsymbol{w}}, \bar{\boldsymbol{y}}, \bar{\lambda}) \boldsymbol{y}' \rangle > 0$$
*for all $\boldsymbol{y}' \neq \boldsymbol{0}$ in the subspace $\mathbb{M} = \Big\{ \boldsymbol{y}' : \boldsymbol{y}' \perp \nabla_{\boldsymbol{y}} g_i(\bar{\boldsymbol{w}}, \bar{\boldsymbol{y}})$ for all $i \in I_1 \Big\}$,*

*then the KKT solution map $(\boldsymbol{c}, \boldsymbol{w}, \boldsymbol{u}) \mapsto (\boldsymbol{y}, \lambda)$ is locally single-valued and Lipschitz around $(\bar{\boldsymbol{c}}, \bar{\boldsymbol{w}}, \bar{\boldsymbol{u}}, \bar{\boldsymbol{y}}, \bar{\lambda})$.*

*Proof of Theorem E.1.* Taking $r = m$ and $d = m + n$. Let $\boldsymbol{a}_i^\top$ be the i-th row of $\boldsymbol{A}$ in standard LP, and let

$$g_i(\boldsymbol{w}, \boldsymbol{y}) = \begin{cases} \boldsymbol{a}_i^\top \boldsymbol{y}, & i = 1, \ldots, m, \\ -y_{i-m}, & i = m+1, \ldots, m+n, \end{cases} \qquad u_i = \begin{cases} b_i, & i = 1, \ldots, m, \\ 0, & i = m+1, \ldots, m+n, \end{cases}$$

with $\boldsymbol{w}$ collecting the coefficients of $\boldsymbol{A}$. The Lagrangian in Lemma E.3 becomes

$$L(\boldsymbol{w}, \boldsymbol{y}, \lambda) = \boldsymbol{c}^\top \boldsymbol{y} + \sum_{i=1}^{m} \lambda_i \, \boldsymbol{a}_i^\top \boldsymbol{y} + \sum_{j=1}^{n} \lambda_{m+j}(-y_j).$$

Introduce the usual dual/primal–slack variables

$$\boldsymbol{z} := -\lambda_{1:m} \in \mathbb{R}^m, \qquad \boldsymbol{s} := \lambda_{m+1:m+n} \in \mathbb{R}_{\geq 0}^n,$$

to rewrite stationarity as $\nabla_{\boldsymbol{y}} L = \boldsymbol{c} - \boldsymbol{A}^\top \boldsymbol{z} - \boldsymbol{s} = \boldsymbol{0}$, i.e., $\boldsymbol{s} = \boldsymbol{c} - \boldsymbol{A}^\top \boldsymbol{z}$. Primal feasibility is $\boldsymbol{A}\boldsymbol{y} = \boldsymbol{b}$, $\boldsymbol{y} \geq 0$; dual feasibility is $\boldsymbol{s} \geq \boldsymbol{0}$; and complementarity is $\boldsymbol{y} \odot \boldsymbol{s} = \boldsymbol{0}$. Thus the KKT system in Lemma E.3 coincides with the standard LP KKT conditions.

Assume $(\boldsymbol{A}, \boldsymbol{b}, \boldsymbol{c}) \in \mathbb{X}_{\text{sub}}$, i.e., the LP admits a tuple $(\bar{\boldsymbol{y}}, \bar{\boldsymbol{z}}, \bar{\boldsymbol{s}})$ satisfying (34), (35), (36) and (37) (A nondegenerate and strict complementary basic point). In this context, the index sets $I_1, I_2, I_3$ at $(\bar{\boldsymbol{y}}, \bar{\boldsymbol{z}}, \bar{\boldsymbol{s}})$ become:

$$I_1 = \{1, \ldots, m\} \cup \{m + j : \bar{y}_j = 0, \bar{s}_j > 0\},$$
$$I_2 = \{m + j : \bar{y}_j = 0, \bar{s}_j = 0\},$$
$$I_3 = \{m + j : \bar{y}_j > 0, \bar{s}_j = 0\}.$$

which implies:

- For each $j$, either $\bar{y}_j > 0$ or $\bar{s}_j > 0$, which implies $I_2 = \varnothing$.

- $I_3$ is substantially the basis set: $I_3 = \{m + j : j \in B\}$

- $I_1$ includes all the indices in the complement of basis: $I_1 = \{1, \ldots, m\} \cup \{m+j : j \in N\}$

To verify the hypotheses of Lemma E.3, we examine the gradients:

$$\{\nabla_{\boldsymbol{y}} g_i\}_{i \in I_1} = \{\boldsymbol{a}_i\}_{i=1}^m \cup \{-\boldsymbol{e}_j\}_{j \in N}$$

In the context of standard LP, $|N| = n - m$. Hence, $\{\nabla_{\boldsymbol{y}} g_i\}_{i \in I_1}$ consists of $n$ vectors in $\mathbb{R}^n$. Now we create a matrix $\boldsymbol{G}$ by stacking these vectors as rows:

$$\boldsymbol{G} := \begin{bmatrix} \boldsymbol{a}_1^\top \\ \cdots \\ \boldsymbol{a}_m^\top \\ \boldsymbol{e}_{j_1}^\top \\ \cdots \\ \boldsymbol{e}_{j_{n-m}}^\top \end{bmatrix}$$

By properly permuting the columns of $\boldsymbol{G}$, it becomes

$$\tilde{\boldsymbol{G}} = \begin{bmatrix} \boldsymbol{B} & \boldsymbol{N} \\ \boldsymbol{0} & \boldsymbol{I} \end{bmatrix}$$

where $\boldsymbol{I}$ represents the identity matrix in $\mathbb{R}^{n-m}$. Since $\boldsymbol{B}$ (the basis matrix) and $\boldsymbol{I}$ are both nonsingular, $\tilde{\boldsymbol{G}}$ (and hence $\boldsymbol{G}$) must be nonsingular. Therefore, the rows of $\boldsymbol{G}$ are linearly independent, i.e., $\{\nabla_{\boldsymbol{y}} g_i\}_{i \in I_1}$ is linearly independent. With $I_2 = \varnothing$, the first hypothesis of Lemma E.3 holds. Moreover, because these gradients $\{\nabla_{\boldsymbol{y}} g_i\}_{i \in I_1}$ span $\mathbb{R}^n$, the $\mathbb{M}$ subspace must be trivial: $\mathbb{M} = \{\boldsymbol{0}\}$. Therefore, the second hypothesis of Lemma E.3 is automatically satisfied.

By Lemma E.3, the KKT solution map is locally single-valued and Lipschitz around the given point, which yields the desired local uniqueness and Lipschitz dependence of $\boldsymbol{y}_*$ on $(\boldsymbol{A}, \boldsymbol{b}, \boldsymbol{c})$ for every $(\boldsymbol{A}, \boldsymbol{b}, \boldsymbol{c}) \in \mathbb{X}_{\text{sub}}$. □

Theorem E.2 can be proved by fundamental concepts in real analysis.

*Proof of Theorem E.2.* To prove $\mathbb{X}_{\text{sub}}$ is dense in $\mathbb{X}$, it's enough to show that: For any $(\boldsymbol{A}, \boldsymbol{b}, \boldsymbol{c}) \in \mathbb{X}$, one can always create a sequence of LP $\{(\boldsymbol{A}_k, \boldsymbol{b}_k, \boldsymbol{c}_k)\}_{k \geq 1} \subset \mathbb{X}_{\text{sub}}$ such that

$$\boldsymbol{A}_k \to \boldsymbol{A}, \quad \boldsymbol{b}_k \to \boldsymbol{b}, \quad \boldsymbol{c}_k \to \boldsymbol{c}.$$

Now let's fix $(\boldsymbol{A}, \boldsymbol{b}, \boldsymbol{c}) \in \mathbb{X}$. As we previously discussed, there must be a tuple $(\boldsymbol{y}, \boldsymbol{z}, \boldsymbol{s})$ satisfying (34) and (35). Define:

$$\boldsymbol{y}_k := \boldsymbol{y} + \frac{1}{k} \boldsymbol{e}_B, \quad \boldsymbol{s}_k := \boldsymbol{s} + \frac{1}{k} \boldsymbol{e}_N, \quad \boldsymbol{z}_k := \boldsymbol{z}$$

so that $(\boldsymbol{y}_k, \boldsymbol{z}_k, \boldsymbol{s}_k)$ must satisfy the nondegeneracy and strict complementary slackness: (35), (36), and (37). Accordingly, define

$$\boldsymbol{A}_k := \boldsymbol{A}, \quad \boldsymbol{b}_k := \boldsymbol{A}_k \boldsymbol{y}_k, \quad \boldsymbol{c}_k := \boldsymbol{A}_k^\top \boldsymbol{z}_k + \boldsymbol{s}_k$$

Then one can verify that the tuple $(\boldsymbol{y}, \boldsymbol{z}, \boldsymbol{s})$ satisfies (34), (35), (36) and (37) for the LP instance $(\boldsymbol{A}_k, \boldsymbol{b}_k, \boldsymbol{c}_k)$, hence $(\boldsymbol{A}_k, \boldsymbol{b}_k, \boldsymbol{c}_k) \in \mathbb{X}_{\text{sub}}$ for all $k \geq 1$. Finally, such a perturbed LP instance can be arbitrarily close to $(\boldsymbol{A}, \boldsymbol{b}, \boldsymbol{c})$ as $k \to \infty$:

$$\|\boldsymbol{A}_k - \boldsymbol{A}\| = 0$$
$$\|\boldsymbol{b}_k - \boldsymbol{b}\| = \left\| \boldsymbol{A} \left( \frac{1}{k} \boldsymbol{e}_B \right) \right\| \leq \frac{1}{k} \|\boldsymbol{A}\| \|\boldsymbol{e}_B\| = \frac{\sqrt{m}}{k} \|\boldsymbol{A}\| \to 0$$
$$\|\boldsymbol{c}_k - \boldsymbol{c}\| = \left\| \frac{1}{k} \boldsymbol{e}_N \right\| = \frac{\sqrt{n-m}}{k} \to 0$$

which finishes the proof. □

## F    TRAINING STRATEGIES

**Unrolling vs implicit differentiation.** There are two training strategies adopted in this paper. One is named "unrolling" (minimizing $\ell(\boldsymbol{y}_T)$):

$$\min_\theta \ell(\boldsymbol{y}_T), \quad \boldsymbol{y}_{t+1} = \mathcal{G}_\theta(\boldsymbol{y}_t, \boldsymbol{x}), \ \ t = 0, 1, 2, \cdots, T-1$$

and the other is named "implicit differentiation" (minimizing $\ell(\boldsymbol{y}_*)$):

$$\min_\theta \ell(\boldsymbol{y}_*), \quad \boldsymbol{y}_* = \mathcal{G}_\theta(\boldsymbol{y}_*, \boldsymbol{x}).$$

These two strategies are closely related. In particular,

- As established in prior literature, unrolled training is mathematically equivalent to a Neumann series approximation of the implicit gradient (Geng et al., 2021). Specifically, implicit differentiation requires inverting the Jacobian $(\boldsymbol{I} - \boldsymbol{J}_{\mathcal{G}_\theta})^{-1}$; finite unrolling effectively approximates this inverse via a Neumann series expansion. This is a widely adopted technique in the implicit model community to avoid the instability and cost of exact inversion.

- Implicit training is simply the limit of unrolled training: as $T \to \infty$, the gradient $\nabla_\theta \ell(\boldsymbol{y}_T)$ converges to the implicit gradient $\nabla_\theta \ell(\boldsymbol{y}_*)$ (Geng et al., 2021).

Overall, unrolling and root-finding are merely two numerical implementations for approximating the same fixed point, $\boldsymbol{y}_*(\boldsymbol{x})$, and technically speaking, there is no significant gap or distinction between the two. Theoretically, infinite unrolling converges exactly to $\boldsymbol{y}_*(\boldsymbol{x})$. In practice, unrolling depth simply controls the trade-off between accuracy and computational cost: a dynamic strictly analogous to setting the error tolerance in implicit root-finding solvers.

Particularly in our paper, for Case Studies 1 & 2, we employ implicit differentiation (minimizing $\ell(\boldsymbol{y}_*)$) via root-finding; for Case Study 3, we adopt unrolling to train implicit GNNs, which serves as a truncated Neumann approximation of the implicit GNN gradient; for Case Study 4, we directly use the pretrained model from Geiping et al. (2025).

**Guarantees of Regularity and the Expressivity Trade-off.** While our experiments demonstrate that standard training (either unrolling and implicit differentiation defined above) empirically results in regular implicit operators, *we do not explicitly enforce this property in the loss function.* Designing training mechanisms that theoretically guarantee regularity without sacrificing the model's unique expressive capabilities remains an open and interesting future topic.

Recall that regularity (Definition 2.3) comprises two conditions: the Lipschitz continuity of the map $\boldsymbol{x} \mapsto \mathcal{G}_\theta(\boldsymbol{y}, \boldsymbol{x})$ and the contractivity of the map $\boldsymbol{y} \mapsto \mathcal{G}_\theta(\boldsymbol{y}, \boldsymbol{x})$. The first condition is largely inherent to standard deep learning architectures; compositions of affine layers with bounded weights and 1-Lipschitz activations (e.g., ReLU) naturally preserve Lipschitz continuity with respect to the input Miyato et al. (2018); Virmaux & Scaman (2018). Therefore, the critical challenge lies in guaranteeing the second condition: contractivity with respect to the state $\boldsymbol{y}$.

A substantial body of literature has sought to enforce this contractivity by construction (e.g., El Ghaoui et al. (2021); Winston & Kolter (2020); Jafarpour et al. (2021); Revay et al. (2020); Havens et al. (2023)). These approaches typically impose rigid structural constraints, such as parameterizing the model as a one-layer nonlinear MLP: $\mathcal{G}_\theta(\boldsymbol{y}, \boldsymbol{x}) = \sigma(\boldsymbol{A}\boldsymbol{y} + \boldsymbol{B}\boldsymbol{x} + \boldsymbol{b})$ and strictly bounding the spectral norm of $\boldsymbol{A}$, or enforcing global monotonicity.

However, these methods generally enforce a *uniform* contraction modulus $\mu$ across the entire domain $\boldsymbol{x} \in \mathbb{X}$. Our theoretical analysis suggests that such uniformity fundamentally undercuts the unique expressive advantage of implicit models. As illustrated in Figure 1, for a sequence of continuous iterates $\boldsymbol{y}_t(\boldsymbol{x})$ to converge to a target $\mathcal{F}(\boldsymbol{x})$ that is discontinuous or has singularities, the convergence *cannot* be uniform. This implies that the convergence rate—and consequently the operator's contraction modulus $\mu(\boldsymbol{x})$—must be adaptive, varying with $\boldsymbol{x}$ to allow for slower convergence in complex regions. Enforcing a globally uniform $\mu$ severs this adaptive capability, thereby severely constraining the model's expressive power.

Therefore, developing novel regularization techniques that can guarantee *adaptive* contractivity (ensuring $0 < \mu(\boldsymbol{x}) < 1$ locally while allowing it to vary over $\boldsymbol{x}$) is a critical direction for future research to balance theoretical stability with maximal expressivity.

## G    EXPERIMENT DETAILS REGARDING IMAGE RECONSTRUCTION

This section complements the main text with additional implementation and dataset details for the inverse-problem experiments.

**Experiment settings.** We consider an image deblurring task, $\boldsymbol{x} = \boldsymbol{A}(\boldsymbol{y}_*) + \boldsymbol{n}$, where $\boldsymbol{A}$ is the blur operator and $\boldsymbol{n}$ is the Gaussian noise ($\sigma = 0.03$). We use a motion-blur operator, and the blur kernel is the first of the eight kernels from Levin et al. (2009). Ground-truth images $\boldsymbol{y}_*$ come from BSDS500 (Martin et al., 2001). We follow the official splits (200 train / 100 validation / 200 test) and apply a random $128 \times 128$ crop to each image. For each $\boldsymbol{y}_*$, we generate the corresponding $\boldsymbol{x}$ by applying $\boldsymbol{A}$ and adding noise. The resulting pairs $(\boldsymbol{x}, \boldsymbol{y}_*)$ form three datasets $\mathbb{D}_{\text{inv,train}}$, $\mathbb{D}_{\text{inv,val}}$, and $\mathbb{D}_{\text{inv,test}}$ for training, validation, and testing, respectively. In both PGD and HQS style parameterizations ((5) and (6)), the operator $\mathcal{H}$ is implemented with DRUNet (Zhang et al., 2021).

**Training.** We initialize $\mathcal{H}$ using pretrained weights from the Deepinv library (Tachella et al., 2025) and then fine-tune the full implicit models on the BSDS500 training set for this deblurring task. Training follows the vanilla Jacobian-based implicit differentiation and is implemented on top of the official Deepinv framework. All models were trained with Adam (learning rate $10^{-4}$, batch size 3). Explicit baselines were trained for 20 epochs, and the implicit models for 10 epochs. After each epoch we evaluated on the validation set and saved the checkpoint; the final model used for testing is the one with the lowest validation loss. These epoch budgets were sufficient for validation-loss convergence.

**PSNR.** PSNR (Peak Signal-to-Noise Ratio) is defined between a reference $\boldsymbol{y}^*$ and reconstructed image $\boldsymbol{y}$ as

$$\text{PSNR}(\boldsymbol{y}, \boldsymbol{y}^*) := 10 \log_{10} \left( \frac{n \cdot \text{MAX}^2}{\|\boldsymbol{y} - \boldsymbol{y}^*\|^2} \right)$$

where $n$ is the dimension of $\boldsymbol{y}$ and $\boldsymbol{y}^*$, and MAX means the max possible pixel value (e.g., 255 for 8-bit, or 1 if images are in $[0, 1]$. In our context, it is 1. Higher PSNR means better (more accurate) reconstruction.

**Standard test set.** Evaluation uses the 200 images from the official BSDS500 test split, randomly cropped to $128 \times 128$. Let $\mathbb{D}_{\text{inv,test}} = \{(\boldsymbol{x}_i, \boldsymbol{y}_i^*)\}_{i=1}^{200}$, where

$$\boldsymbol{x}_i = \boldsymbol{A}(\boldsymbol{y}_i^*) + \boldsymbol{n}_i, \qquad \boldsymbol{n}_i \sim \mathcal{N}(\boldsymbol{0}, \sigma^2 \boldsymbol{I}), \;\; \sigma = 0.03.$$

Here $\boldsymbol{y}_i^*$ denotes the clean (ground-truth) image and $\boldsymbol{x}_i$ its corresponding blurred–noisy observation under the forward model $\boldsymbol{A}$.

**Perturbed test set.** To empirically validate our theory, we created a perturbed version of the test set. To create a diverse and representative set of perturbations, we generate perturbations that correspond to different frequency levels. Image frequencies represent different levels of detail, where low frequencies capture smooth, large-scale areas, and high frequencies capture sharp edges and fine textures. By probing the model with perturbations across this spectrum, we can comprehensively evaluate its behavior.

Specifically, we construct each perturbation by targeting a singular vector of the forward operator $\boldsymbol{A}$. Because $\boldsymbol{A}$ is (circular) convolution, its singular vectors are Fourier modes. For each image $\boldsymbol{y}_i^*$ and each frequency magnitude $f \in \{0.1, 0.3, 0.5, 0.7, 0.9\}$, we first identify the 2D discrete Fourier frequencies and sort them by their geometric distance from the origin. We then select the frequency coordinate $(u, v)$ at the $f$-th percentile of this sorted list. A one-hot tensor is created in the Fourier domain with a value of 1.0 at the chosen $(u, v)$ position and zeros elsewhere. This sparse frequency representation is transformed back into the image domain by applying the adjoint of the blur operator, $\boldsymbol{A}^\top$. These perturbations are visualized in Figure 8. Adding them to $\boldsymbol{y}_i^*$ respectively yields perturbed clean images $\boldsymbol{y}_{i,j}^*$ ($j = 1, \ldots, 5$); we then form the corresponding observation

$$\boldsymbol{x}_{i,j} = \boldsymbol{A}(\boldsymbol{y}_{i,j}^*) + \boldsymbol{n}_i, \qquad \boldsymbol{n}_i \sim \mathcal{N}(\boldsymbol{0}, \sigma^2 \boldsymbol{I}).$$

The perturbed evaluation set is

$$\mathbb{D}'_{\text{inv,test}} = \left\{ (\boldsymbol{x}_{i,j}, \boldsymbol{y}_{i,j}^*) : 1 \le i \le 200, \; 1 \le j \le 5 \right\}.$$

For convenience we also define the unperturbed index $j = 0$ by $\boldsymbol{x}_{i,0} := \boldsymbol{x}_i$ and $\boldsymbol{y}_{i,0}^* := \boldsymbol{y}_i^*$.

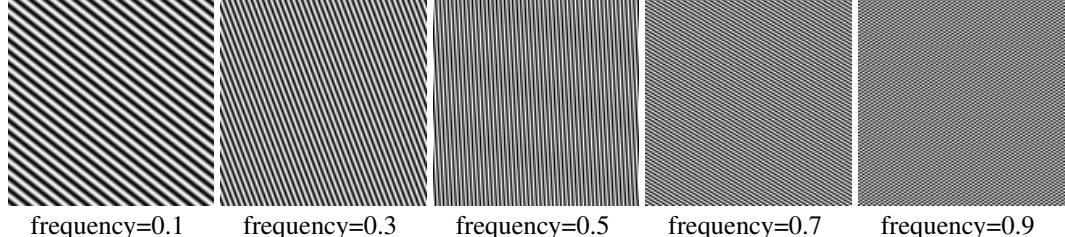

| frequency=0.1 | frequency=0.3 | frequency=0.5 | frequency=0.7 | frequency=0.9 |

Figure 8: Visualized perturbations for inverse problems.

Table 3: Deeper explicit models vs implicit models for image deblurring (PGD architecture). "Exp ($\times T$)" represents an explicit model $T$ times deeper than the implicit baseline. O/M denotes CUDA Out of Memory during training.

|          | Exp ($\times 1$) | Exp ($\times 2$) | Exp ($\times 4$) | Exp ($\times 8$) | Exp ($\times 16$) | Exp ($\times 32$) | Implicit |
|----------|---------|---------|----------|----------|----------|---------|----------|
| Params.  | 32.641 M | 65.282 M | 130.56 M | 261.13 M | 522.26 M | 1044.5 M | 32.641 M |
| PSNR     | 27.14 dB | 27.64 dB | 27.89 dB | 28.11 dB | 28.27 dB | O/M | 28.21 dB |

Table 4: Deeper explicit models vs implicit models for image deblurring (HQS architecture). "Exp ($\times T$)" represents an explicit model $T$ times deeper than the implicit baseline. O/M denotes CUDA Out of Memory during training.

|          | Exp ($\times 1$) | Exp ($\times 2$) | Exp ($\times 4$) | Exp ($\times 8$) | Exp ($\times 16$) | Exp ($\times 32$) | Implicit |
|----------|---------|---------|----------|----------|----------|---------|----------|
| Params.  | 32.641 M | 65.282 M | 130.56 M | 261.13 M | 522.26 M | 1044.5 M | 32.641 M |
| PSNR     | 26.94 dB | 28.02 dB | 28.35 dB | 28.69 dB | 28.87 dB | O/M | 29.18 dB |

**Platform.** All experiments were run on a workstation with eight Quadro RTX 6000 GPUs.

**Additional Experiments.** Implicit models often excel on imaging tasks, but a natural question is whether simply stacking more explicit layers (i.e., deepening the model) can close the gap. To probe this, we construct explicit counterparts to implicit models by untying the parameters across iterations:

$$\min_{\Theta} \mathbb{E}_{\boldsymbol{x}} \ell(\boldsymbol{y}_T, \boldsymbol{y}_*), \quad \text{s.t. } \boldsymbol{y}_t = \mathcal{G}_{\Theta^{(t)}}(\boldsymbol{y}_{t-1}, \boldsymbol{x}), \ t = 1, \cdots, T$$

where each block $\mathcal{G}_{\Theta^{(t)}}$ has the same architecture as in the implicit case (PGD or HQS), but $\Theta^{(t)}$ are separate for each $t$. This is equivalent to stacking $T$ blocks to form a deeper explicit model with more learnable parameters. Unlike implicit models (which can use different iteration counts at train vs. test), these explicit models must use the same $T$ for both training and testing. We evaluated $T \in \{1, 2, 4, 8, 16, 32\}$ to compare against the corresponding implicit models.

Tables 3 and 4 report results on image deblurring. Across both PGD and HQS settings, deepening explicit models increases parameter counts massively (up to $\sim 1$ billion) but yields diminishing returns in PSNR. Crucially, the implicit models achieve performance comparable to or better than explicit models that are $16\times$ deeper, while using a fraction of the parameters (32.6 M vs. 522 M). For instance, in the HQS setting, the implicit model (29.18 dB) outperforms the explicit model with 16 unrolled blocks (28.87 dB).

Furthermore, training extremely deep explicit models (e.g., $T = 32$) becomes infeasible due to memory constraints (O/M). This highlights the distinct efficiency advantage of the weight-tied implicit approach: it theoretically allows for infinite depth (realized here as 100 test-time iterations) while maintaining constant parameter counts (32.6 M) and memory usage.

## H    EXPERIMENT DETAILS REGARDING SCIENTIFIC COMPUTING

**Model structure and training.** Given cell-averaged forces $\boldsymbol{x} \in \mathbb{R}^{H \times W \times 2}$ and vorticities $\boldsymbol{y} \in \mathbb{R}^{H \times W \times 1}$, where $H$ means the height and $W$ means the width, we learn

$$\boldsymbol{z}_* = \mathcal{G}_{\Theta}\big(\boldsymbol{z}_*, \ \mathcal{Q}_{\Phi}(\boldsymbol{x})\big), \qquad \boldsymbol{y}_* = \mathcal{Q}_{\Psi}(\boldsymbol{z}_*),$$

where $\boldsymbol{z}_* \in \mathbb{R}^{H \times W \times C}$ is a latent field with $C$ channels. At inference, we iterate

$$\boldsymbol{z}_t = \mathcal{G}_\Theta\big(\boldsymbol{z}_{t-1},\ \mathcal{Q}_\Phi(\boldsymbol{x})\big),$$

for $1 \leq t \leq T$ and finally call $\boldsymbol{y}_T = \mathcal{Q}_\Phi(\boldsymbol{z}_T)$.

The projection $\mathcal{Q}_\Phi$ is a *pointwise* linear encoder applied at each grid cell to lift into $C$ channels. In particular, $\boldsymbol{g} = \mathcal{Q}_\Phi(\boldsymbol{x})$ reads

$$\boldsymbol{g} = \boldsymbol{W}_1 \boldsymbol{x} + \boldsymbol{b}_1 \in \mathbb{R}^{H \times W \times C}$$

where $\Phi = (\boldsymbol{W}_1, \boldsymbol{b}_1)$ are learnable parameters.

The core map $\mathcal{G}_\Theta(\boldsymbol{z}, \boldsymbol{g})$ stacks $L$ identical FNO layers with input injection:

$$\boldsymbol{z}^{(0)} = \boldsymbol{z}$$
$$\boldsymbol{z}^{(l)} = \sigma\left(\boldsymbol{g} + \boldsymbol{W}_2^{(l)} \boldsymbol{z}^{(l-1)} + \boldsymbol{b}_2^{(l)} + \mathrm{IFFT}(\boldsymbol{R}^{(l)} \cdot \mathrm{FFT}(\boldsymbol{z}^{(l-1)}))\right), \quad l = 1, 2, \cdots, L,$$
$$\mathcal{G}_\Theta(\boldsymbol{z}, \boldsymbol{g}) = \boldsymbol{z}^{(L)}$$

where $\Theta = \{\boldsymbol{W}_2^{(l)}, \boldsymbol{b}_2^{(l)}, \boldsymbol{R}^{(l)}\}_{l=1}^L$ are learnable parameters. Each layer: (i) performs a global spectral convolution on $\boldsymbol{z}$: take an FFT of the $C$-channel tensor, keep only a small set of low Fourier modes. Suppose the number of retained Fourier modes is $K \times K$ (2D FFT), $\mathrm{FFT}(\boldsymbol{z}) \in \mathbb{C}^{K \times K \times C}$. For each retained mode $(k_1, k_2)$ multiply the $C$-dimensional channel vector by a learnable dense matrix $\boldsymbol{R}_{k_1,k_2}^{(l)} \in \mathbb{C}^{C \times C}$ (mixing channels) and hence the overall matrix is of size $\boldsymbol{R}^{(l)} \in \mathbb{C}^{K \times K \times C \times C}$, then apply an inverse FFT; (ii) adds a local pointwise transform, adds the injected encoder features $\mathcal{Q}_\Phi(\boldsymbol{x})$, and applies a nonlinearity. This realizes a resolution-invariant, globally receptive operator that naturally respects periodic boundary conditions.

Finally, we decode with the pointwise readout $\mathcal{Q}_\Psi$ (a small per-cell two-layer MLP) to produce $\boldsymbol{y} \in \mathbb{R}^{H \times W \times 1}$ where $\Psi$ are learnable parameters.

All samples use $H = W = 128$. Unless stated otherwise, we set the latent width $C = 32$, retain $K = 12$ Fourier modes per dimension in the FNO blocks, and use $L = 3$ FNO layers inside $\mathcal{G}_\Theta$. Training differentiates implicitly through the fixed point, and the fixed-point solver uses Anderson acceleration. We optimize with Adam (learning rate $5 \times 10^{-3}$, batch size 16). For explicit baselines, we train for 500 epochs, which suffices for the training loss to converge.

**Perturbed data generation.** In this paragraph, we describe how we generate perturbed samples in $\mathbb{D}'_{\mathrm{pde,test}}$. We take the dataset of Marwah et al. (2023) as the unperturbed set $\mathbb{D}_{\mathrm{pde,test}}$ and create perturbations by linearizing the steady NS equation (7). Each sample $(f, \omega)$ comprises a forcing term $f$ and its vorticity solution $\omega$. Directly prescribing $f$ and solving for $\omega$ is computationally costly; following Marwah et al. (2023), we instead prescribe $\omega$ and obtain the corresponding $f$ by evaluating the PDE operator (not by solving the PDE). In our setting, the base samples are given; thus we first construct a solution perturbation $\delta\omega$ and then compute the induced forcing perturbation $\delta f$ via the linearization, yielding the perturbed pair $(f + \delta f,\ \omega + \delta\omega)$.

Note that, while the dataset is discrete, we use the continuous notation $f, \omega, u$ in this section to ease reading and to remain consistent with the PDE literature. In addition, we use $\xi = (\xi_1, \xi_2)$ as the special domain variable to keep consistent with our main text, and use $k = (k_1, k_2)$ as the frequency domain variable.

(Generate $\delta\omega$). Fix a target wavenumber $k_* \in \mathbb{N}$ and a desired $L^2$–magnitude $\eta > 0$. We construct $\delta\omega$ by

$$\delta\omega(\xi_1, \xi_2) = A \sin\big(k_* \xi_1 + k_* \xi_2\big), \qquad A \text{ chosen so that } \|\delta\omega\|_{L^2(\Omega)} = \eta.$$

The wavenumber is selected from a user–specified frequency percentile $p_{\mathrm{freq}}$ relative to the maximum resolvable frequency $k_{\max} = H/2 = W/2$, namely

$$k_* = p_{\mathrm{freq}} \times k_{\max} \quad \text{(rounded to the nearest integer mode)}.$$

In our code we set the grid size $H = W = 128$, the perturbation strength $\eta = 0.01$, and choose

$$p_{\mathrm{freq}} \in \{0.01, 0.02, 0.03, 0.04, 0.05, 0.1, 0.3, 0.5, 0.7, 0.9, 0.95, 0.96, 0.97, 0.98, 0.99\}.$$

Accordingly, each original sample yields 15 perturbed samples.

Table 5: Implicit FNO vs deeper Explicit FNO. "Exp($\times l$)" denotes an explicit model that is $l$ times deeper than the implicit FNO. O/M indicates a CUDA out-of-memory error during training.

|  | Exp($\times 1$) | Exp($\times 2$) | Exp($\times 4$) | Exp($\times 8$) | Exp($\times 16$) | Exp($\times 32$) | Implicit |
|---|---|---|---|---|---|---|---|
| Params. | 2.376 M | 4.155 M | 7.713 M | 14.83M | 29.06M | 57.52M | 2.376 M |
| Rel. Err. | 0.1787 | 0.1526 | 0.1410 | 0.1380 | 0.1360 | O/M | 0.0785 |

Table 6: Implicit FNO vs wider Explicit FNO. "Exp($\times w$)" denotes an explicit model that is $w$ times wider than the implicit FNO. O/M indicates a CUDA out-of-memory error during training.

|  | Exp($\times 1$) | Exp($\times 2$) | Exp($\times 4$) | Exp($\times 8$) | Implicit |
|---|---|---|---|---|---|
| Params. | 2.376 M | 9.504 M | 38.01 M | 152.0M | 2.376 M |
| Rel. Err. | 0.1787 | 0.1555 | 0.1401 | O/M | 0.0785 |

(Generate velocity from vorticity). Given a scalar vorticity field $\omega$ (and its perturbation $\delta\omega$), we recover the corresponding velocities $u$ via a streamfunction $\psi$ given by

$$u = (\partial_2\psi, -\partial_1\psi), \qquad \omega = -\Delta\psi.$$

Hence $\psi$ is obtained by solving the Poisson equation $\Delta\psi = -\omega$, after which we obtain $u$. On a periodic grid, these operators are implemented efficiently in the Fourier domain.

(Linearization and the perturbed vorticity forcing $\delta g$). Applying the (scalar) curl "$\nabla\times$" to both sides of (7) yields the steady vorticity form

$$(u{\cdot}\nabla)\omega \,-\, \nu\Delta\omega = g, \qquad g = \nabla\times f = \partial_1 f_2 - \partial_2 f_1,$$

where $f = (f_1, f_2)$ is the body force and $g$ is its curl. Introducing perturbations $(\delta u, \delta\omega, \delta g)$ and expanding

$$(u + \delta u){\cdot}\nabla\,(\omega + \delta\omega) \,-\, \nu\Delta(\omega + \delta\omega) = g + \delta g,$$

then subtracting the base equation and discarding higher–order terms gives the first–order relation

$$\delta g = (u{\cdot}\nabla)\,\delta\omega \,+\, (\delta u{\cdot}\nabla)\,\omega \,-\, \nu\Delta\,\delta\omega.$$

Again, for numerical implementation on a periodic grid, the differential operators are applied efficiently in the Fourier domain.

(Recover the vector force $\delta f$ from its curl $\delta g$). We recover a periodic $\delta f = (\delta f_1, \delta f_2)$ satisfying $\nabla\times\delta f = \delta g$ by solving a Poisson equation for an auxiliary streamfunction $\psi$ and obtain $\delta f$ exactly as in "Generate velocity from vorticity."

**Additional Experiments: Scaling Explicit Models.** A natural question is whether the performance gap between implicit and explicit models can be bridged simply by scaling up the explicit architecture (i.e., stacking more layers or increasing channel width). To investigate this, we compared the implicit FNO against explicit baselines scaled significantly in two dimensions: depth (up to $32\times$) and width (up to $8\times$). The results, summarized in Table 5 and Table 6, demonstrate that while scaling explicit models yields modest accuracy gains, it faces severe diminishing returns and computational bottlenecks (eventually leading to CUDA Out-of-Memory errors). Crucially, the implicit model achieves markedly better performance (lowest relative error of $0.0785$) than even the largest viable explicit models, despite the explicit counterparts using over $10\times$ the number of parameters (e.g., $29.06$ M for Exp($\times 16$) vs. $2.376$ M for Implicit). This confirms that the implicit formulation provides an expressive advantage that cannot be efficiently replicated by simply allocating more capacity to an explicit solver.

Note: These findings are broadly consistent with Marwah et al. (2023). We follow their setup with two minor deviations: we use a smaller training batch size (16) due to hardware limits, and while we keep $T = 24$ training iterations for the implicit model, at inference we run $T = 50$, because we observe that the trained implicit models remain stable and often benefit from additional fixed-point iterations at test time.

# I    EXPERIMENT DETAILS REGARDING LP

**GNN model details.** We implement (9):

$$\boldsymbol{z}_* = \mathcal{G}_\Theta(\boldsymbol{z}_*, \mathcal{Q}_\Phi(\boldsymbol{x})), \quad \boldsymbol{y}_* = \mathcal{Q}_\Psi(\boldsymbol{z}_*)$$

with an $L$-layer message-passing GNN (Scarselli et al., 2008; Xu et al., 2019) on the bipartite graph. Let $\mathcal{N}(i)$ (resp. $\mathcal{N}(j)$) be the neighbors of constraint node $W_i$ (resp. variable node $V_j$). With shared MLPs across all nodes and edges, the GNN structure is given by:

Input-embedding:
$$W_i^{(0)} = \mathrm{MLP}_{\phi_1}(b_i, \circ_i),$$
$$V_j^{(0)} = \mathrm{MLP}_{\phi_2}(c_j, l_j, u_j, z_{\mathrm{in},j})$$

Message-passing $(1 \le l \le L-1):$
$$W_i^{(l)} = \mathrm{MLP}_{\theta_1^{(l)}}\left(W_i^{(l)}, \sum_{j \in \mathcal{N}(i)} A_{ij} \cdot \mathrm{MLP}_{\theta_2^{(l)}}\left(V_j^{(l-1)}\right)\right),$$

$$V_j^{(l)} = \mathrm{MLP}_{\theta_3^{(l)}}\left(V_j^{(l)}, \sum_{i \in \mathcal{N}(j)} A_{ij} \cdot \mathrm{MLP}_{\theta_4^{(l)}}\left(W_i^{(l-1)}\right)\right)$$

Output-embedding:
$$z_{\mathrm{out},j} = \mathrm{MLP}_{\theta_5}\left(V_j^{(L)}\right)$$

We write this compactly as follows.

$$\boldsymbol{z}_{\mathrm{out}} = \mathcal{G}_\Theta(\boldsymbol{z}_{\mathrm{in}}, \mathcal{Q}_\Phi(\boldsymbol{x}))$$

where $\Theta = \left\{\{\theta_1^{(l)}\}_{l=1}^{L-1}, \{\theta_2^{(l)}\}_{l=1}^{L-1}, \{\theta_3^{(l)}\}_{l=1}^{L-1}, \{\theta_4^{(l)}\}_{l=1}^{L-1}, \theta_5\right\}$ are trainable parameters in the GNN, $\Phi = \{\phi_1, \phi_2\}$ includes the trainable parameters of the input embedding. The input $\boldsymbol{x}$ includes all static information $\boldsymbol{x} := (\boldsymbol{A}, \boldsymbol{b}, \boldsymbol{c}, \circ, \boldsymbol{l}, \boldsymbol{u})$. Finally, the output embedding $\boldsymbol{y} = \mathcal{Q}_\Psi(\boldsymbol{z})$ is given by

$$y_j = \mathrm{MLP}_\Psi(z_j)$$

for every variable node $j$. All MLPs in $\mathcal{G}_\Theta$, $\mathcal{Q}_\Phi$, and $\mathcal{Q}_\Psi$ use two layers with ReLU activations. We sweep widths (or embedding sizes) in $\{4, 8, 16, 32\}$ and report results in the main text.

Note that $l$ is the layer index within the GNN structure, not the iteration number $t$. All parameters in $\Theta$ are independent of the iteration number, so this GNN can be applied iteratively. $\boldsymbol{x}$ is the static features and $\boldsymbol{z}$ is the dynamic feature. In addition, removing the dynamic input $z_{\mathrm{in}}$ and decoding directly to $\boldsymbol{y}$ recovers the standard (explicit) GNN baseline.

**Dataset generation.** We largely follow Chen et al. (2023) to construct the training set $\mathbb{D}_{\mathrm{LP,train}}$ and test set $\mathbb{D}_{\mathrm{LP,test}}$, drawing $(\boldsymbol{A}, \boldsymbol{b}, \boldsymbol{c}, \circ, \boldsymbol{l}, \boldsymbol{u})$ i.i.d. from the same distribution. Each LP has 50 variables and 10 constraints. The matrix $\boldsymbol{A}$ is sparse with 100 nonzeros whose locations are chosen uniformly at random and whose values are sampled from a standard normal distribution. Entries of $\boldsymbol{b}$ and $\boldsymbol{c}$ are sampled i.i.d. from $\mathrm{Unif}[-1, 1]$, after which $\boldsymbol{c}$ is scaled by 0.01. Variable bounds $\boldsymbol{l}, \boldsymbol{u}$ are sampled coordinatewise from $\mathcal{N}(0, 10)$; whenever $l_j > u_j$ we swap them. Constraint types are sampled independently with $\Pr(\circ_i = \text{ ``}\le\text{''}) = 0.7$ and $\Pr(\circ_i = \text{ ``}=\text{''}) = 0.3$. Under this generator, the feasibility probability is approximately 0.53; we retain only feasible instances, yielding 2,500 LPs for training and 1,000 for testing. Solutions are computed with `scipy.optimize`.

To build the perturbed datasets $\mathbb{D}_{\mathrm{LP,test}}^{(j)}$, we perturb one component at a time while holding the others fixed. For $\boldsymbol{c}$, draw $\delta\boldsymbol{c}$ with i.i.d. standard normal entries, normalize, and scale to magnitude $10^{-4}$:

$$\boldsymbol{c}' = \boldsymbol{c} + 10^{-4} \times \frac{\delta\boldsymbol{c}}{\|\delta\boldsymbol{c}\|}.$$

We apply the same procedure to $\boldsymbol{b}$, $\boldsymbol{l}$, and $\boldsymbol{u}$. For $\boldsymbol{A}$, we perturb only existing nonzeros to preserve the sparsity pattern: let $\mathbb{S} = \{(i_k, j_k)\}_{k=1}^{\mathrm{nnz}(\boldsymbol{A})}$ be the nonzero locations and draw $\delta\boldsymbol{a} \in \mathbb{R}^{|\mathbb{S}|}$ i.i.d. standard normal; normalize and scale so $\|\delta\boldsymbol{a}\| = 10^{-4}$, then set

$$\boldsymbol{A}'_{i_k, j_k} = \boldsymbol{A}_{i_k, j_k} + (\delta\boldsymbol{a})_k \text{ for } (i_k, j_k) \in \mathbb{S}, \qquad \boldsymbol{A}'_{i,j} = \boldsymbol{A}_{i,j} \text{ otherwise.}$$

This yields five perturbed versions (perturbing $\boldsymbol{A}$, $\boldsymbol{b}$, $\boldsymbol{c}$, $\boldsymbol{l}$, or $\boldsymbol{u}$ separately). We evaluate the estimated Lipschitz constants $L_t$ and relative errors $E_t$ on each version and report the results in the main text.

**Training method.** To train our implicit GNNs, we employ a two-stage curriculum strategy. The model is trained by unrolling its iterative updates for a fixed number of steps, T, and minimizing the loss on the final output:

$$\min_{\Theta,\Phi,\Psi} \sum_{(\boldsymbol{x},\boldsymbol{y}_*)\in\mathbb{D}_{\text{LP,train}}} \ell(\boldsymbol{y}_T, \boldsymbol{y}_*)$$
$$\text{s.t. } \boldsymbol{z}_0 = \boldsymbol{0}$$
$$\boldsymbol{z}_t = \mathcal{G}_\Theta(\boldsymbol{z}_{t-1}, \mathcal{Q}_\Phi(\boldsymbol{x})), \quad t = 1, 2, \cdots, T$$
$$\boldsymbol{y}_T = \mathcal{Q}_\Psi(\boldsymbol{z}_T)$$

We set the final unroll horizon to $T = 6$, as we observed no significant improvements with longer sequences. Training directly with $T = 6$ is inefficient, so we adopt a two-stage curriculum. This approach is a standard practice in the Learning to Optimize field for training implicit or unrolled models that solve optimization problems (Chen et al., 2022c). This approach begins with a shorter unroll horizon and a larger learning rate, using the trained model to warm-start the subsequent stage with a longer horizon and a reduced learning rate. This strategy is often described as "layerwise training" (Chen et al., 2018; Liu et al., 2019) or "curriculum learning" (Chen et al., 2020). In our settings: Stage 1 uses $T = 3$ with a learning rate $0.01$; Stage 2 uses $T = 6$ with a learning rate $10^{-4}$. Both stages use Adam optimizer.

For a fair comparison, the non-iterative explicit GNNs are trained using the same two-stage learning rate schedule. This regimen proved effective, as the training errors for our explicit baselines surpassed those reported in prior work (Chen et al., 2023).

At the inference time, $T$ can be chosen as the unroll length in the training stage, or moderately longer. In our experiments, we use $T = 8$ at the inference time, as we do not observe significant improvement with a larger number of iterations.

*Remark.* While we employ unrolled training rather than the vanilla Jacibian-based implicit differentiation, we classify our approach as an "implicit model" because the underlying architecture, a weight-tied update $\boldsymbol{z}_t = \mathcal{G}_\Theta(\boldsymbol{z}_{t-1}, \mathcal{Q}_\Phi(\boldsymbol{x}))$, remains identical. The distinction lies solely in the numerical implementation: as established by Geng et al. (2021), unrolled training is mathematically equivalent to a Neumann series approximation of the implicit gradient. Thus, unrolling and root-finding are simply two valid strategies for approximating the same fixed point, $\boldsymbol{y}_*(\boldsymbol{x})$. This equivalence is widely recognized in the Implicit GNN literature, where Neumann approximations are standard for scaling to large graphs (e.g., (Baker et al., 2023)). Since our focus is on expressivity rather than optimization mechanics, we treat both formulations as belonging to the same model class.

## J   ADDITIONAL RESULTS REGARDING LLM REASONING

In the main text (Section 3.4), we present a qualitative example showing how additional recurrent blocks let the model separate nearby prompts into different semantic contexts. Here we complement that illustration with a **quantitative measure** of this effect.

To quantitatively measure this, we define an "Empirical Lipschitz" constant $L_t$ using Levenshtein distance $d(\cdot, \cdot)$:

$$L_t(i) := \frac{d(\boldsymbol{y}_t(\boldsymbol{x}_i), \boldsymbol{y}_t(\boldsymbol{x}_i'))}{d(\boldsymbol{x}_i, \boldsymbol{x}_i')}$$

We construct $\{(\boldsymbol{x}_i, \boldsymbol{x}_i')\}_{i=1}^{200}$, a dataset of 200 pairs where inputs differ by only 1-2 words but require vastly different semantic contexts. Figure 9 plots the geometric mean of $L_t$, which rises from $\approx 29.2$ at $t = 2$ (indicating relative insensitivity) to saturate at $\approx 52.5$ by $t = 16$.

Consistent with our theory, this growth reflects the model's emergent capacity to map proximal inputs to semantically distinct outputs. Even in the discrete domain of language reasoning, iterating a fixed operator allows the model to scale its expressive power, evolving from simple surface-level processing to complex, context-aware reasoning.

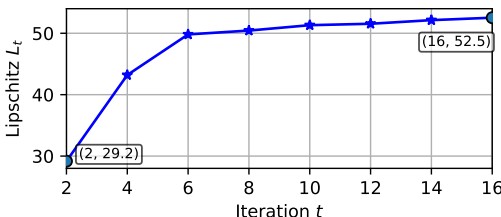

Figure 9: Empirical Lipschitz of the output sequence $\boldsymbol{y}_t(\cdot)$ generated by Geiping et al. (2025) using $t$ recurrent blocks. $L_t$ grows as $t$ increases.

## K  BROADER CONTEXTUAL DISCUSSIONS

While our work primarily establishes the expressive power of implicit models through the lens of fixed-point iterations, we situate our contributions within the broader landscape of implicit model theory in this section.

**Universality and expressivity.** Despite its foundational importance, a general and systematic theory of expressive power for implicit models remains largely open. To our knowledge, existing results address only specific facets of the problem. For example, while Bai et al. (2019) demonstrated that an operator $g$ exists such that its fixed point reproduces any explicit network $f$, this existence result crucially does not guarantee that the fixed-point iteration (1) actually converges. Other works have established universality within restricted domains, such as steady-state PDEs Marwah et al. (2023), or proven separation results where implicit models outperform explicit counterparts in specific settings Wu et al. (2024). While insightful, none of these studies provide a complete characterization of the general function class representable by implicit models, nor do they directly address the fundamental questions (Q1) and (Q2) raised in our introduction.

**Contrast with explicit models.** First, consider the setting where the model is subject to a global Lipschitz constraint (e.g., $\mathrm{Lip}(f_\theta) \leq 1$), which is common for robustness and stability. In this case, explicit feedforward networks are mathematically strictly limited to representing globally 1-Lipschitz maps (Murari et al., 2025). Consequently, they are fundamentally incapable of expressing locally Lipschitz targets whose gradients become arbitrarily large (such as $1/x$ near zero). In contrast, our work demonstrates that implicit models break this barrier: a simple, globally regular operator (Lipschitz in $\boldsymbol{x}$, contractive in $\boldsymbol{y}$) can generate complex, locally Lipschitz fixed-point maps via iteration. This "Simple Operator $\to$ Complex Fixed Point" mechanism is the core difference claimed in our paper.

Second, if we remove constraints, Beneventano et al. (2021) have shown that deep ReLU networks can indeed approximate locally Lipschitz functions on arbitrary compact sets. However, achieving high precision for such complex targets requires the explicit model size (depth/width) to grow arbitrarily large. Here lies the crucial distinction: explicit models scale expressivity with model size, whereas implicit models are able to scale expressivity with test-time iterations. This allows implicit models to represent increasingly complex functions dynamically without adding parameters.

**Training Dynamics and Convergence.** A significant body of work focuses on the optimization mechanics of implicit models. Geng et al. (2021) rigorously established the equivalence between unrolled training and implicit differentiation via Neumann series approximations, validating the training methodologies used in our case studies. Ling et al. (2023); Truong (2025) provide global convergence guarantees and rate analyses for the training in over-parameterized deep equilibrium models. While these studies ensure that training algorithms can successfully minimize the loss, our work addresses the fundamental antecedent question: whether a model exists that is capable of representing the target function in the first place.

**Generalization.** Distinct from expressivity, Fung & Berkels (2024) derive generalization bounds for families of implicit networks, characterizing their ability to perform on unseen data. Our analysis focuses on approximation capacity—the ability to construct an operator that exactly reproduces a target map—which is orthogonal to the sample complexity and generalization bounds discussed in their work.

**Infinite-Width Limits and Kernel Connections.** Recent research has sought to bridge the gap between implicit models, explicit deep networks, and kernel methods. Gao et al. (2022) extend the over-parameterization theory of explicit networks to implicit models, establishing well-posedness and convergence even in finite-width regimes where standard infinite-depth results do not directly apply. In the infinite-width limit, Feng & Kolter (2023) formally derive the Neural Tangent Kernel (NTK) for equilibrium models, characterizing their training dynamics in the linear regime. On the architectural side, Ling et al. (2024) show that for high-dimensional Gaussian mixtures, deep equilibrium models can be functionally equivalent to shallow explicit networks. In contrast to these kernel-based or distribution-specific analyses, *our work adopts a non-parametric function-space perspective*; we demonstrate that for general locally Lipschitz targets, the expressive power of implicit models is not static but scales dynamically with test-time computation, a property distinct from the linear regimes often studied in kernel theory.

## L   LLM USAGE STATEMENT

We used LLMs solely as a writing-polish assistant across all sections in the main text and appendix. Its role was limited to grammar fixes, wording/flow improvements, and rephrasing of text that we originally drafted. All model suggestions were reviewed, verified, and, when necessary, edited by the authors to ensure accuracy. The authors take full responsibility for the final manuscript, including any text influenced by LLM assistance.

