# OpenReview forum: "Expressive Power of Implicit Models: Rich Equilibria and Test-Time Scaling"
_ICLR.cc/2026/Conference — ICLR 2026 Poster_

### Official Review · Reviewer_KADN · 2025-10-26

**Soundness:** 3
**Presentation:** 3
**Contribution:** 3
**Rating:** 8
**Confidence:** 2

**Summary:**

This work is concerned with trying to explain the viability of equilibrium models. It is shown using Banach's fixed point theorem that using a regular map can approximate a function class that is only locally Lipschitz, where regularity is defined by contractiveness and global Lipschitz property. The more the map is applied the closer the approximation, thus scaling with test-time compute. This is illustrated with a toy example and further validated by case studies in different domains as image reconstruction, scientific computing, and operational research. The result sheds light on how implicit models can outperform explicit models by finding that implicit models only need to approximate a more regular function, whereas an explicit network would need to be able to approximate the actual function.

**Strengths:**

The study of the expressivity of implicit models is well motivated. The work presents a clear advantage of equilibrium models in this context. The exposition of work is good with an illustrative toy example and various case studies, especially Fig1 helps to substantiate the theoretical claims made. The argument that we need to be able to approximate locally Lipschitz functions provides also a new generalization perspective.

**Weaknesses:**

1) The theory is an application of a well known result: Banach's fixed point theorem. The theory could be strengthened by providing a rate of minimal/maximal possible growth for the Lipschitz constant. For example Figure 4a seems to suggest non-monotone behavior of the Lipschitz constant growth. Providing a bound could help with heuristics of how many iterations we have to apply in practice.

2) Even tough the work validates their finding in multiple applications they are all from areas in applied mathematics. The work could be strengthened by showing results on a language task or a more open ended task.

3) The illustrations of the implicit and non-implicit FNO in Figure 5 seem to be not very different can this be highlighted better? The only visual difference is the black region in the bottom right corner. In addition, the channel figures might not convey much information. Perhaps removing them and then highlighting the difference more could benefit the exposition.

**Questions:**

See weaknesses as well.

Is the comparison in Figure 5 fair, as the two models have the same amount of parameters, and the implicit model is allowed to reiterate effectively using more compute?

Is the finding of the ability of training implicit GCNs with more overparameterization a new finding or is it more a confirmation of previous work?

---

> ### Author Response · Authors · 2025-11-23
> **Official Rebuttal: Part I**
>
> Thank you very much for your encouraging comments! Please find our responses below:
>
> > [W1] The theory is an application of a well known result: Banach's fixed point theorem. The theory could be strengthened by providing a rate of minimal/maximal possible growth for the Lipschitz constant. For example Figure 4a seems to suggest non-monotone behavior of the Lipschitz constant growth. Providing a bound could help with heuristics of how many iterations we have to apply in practice.
>
> First We respectfully clarify that our theory is NOT merely an application of Banach's theorem. Banach's theorem addresses the problem: given a contractive operator $\mathcal{G}(\cdot)$, does it yield a unique fixed point $y_*$, and does the iteration $y_{t+1}=\mathcal{G}(y_t)$ converge? However, our theorems are totally different:
> * Theorem 2.4 addresses the existence of the operator itself. It asks: Given a target map $\mathcal{F}(x)$, does there exist a regular implicit operator $\mathcal{G}(\cdot, x)$ such that its fixed point exactly reproduces $\mathcal{F}(x)$? These are fundamentally different mathematical questions: **one concerns existence of fixed point $y_\ast$, the other concerns existence of operators.**
> * Theorem 2.5 characterizes the properties of the resulting fixed-point map $x \mapsto y_*(x)$ as a function of the input. **While Banach’s theorem concerns the existence of $y_\ast(x)$ for each $x$, our theorem establishes local Lipschitz continuity of the fixed-point map $x \mapsto y_*(x)$** across the input domain $\mathbb{X}$. They are foudamentally different.
> * **Novelty of Proofs:** We highlight that the core constructions in our proofs, specifically the design of the operator $\mathcal{G}$ in Eq. (10) and the construction of the function $\epsilon(x)$ in Theorem A.1, are novel technical elements that are orthogonal to standard fixed-point theory.
>
> Regarding the growth rate of the Lipschitz constant, we agree that establishing strict lower and upper bounds is an interesting and important topic. However, deriving tight bounds for this purpose requires a granular analysis **tied to specific model architectures.** For instance, the bound dynamics for an implicit FNO (Case Study 2) would differ significantly from those of an implicit GNN (Case Study 3), and our numerical observations (Figure 5 vs Figure 7) support this point. A comprehensive treatment of these architecture-specific rates would constitute a substantial study in its own right. Given the novelty of Theorems 2.4 & 2.5, we maintain these as the primary theoretical contributions of this work and view the characterization of specific growth rates as a valuable direction for future research.
>
> To address the reviwer's concern, we've included this future direction in the Conclusion section of our revised draft.
>
>
>
> > [W2] Even tough the work validates their finding in multiple applications they are all from areas in applied mathematics. The work could be strengthened by showing results on a language task or a more open ended task.
>
> **(Additional experiments on language tasks).** We added a new case study on LLM reasoning in **Section 3.4.** In particular, we extended our validation to the domain of language modeling using the pre-trained looped transformer from [6]. We found that our theory holds even in this discrete domain: as test-time iterations increase, the "empirical Lipschitz" complexity of the model's output rises (saturating around $t=16$), allowing the model to progressively resolve semantic ambiguities (e.g., distinguishing the context of "charge" in physics vs. finance) that earlier iterations fail to capture. This demonstrates that the principle of scaling expressivity with compute extends beyond continuous mathematical problems.
>
> > [W3] The illustrations of the implicit and non-implicit FNO in Figure 5 seem to be not very different can this be highlighted better? The only visual difference is the black region in the bottom right corner. In addition, the channel figures might not convey much information. Perhaps removing them and then highlighting the difference more could benefit the exposition.
>
> We agree and have replaced the two input images with their corresponding difference maps (prediction minus ground truth) to better highlight the disparity between the two models. This change is reflected in Figure 5 of the revised paper.

---

> ### Author Response · Authors · 2025-11-23
> **Official Rebuttal: Part II**
>
> > Is the comparison in Figure 5 fair, as the two models have the same amount of parameters, and the implicit model is allowed to reiterate effectively using more compute?
>
> We believe Figure 5 precisely validates our core theoretical claim: while both models possess the same static parameter capacity, the implicit model's effective expressivity scales with test-time iterations. Consequently, it naturally surpasses the vanilla explicit model once allocated sufficient inference steps.
>
> To further address the reviewer's concern and comprehensively compare implicit models with explicit models, we conducted additional experiments. Results are highlighted as blue texts in **Appendices G and H: Tables 3 & 4 (on Page 42) for inverse problems and Tables 5 & 6 (on Page 44) for PDEs.**
>
> **(Additional experiments on scaling explicit models).** We aggressively scaled the explicit baseline models in depth and width until they hit hardware memory limits (CUDA Out-of-Memory during training). The results were decisive: simply adding capacity to explicit models yields diminishing returns. For example, in image deblurring, the implicit model (32.6M parameters) outperformed an explicit model that was 16$\times$ deeper (522M parameters). Similarly, for Navier-Stokes equations, the implicit FNO achieved lower error than an explicit counterpart with $>10\times$ more parameters. This confirms that the implicit formulation offers an expressive advantage that cannot be efficiently replicated by merely scaling up explicit solvers.
>
> **Regarding the definition of "fairness,"** if one strictly equates fairness to inference FLOPs, a comparable setup might be a $10\times$ deeper explicit network versus a 10-iteration implicit model. However, we argue that this comparison ignores three critical practical constraints where implicit models exhibit superior efficiency:
> * Training memory: Implicit models enjoy constant training memory cost due to weight tying. In contrast, scaling the depth of explicit models causes linear growth in activation storage, quickly leading to OOM errors during training (as confirmed by our scaling experiments).
> * Parameter Efficiency: The implicit update $y_{t+1} = G_\theta(y_t,x)$ reuses the same $\theta$. The explicit equivalent $y_{t+1} = G_{\theta_t}(y_t,x)$ requires storing distinct parameters for every layer. Consequently, explicit models cannot scale depth without a linear increase in parameter storage size, whereas implicit models can extend iterations indefinitely with a fixed memory footprint.
> * Inference Flexibility: An explicit model has a fixed computational cost baked into its architecture; it cannot adapt to varying constraints. An implicit model offers a dynamic trade-off: the same trained model can run for fewer iterations on resource-constrained hardware (fast, lower precision) or for more iterations on powerful hardware (slower, high precision) without retraining. This capability is unique to the implicit formulation.
>
>
>
> > Is the finding of the ability of training implicit GCNs with more overparameterization a new finding or is it more a confirmation of previous work?
>
> Training implicit GCNs is not new, as we have pointed out at the bottom of Page 8 (Lines 428-431 in the revised paper), but applying implicit GCNs in the context of solving LP via bipartite graph representation is new.

---

> > ### Comment · Reviewer_KADN · 2025-11-25
> >
> > I would like to thank the authors for their clarifications and additional experiments.
> > The responses to the more critical reviewers are satisfactory to me as well.
> > I believe this work is a clear accept and keep my score.

---

### Official Review · Reviewer_xk5a · 2025-10-28

**Soundness:** 4
**Presentation:** 4
**Contribution:** 4
**Rating:** 8
**Confidence:** 4

**Summary:**

The paper defines a theory of expressivity for implicit models (also called deep equilibrium models or fixed point models). Doing so, the paper provides a theoretically sound contribution towards (and essentially a solution to) the open problem of understanding the richness of implicit neural networks. Implicit architectures have been the subject of much attention over the last several years, but a precise characterization of what can be represented by implicit models is absent in the literature.  The paper contains three case studies illustrating the new expressivity results and defining appropriate "regular" implicit models.

**Strengths:**

The key result is that any locally Lipschitz map (on a bounded domain) can be represented as the fixed point of a "regular" implicit operator, and vice versa.  Here "regular" describes a map that is globally Lipschitz in both variables, but not uniformly so. This result means that a regular implicit model can express maps with singularities and unbounded locally-Lipschitz behavior.

In short, I find this result, stated in Theorems 2.4 (sufficiency) and 2.5 (necessity), to be remarkable and clearly deserving acceptance.

The presentation in Introduction and Main result (sections 1 and 2) is lucid.

In each of the case studies (image reconstruction, scientific computing, and operations research), (i) the target map is locally Lipschitz (ii) an implicit model is proposed that, after standard training, is verified to be regular, and (iii) indeed as the interation count increases the implicit model realized increasingly accurate maps.

**Weaknesses:**

The abstract sats "these compact models can match or even exceed larger explicit networks" --- it is unclear what "exceed" means. In what sense? Accuracy? Memory or computational efficiency?  (the same confusing language at line 42)

The document should address the following topic: Do there exist explicit/feedforward networks  capable of expressing locally Lipschitz target maps?  I feel such a comparison should be prominently addressed in the introduction.

Could sparse reconstruction problems (as in compressive sensing) be a special case of the image reconstruction approach in case study 1?

**Questions:**

Line 32: "at inference time we apply"--- but there might be better algorithms to apply than the simple Picard iteration, correct?

Line 128: definition of regular implicit operator. Perhaps you could clarify whether the map has any joint Lipschitz properties.

Theorem 2.5: "must be locally Lipschitz" -> "is locally Lipschitz".  (but I agree that this is simply a stilistic matter).

Line 174: "growth" -> "grows"

Line 304: "Solving ... solves" = typo

Line 395: "Programm"

Line 1157 "postive"

---

> ### Author Response · Authors · 2025-11-23
> **Official Rebuttal**
>
> Thank you very much for your encouraging comments! Please find our responses below:
>
> > The abstract sats "these compact models can match or even exceed larger explicit networks" --- it is unclear what "exceed" means. In what sense? Accuracy? Memory or computational efficiency? (the same confusing language at line 42)
>
> By "exceed," we specifically refer to predictive accuracy. We intended to highlight the empirical observation that implicit models can often achieve comparable or superior accuracy compared to explicit counterparts that are significantly larger in parameter size.
>
> In the new draft, we have revised the Abstract and Introduction to explicitly state "match or even exceed the accuracy of..." to ensure precision.
>
> > The document should address the following topic: Do there exist explicit/feedforward networks capable of expressing locally Lipschitz target maps? I feel such a comparison should be prominently addressed in the introduction.
>
> First, consider the setting where the model is subject to a global Lipschitz constraint (e.g., $\text{Lip}(f_\theta) \le 1$), which is common for robustness and stability. In this case, explicit feedforward networks are mathematically strictly limited to representing globally 1-Lipschitz maps [1]. Consequently, they are fundamentally incapable of expressing locally Lipschitz targets whose gradients become arbitrarily large (such as $1/x$ near zero). In contrast, our work demonstrates that implicit models break this barrier: a simple, globally regular operator (Lipschitz in $x$, contractive in $y$) can generate complex, locally Lipschitz fixed-point maps via iteration. This "Simple Operator $\to$ Complex Fixed Point" mechanism is the core difference claimed in our paper.
>
> Second, if we remove constraints, [2] have shown that deep ReLU networks can indeed approximate locally Lipschitz functions on arbitrary compact sets. However, achieving high precision for such complex targets requires the explicit model size (depth/width) to grow arbitrarily large. Here lies the crucial distinction: explicit models scale expressivity with model size, whereas implicit models are able to scale expressivity with test-time iterations. This allows implicit models to represent increasingly complex functions dynamically without adding parameters.
>
> Following the reviewer's suggestion, we have incorporated a concise version of this comparison **at the end of the Introduction** to ensure it is prominently addressed. Additionally, we have included a comprehensive detailed analysis in **Appendix J**, and added a brief comparison following our main results (at **the bottom of Page 3**).
>
> [1] Murari et al. (2025) "Approximation theory for 1-Lipschitz ResNets."
> [2] Beneventano et al. (2021) "Deep neural network approximation theory for high-dimensional functions."
>
> > Could sparse reconstruction problems (as in compressive sensing) be a special case of the image reconstruction approach in case study 1?
>
> Yes, by setting the manifold as the union of subspaces spanned by sparse vectors, $$\mathbb{M}=\{ x \in \mathbb{R}^d : \|x\|_0 \leq s \}$$ where $s$ is the sparsity level. In that case, the manifold penalty $d^2(y,\mathbb{M})$ encourages sparsity (instead of “natural image” structure), and with $\mathbf{A}$ taken as the sensing matrix, the resulting variational problem reduces to a standard sparse reconstruction / compressive sensing model, where the "learned deep prior" is replaced by a fixed, analytical "sparsity prior."
>
>
> > Line 32: "at inference time we apply"--- but there might be better algorithms to apply than the simple Picard iteration, correct?
>
> Yes, we revised the sentence as follows in the new draft:
>
> "At inference time, the fixed point is obtained via a numerical root-finding solver. While advanced algorithms (e.g., Anderson acceleration or Broyden's method) exist, the canonical approach is the Picard iteration."
>
> > Line 128: definition of regular implicit operator. Perhaps you could clarify whether the map has any joint Lipschitz properties.
>
> Regular implicit operator is not required to be joint Lipschitz. We clarified this point in the revised paper.
>
> > Theorem 2.5: "must be locally Lipschitz" -> "is locally Lipschitz". (but I agree that this is simply a stilistic matter).
>
> Fixed.
>
> > Line 174: "growth" -> "grows"
>
> Fixed.
>
> > Line 304: "Solving ... solves" = typo
>
> Fixed.
>
> > Line 395: "Programm"
>
> Fixed.
>
> > Line 1157 "postive"
>
> Fixed.

---

### Official Review · Reviewer_La71 · 2025-11-01

**Soundness:** 4
**Presentation:** 4
**Contribution:** 2
**Rating:** 2
**Confidence:** 3

**Summary:**

This paper adds to the understanding of how expressive implicit models can be.

Implicit models are an interesting vein for adaptive test-time computation for adaptive model quality. Older related work in adaptive test time compute has focused on using more/less features (see e.g. Parrish et al. JMLR 2013) or more/less base models in an ensemble (see e.g. Wang 2021 Quit When You Can), this approach is a bit more like the strategy of only using as many base models as you have time for from an ensemble, but with the nice difference that you just need to re-use the same model over and over.

**Strengths:**

This paper argues that implicit models are even more interesting because they can more easily express certain kinds of complexity than standard explicit models, especially certain kinds of nonlinearities where the iterative mapping G can be reasonably smooth but iterating on it leads to sharp behavior that may otherwise be difficult to model.

They conclude that their insights mean one popular strategy of implicit models is often over-regularized, and suggest using other regularization strategies instead.

**Weaknesses:**

Modeling Novelty: Bit weird to act like this is a totally new idea guys - using iterative models is an ancient strategy in signal processing (and image processing) and controls. Kalman filtering, iterative filtering, etc,  the idea of repeatedly applying the same model, and that model having free parameters, is not at all a new idea, especially in areas like image deblurring.

Theory Novelty: Theorems 2.4 & 2.5 really seem like something that mathematicians who study fixed-point stuff would already know and have studied in the 70’s.   Does the ML angle really bring a novelty to this problem that means mathematicians who study similar compositions wouldn’t have already studied this exact question? Sadly I’m not a fixed-point math expert, but I’d like to hear from one before being willing to believe the presented theorem is at all novel.

Experiments: 3 case studies were interesting but small and lacked enough comparisons that I was able to be convinced of anything.

Reminder to use {} in your bib tex for capitalization (e.g. Euclidean) and make sure the same conference has the same title throughout your references.

Overall Contribution: even ignoring my kvetching about novelty, I didn't the contributions significant enough to merit inclusion at NeurIPS, but I could be swayed if other reviewers helped me see why their insights are more unexpected and interesting than my read.

**Questions:**

Have you talked to pure mathematicians about Theorems 2.4 and 2.5? Does the ML angle really bring a novelty to this problem that means mathematicians who study similar compositions wouldn’t have already studied this exact question and you could just be citing their work?

---

> ### Author Response · Authors · 2025-11-23
> **Official Rebuttal: Modeling Novelty**
>
> Thank you very much for your feedback! Please find our responses below:
>
> > Modeling Novelty: Bit weird to act like this is a totally new idea guys - using iterative models is an ancient strategy in signal processing (and image processing) and controls. Kalman filtering, iterative filtering, etc, the idea of repeatedly applying the same model, and that model having free parameters, is not at all a new idea, especially in areas like image deblurring.
>
> We respectfully clarify that we do NOT claim to have invented the implicit model architecture or iterative strategies. We fully recognize that iterative methods (such as Kalman filtering and iterative restoration) have a foundational history in signal processing and control. As explicitly stated in our Abstract and Introduction, we characterize implicit models as an "emerging model class" within the specific context of modern deep learning, where they have recently already demonstrated impressive empirical results.
>
> **Our novel contribution lies NOT in proposing the architecture itself, but in providing a new theoretical framework for understanding its capabilities.** Specifically, we offer a rigorous proof that the expressive power of these implicit models are able to scale with test-time iterations without adding parameters. This highlights a fundamental distinction from explicit models, which scale expressivity only through predetermined model sizes (width and depth). Furthermore, we establish a sharp boundary for this model class, mathematically defining exactly which types of target mappings can (and cannot) be expressed by iterating a fixed operator.

---

> ### Author Response · Authors · 2025-11-23
> **Official Rebuttal: Theory Novelty - Part I**
>
> > Theory Novelty: Theorems 2.4 & 2.5 really seem like something that mathematicians who study fixed-point stuff would already know and have studied in the 70’s. Does the ML angle really bring a novelty to this problem that means mathematicians who study similar compositions wouldn’t have already studied this exact question? Sadly I’m not a fixed-point math expert, but I’d like to hear from one before being willing to believe the presented theorem is at all novel.
>
>
> We respectfully disagree with the reviewer's concern that our theoretical contributions are merely restatements of studies from the 1970s. To verify this, we conducted an extensive review of four authoritative texts that comprehensively cover the field from classical foundations to modern analysis:
> * Dontchev & Rockafellar (2009), "_Implicit Functions and Solution Mappings_": This text is widely regarded as the definitive reference for the stability analysis of solution mappings and implicit functions
> * Granas & Dugundji (2003), "_Fixed Point Theory_": Described by Felix Browder as "certainly the most learned book ever likely to be published on this subject," this text provides an exhaustive account of classical and modern fixed-point theory, making it a standard for verifying the novelty of results.
> * Krantz & Parks (2002), "_The Implicit Function Theorem: History, Theory, and Applications_": This work provides an exhaustive historical and theoretical treatment of the Implicit Function Theorem, tracing its development from early analysis to modern applications.
> * Kirk & Goebel (2001), "_Handbook of Metric Fixed Point Theory_": This handbook is the authoritative collection of results specifically focused on metric fixed-point theory and contraction mappings.
>
> These works are exhaustive and were published well after the 1970s; it is highly unlikely that fundamental results from that era would be omitted. **After a careful review, we found no existing results that exactly restate or cover our Theorem 2.4 (Sufficiency) and Theorem 2.5 (Necessity).** In particular:
> * **Theorem 2.4 is brand new.** Classic texts focus on the forward problem: Given an operator $\mathcal{G}$, does a fixed point $y_\ast$ exist, is it unique, and is it stable?. In contrast, our Theorem 2.4 addresses another problem: Given a target map $y_\ast(x)$, does there exist a implicit operator $\mathcal{G}(\cdot,x)$ such that its fixed point exactly reproduces $y_\ast$? These are fundamentally different mathematical questions: **one concerns existence of fixed point $y_\ast$, the other concerns existence of implicit operators.**
> * **Theorem 2.5 is an essential extension.** The exact form of this theorem does not appear in standard references and hence we prove it. **We have already explicitly clarified its relationship to the literature in the paper.** As noted in Remark A.6 (Page 23), the closest statement is Theorem 1A.4 in Dontchev & Rockafellar (2009). Our result extend this by relaxing two uniformity requirements: (i) the contraction modulus $\mu(x)$ is allowed to vary continuously with $x$ rather than being a single global constant; and (ii) the Lipschitz constant of $x \mapsto \mathcal{G}(y,x)$ may grow linearly with $\|y\|$ rather than being uniformly bounded. **Crucially, these non-uniform requirements are key to our findings.** It is precisely this lack of global uniformity that allows the model to capture the behavior illustrated in Figure 1: a sequence of continuous functions can approximate a discontinuous (or arbitrarily steep) function only if the convergence is not uniform; otherwise, the limit of continuous functions would necessarily remain globally continuous.

---

> ### Author Response · Authors · 2025-11-23
> **Official Rebuttal: Theory Novelty - Part II**
>
> **(Clarification on Potential Misunderstandings).** We anticipate that the reviewer might conflate our contributions with three specific classical results. We explicitly distinguish our work from them as follows:
> * **Parametric Contraction Mapping Theorem (Standard Stability):** We acknowledge that standard results, such as Theorem 1A.4 in Dontchev & Rockafellar (2009), provide a baseline for fixed-point stability. However, as detailed earlier, our Theorem 2.5 is an essential extension of this classical theory, specifically extended to handle the non-uniform conditions inherent to our setting.
> * **Bessaga's Converse Theorem (1959):** This topological theorem states that if a map $f$ has a unique fixed point, then there exists a metric under which $f$ is a contraction. This is fundamentally different from our Theorem 2.4 (Sufficiency). Bessaga’s theorem focuses on the existence of a metric to suit a given map. In contrast, our work focuses on the construction of an operator $\mathcal{G}$ on a fixed standard metric space (Euclidean) to exactly match a target function $y_*(\mathbf{x})$. One is a result on topological remetrization; ours is a result on model expressivity and construction.
> * **Collage Theorem (Barnsley, 1988):** Central to fractal compression, this theorem provides an approximation bound: intuitively, it states that if a contraction map $f$ maps a target $y$ close to itself (i.e., $d(y, f(y)) < \epsilon$), then the fixed point of $f$ is close to $y$. This is an approximation result used to minimize reconstruction error. In contrast, our Theorem 2.4 characterizes the exact representational capacity of implicit models.
>
> **(Why such a result appears in this era, rather than the 1970s).** As the reviewer correctly noted, the strategy of using iterative models—such as Kalman filtering or iterative signal processing—is well-established in classical control and engineering. However, a key historical distinction explains why our expressivity result (Theorem 2.4) is a product of the modern deep learning era:
> * **The Classical Context (1970s):** Due to computational limitations, classical models typically employed an operator $\mathcal{G}$ with very few tunable parameters. Consequently, the theoretical focus was primarily analytic: Given a specific, pre-defined $\mathcal{G}$, does a fixed point exist, and under what conditions will iterations converge?
> * **The Modern Context (Deep Learning):** Deep neural networks provide a fundamentally new toolkit. Today, $\mathcal{G}$ is parameterized by millions of weights and can be trained to fit arbitrarily complex data. **This shift from "analyzing a fixed operator" to "learning a flexible operator" motivates a new theoretical question: What is the ultimate limit of this flexibility?** Instead of analyzing specific, rigid structures, we must now characterize the expressivity of the **entire function space** of regular implicit operators (Definition 2.3). Our work addresses this modern problem, determining whether a $\mathcal{G}$ exists to represent a given target $\mathcal{F}$, which is a question of model capacity central to deep learning, but less relevant to the constrained modeling paradigms of the 1970s.

---

> ### Author Response · Authors · 2025-11-23
> **Official Rebuttal: Experiments**
>
> > Experiments: 3 case studies were interesting but small and lacked enough comparisons that I was able to be convinced of anything.
>
> We respectfully point out that while our experiments are smaller than industrial-scale LLM training, they are rigorously aligned with—and often exceed—the standard benchmarks used in recent literature for these specific domains. The scale of our data and models matches recent studies such as [1,2] for inverse problems, [3] for PDEs, and [4,5] for operations research.
>
> To further address the reviewer's concern regarding comparisons and scale, **we have added significant new experiments:**
> * **Scaling Baselines to Limit** (Inverse Problems & PDEs): We aggressively scaled the explicit baseline models in depth and width until they hit hardware memory limits (CUDA Out-of-Memory). The results were decisive: simply adding capacity to explicit models yields diminishing returns. For example, in image deblurring, the implicit model (32.6M parameters) outperformed an explicit model that was 16$\times$ deeper (522M parameters). Similarly, for Navier-Stokes equations, the implicit FNO achieved lower error than an explicit counterpart with $>10\times$ more parameters. This confirms that the implicit formulation offers an expressive advantage that cannot be efficiently replicated by merely scaling up explicit solvers. **Details are in Appendices G and H (Pages 42 & 44).**
> * **New Case Study: LLM Reasoning.** We extended our validation to the domain of language modeling using the pre-trained looped transformer from [6]. We found that our theory holds even in this discrete domain: as test-time iterations increase, the "empirical Lipschitz" complexity of the model's output rises (saturating around $t=16$), allowing the model to progressively resolve semantic ambiguities (e.g., distinguishing the context of "charge" in physics vs. finance) that earlier iterations fail to capture. This demonstrates that the principle of scaling expressivity with compute extends beyond continuous mathematical problems. **See Section 3.4 for details.**
>
> [1] Zou et al. (2023) "Deep equilibrium learning of explicit regularization functionals for imaging inverse problems."
>
> [2] Daniele et al. (2025) "Deep Equilibrium models for Poisson imaging inverse problems via Mirror Descent."
>
> [3] Marwah et al. (2023) "Deep equilibrium based neural operators for steady-state pdes."
>
> [4] Chen et al. (2023) "On representing linear programs by graph neural networks."
>
> [5] Chen et al. (2024) "Rethinking the capacity of graph neural networks for branching strategy."
>
> [6] Geiping et al. (2025) "Scaling up test-time compute with latent reasoning: A recurrent depth approach."

---

> ### Author Response · Authors · 2025-11-23
> **Official Rebuttal: Others**
>
> > Reminder to use {} in your bib tex for capitalization (e.g. Euclidean) and make sure the same conference has the same title throughout your references.
>
> Fixed in the revised draft.
>
> > Overall Contribution: even ignoring my kvetching about novelty, I didn't the contributions significant enough to merit inclusion at NeurIPS, but I could be swayed if other reviewers helped me see why their insights are more unexpected and interesting than my read.
>
> While we believe we have addressed the specific concerns regarding novelty and experimental scale, we wish to emphasize the core insight that renders our work surprising and significant. We offer a distinct perspective on model's expressive power: whereas standard approaches typically rely on **increasing number of parameters to gain expressivity**, we highlight that implicit models possess a unique capability to **scale their expressive power dynamically via test-time iterations.** This allows them have the capacity to represent increasingly complex functions without adding a single parameter.
>
> Our theory characterizes this mechanism, showing how iterating a regular operator $\mathcal{G}$ allows the model to progressively represent increasingly complex target functions. This allows a single trained model to adapt its precision and complexity based purely on the available inference budget. We believe this rigorous characterization of "test-time scaling"—treating depth as a flexible run-time resource—provides a significant insight into the efficiency and versatility of implicit models.
>
> Finally, as a minor administrative note regarding the venue mentioned in the review, we wish to clarify that **this submission is under consideration for ICLR, not NeurIPS.**
>
> > Have you talked to pure mathematicians about Theorems 2.4 and 2.5? Does the ML angle really bring a novelty to this problem that means mathematicians who study similar compositions wouldn’t have already studied this exact question and you could just be citing their work?
>
> Due to the double-blind policy, we cannot disclose who we are and who we have talked to. But we can confirm that **the majority of the co-authors hold a Ph.D. in Mathematics, and the entire team possesses extensive expertise in fixed-point theory.**
>
> Theorem 2.4 (Sufficiency): **The machine learning perspective indeed brings a fundamental novelty that classical mathematics did not address.** As detailed in our response regarding the "1970s context," classical theory focused on the analysis of fixed points for pre-defined, often rigid operators. In contrast, deep learning motivates a different problem: characterizing the entire function space of learnable operators to determine if a $\mathcal{G}$ exists to exactly represent a specific target $\mathcal{F}$. This question of representational capacity is unique to the modern era of flexible neural parameterization and is not covered by classical fixed-point existence theorems.
>
> Theorem 2.5 (Necessity): We agree that this result follows the spirit of the classic parametric implicit function theorem. However, after an exhaustive search of authoritative texts (mentioned before), we could not find an existing theorem that matches our specific assumptions—namely, one that relaxes global uniformity to allow for adaptive contraction moduli and linearly growing Lipschitz constants. Therefore, **we provide the proof for this necessary extension to the standard theory, while explicitly citing the foundational literature.**

---

### Official Review · Reviewer_Ybb3 · 2025-11-01

**Soundness:** 2
**Presentation:** 2
**Contribution:** 1
**Rating:** 2
**Confidence:** 4

**Summary:**

This paper studies the expressive power of implicit neural models, analyzing how iterative fixed-point dynamics affect model capacity both theoretically and empirically.

**Strengths:**

The paper provides a rigorous theoretical analysis of the expressive power of implicit neural models, offering how iterative fixed-point dynamics influence expressive capacity.

**Weaknesses:**

- The paper’s framing of "implicit models" is confusing and deviates from the standard definition in the community, where implicit models such as Deep Equilibrium Models find an **equilibrium point** via a numerical solver like root-finding techniques rather than explicitly unrolling the recurrence; here, the authors effectively treat implicit models as **recurrent networks** and study the number of unrolled iterations, which misrepresents the core concept, and the term "test-time computation" is also confusing in this context.

- The theoretical analysis relies on well-known tools such as function composition, contraction mappings, and ODE formulations, offering limited novelty or new insights beyond existing expressivity analyses.

- The experiments focus on small toy problems and shallow MLP-like architectures without evaluation on modern large-scale tasks or established implicit models; moreover, there is no discussion of how the proposed analysis or conclusions scale with model size, data, or computation, which is central to understanding expressivity and scaling laws in contemporary deep learning.

- The paper overlooks key studies on implicit and equilibrium models, including those addressing training convergence, generalization, and the theoretical connections between implicit models, explicit deep networks, and kernel methods, which weakens its positioning within the broader literature and limits its conceptual impact.

**Questions:**

See the weakness section.

---

> ### Author Response · Authors · 2025-11-23
> **Official Rebuttal: Terminology - Part I**
>
> Thank you very much for your feedback! Please find our responses below:
>
> > The paper’s framing of "implicit models" is confusing and deviates from the standard definition in the community, where implicit models such as Deep Equilibrium Models find an equilibrium point via a numerical solver like root-finding techniques rather than explicitly unrolling the recurrence; here, the authors effectively treat implicit models as recurrent networks and study the number of unrolled iterations, which misrepresents the core concept, and the term "test-time computation" is also confusing in this context.
>
> We respectfully clarify that our framing is fully consistent with the standard definitions in the deep equilibrium literature. The reviewer’s concern appears to stem from a distinction between implementation (how the gradient is computed) and definition (what the model represents).
>
> **Theoretical Definition:** Our main theoretical contributions (Theorems 2.4 and 2.5) are explicitly defined in terms of the fixed point (equilibrium) $y_*(x)$ satisfying $y_* = \mathcal{G}(y_*, x)$, rather than any specific finite unrolling. Our theory characterizes the properties of this equilibrium map itself.
>
> **Model Structure (Weight-Tying):** The defining structural characteristic of an implicit model is that the operator $\mathcal{G}\_{\theta}$ shares parameters $\theta$ across all iterations $t$:
> $$y\_{t+1} = \mathcal{G}\_{\theta}(y\_{t},x)$$
> This weight-tied structure allows the model to be conceptually infinite-depth and converge to an equilibrium. **This stands in sharp contrast to standard explicit networks or unrolled algorithms with untied weights** ($y\_{t+1} = \mathcal{G}\_{\theta\_t}(y\_{t},x)$) Such models are strictly confined to a finite depth $T$, as storing an infinite sequence of distinct parameters $\\{\theta\_t\\}_{t=0}^\infty$ is computationally infeasible. All our findings are strictly built on the weight-tied formulation. All our findings are strictly built on the weight-tied formulation.
>
> **Training Dynamics (Equivalence of Approaches):** The distinction between "unrolling" training (minimizing $\ell(y_T)$):
> $$\min\_\theta \ell(y\_T), ~~~ y\_{t+1} = \mathcal{G}\_{\theta}(y\_{t},x),~~ t=0,1,2,\cdots,T-1$$ and "implicit differentiation" (minimizing $\ell(y_*)$):
> $$\min\_\theta \ell(y\_\ast), ~~~ y\_{\ast} = \mathcal{G}\_{\theta}(y\_{\ast},x)$$ **is a matter of numerical implementation, not model category.**
> * As established in prior literature [1], **unrolled training is mathematically equivalent to a standard Neumann series approximation of the implicit gradient**. Specifically, implicit differentiation requires inverting the Jacobian $(I - J\_{\mathcal{G}\_\theta})^{-1}$; finite unrolling effectively approximates this inverse via a Neumann series expansion. This is a widely adopted technique in the implicit model community to avoid the instability and cost of exact inversion.
> * According to [1], implicit training is simply the limit of unrolled training: as $T \to \infty$, the gradient $\nabla_\theta \ell(y\_T)$ converges to the implicit gradient $\nabla_\theta \ell(y\_*)$.
>
> Overall, unrolling and root-finding are merely two numerical implementations for approximating the same fixed point, $y_*(x)$, and technically speaking, **there is no significant gap or distinction between the two.** Theoretically, infinite unrolling converges exactly to $y_*(x)$. In practice, unrolling depth simply controls the trade-off between accuracy and computational cost—a dynamic strictly analogous to setting the error tolerance in implicit root-finding solvers. Particularly in our paper,
> * **Case Studies 1 & 2:** We strictly employ implicit differentiation (minimizing $\ell(y_*)$) via root-finding, not unrolling. This detail is explicitly documented in the Appendix. **We would like to clarify that unrolling was never used to train implicit models for these experiments,** which appears to be a misunderstanding by the reviewer.
> * **Case Study 3:** We adopt unrolling to train implicit GNNs, which serves as a truncated Neumann approximation of the implicit GNN gradient. This is not a deviation from the core concept but a standard practice in the Implicit GNN literature, where Neumann approximations are frequently employed to scale training to large graphs (e.g., [2]). **To further address the reviewer's concern, we have added the above discussions as a specific remark in the revised draft (Appendix I, Page 46) and detailed discussions in (Appedix F, Page 40).**
>
> Overall, whether viewed through the lens of model structure or training methodology, our terminology is accurate and consistent with standard definitions.

---

> ### Author Response · Authors · 2025-11-23
> **Official Rebuttal: Terminology - Part II**
>
> **(Test-time compute).** In our context, "test-time compute" refers specifically to the computational budget (measured by the number of solver iterations) allocated to approximate the fixed point at inference. Our results demonstrate that the expressive power of a simple, regular operator $\mathcal{G}$ is not static; rather, it scales with these iterations, allowing the model to represent increasingly complex mappings. This terminology aligns with recent literature [3] and has been explicitly defined in our Introduction (Page 1, Lines 35-36).
>
> [1] Geng et al. "On training implicit models." NeurIPS 2021.
>
> [2] Baker et al. "Implicit Graph Neural Networks: A Monotone Operator Viewpoint." ICML 2023.
>
> [3] Geiping et al. "Scaling up Test-Time Compute with Latent Reasoning: A Recurrent Depth Approach." 2025.

---

> ### Author Response · Authors · 2025-11-23
> **Official Rebuttal: Theory Novelty**
>
> > The theoretical analysis relies on well-known tools such as function composition, contraction mappings, and ODE formulations, offering limited novelty or new insights beyond existing expressivity analyses.
>
> **Utilizing standard tools doesn't imply that the resulting analysis itself is standard.** On the contrary, we believe that the core constructions in our proofs (specifically the design of the operator $\mathcal{G}$ in Eq. (10) and the construction of $\varepsilon(x)$ in Theorem A.1) are novel in mathematics and yield distinct, practical insights into the machine learning context.
>
> **Regarding "existing expressivity analyses" on implicit models, we have comprehensively reviewed the relevant literature in our Introduction (Page 2, Lines 54–61):** "_To our knowledge, these questions remain largely open. While universality has been touched upon in specific settings (Bai et al., 2019; Marwah et al., 2023) and separation results have demonstrated advantages over explicit models (Wu et al., 2024), a complete characterization of the representable function class of implicit models (and hence a direct answer to questions (Q1) and (Q2)) is still missing. Unlike studies focusing on infinite-width limits and kernel connections (Gao et al., 2022; Feng & Kolter, 2023; Ling et al., 2024), our work fills this gap from a nonparametric, function-space perspective, establishing that an implicit model's expressive power scales with test-time compute._"
>
> If the reviewer has **_specific references_** in mind that we may have overlooked, we would be grateful for the pointers.
>
>
>
> To further address the reviewer's concern, we would like to further discuss the novelty of the main theorems. After extensive review on the literature, we found no existing results that exactly restate or cover our Theorem 2.4 (Sufficiency) and Theorem 2.5 (Necessity). In particular:
> * **Theorem 2.4 is brand new.** Classical texts focus on the forward problem: Given an operator $\mathcal{G}$, does a fixed point $y_\ast$ exist, is it unique, and is it stable?. In contrast, our Theorem 2.4 addresses another problem: Given a target map $y_\ast(x)$, does there exist a implicit operator $\mathcal{G}(\cdot,x)$ such that its fixed point exactly reproduces $y_\ast$? These are fundamentally different mathematical questions: one concerns existence of fixed point $y_\ast$, the other concerns existence of implicit operators.
> * **Theorem 2.5 is an essential extension.** The exact form of this theorem does not appear in standard references and hence we prove it. **We have already explicitly clarified its relationship to the literature in the paper.** As noted in Remark A.6 (Page 25), the closest statement is Theorem 1A.4 in Dontchev & Rockafellar (2009). **Our result extend this by relaxing two uniformity requirements:** (i) the contraction modulus $\mu(x)$ is allowed to vary continuously with $x$ rather than being a single global constant; and (ii) the Lipschitz constant of $x \mapsto \mathcal{G}(y,x)$ may grow linearly with $\|y\|$ rather than being uniformly bounded. **Crucially, these non-uniform requirements are key to our findings.** It is precisely this relaxation of global uniformity that allows the model to capture the behavior illustrated in Figure 1: a sequence of continuous functions can approximate a discontinuous (or arbitrarily steep) function only if the convergence is not uniform; otherwise, the limit of continuous functions would necessarily remain globally continuous.

---

> ### Author Response · Authors · 2025-11-23
> **Official Rebuttal: Experiments**
>
> > The experiments focus on small toy problems and shallow MLP-like architectures without evaluation on modern large-scale tasks or established implicit models; moreover, there is no discussion of how the proposed analysis or conclusions scale with model size, data, or computation, which is central to understanding expressivity and scaling laws in contemporary deep learning.
>
> First, we would like to respectfully clarify some **obvious misunderstandings** regarding our experimental setup and architecture choices:
> * **Modern Architectures vs. "Shallow MLPs":** We do **NOT** rely on shallow MLPs. Instead, we employ state-of-the-art, domain-specific backbones: DRUNet for inverse problems, Fourier Neural Operators (FNO) for PDEs, and Graph Neural Networks (GNNs) for linear programming. These are widely recognized, deep architectures in their respective fields.
> * **Established Implicit Models:** Our evaluation is grounded in established implicit formulations. Case Study 1 utilizes Deep Equilibrium (DEQ) PGD and HQS models , and Case Study 2 employs the DEQ-FNO. They are both established implicit models and we have pointed them out (Lines 236-241 on Page 5; Lines 354-361 on Page 7). In addition, Case Study 3 introduces a novel application of implicit models to the field of "Learning to Optimize" (L2O). Here, we demonstrate that our implicit approach not only validates our theoretical predictions but also achieves state-of-the-art performance compared to existing explicit baselines. Therefore, our experimental suite is designed to provide a comprehensive evaluation, **including both established benchmarks and demonstrations on new frontiers.**
>
> Next, let's discuss "toy problems vs modern large-scale tasks." We respectfully point out that, while our experiments are smaller than industrial-scale LLM training, they are rigorously aligned with—and often exceed—the standard benchmarks used in recent literature for these specific domains. The scale of our data and models matches recent studies such as [1,2] for inverse problems, [3] for PDEs, and [4,5] for operations research.
>
> Finally, to further address the reviewer's concern regarding experiment sizes and scale, **we have added significant new experiments:**
> * **Scaling Baselines to Limit** (Inverse Problems & PDEs): We aggressively scaled the explicit baseline models in depth and width until they hit hardware memory limits (CUDA Out-of-Memory during training). The results were decisive: simply adding capacity to explicit models yields diminishing returns. For example, in image deblurring, the implicit model (32.6M parameters) outperformed an explicit model that was 16$\times$ deeper (522M parameters). Similarly, for Navier-Stokes equations, the implicit FNO achieved lower error than an explicit counterpart with $>10\times$ more parameters. This confirms that the implicit formulation offers an expressive advantage that cannot be efficiently replicated by merely scaling up explicit solvers. **Details are in Appendices G and H (Pages 42 & 44).**
> * **New Case Study: LLM Reasoning.** We extended our validation to the domain of language modeling using the pre-trained looped transformer from [6]. We found that our theory holds even in this discrete domain: as test-time iterations increase, the "empirical Lipschitz" complexity of the model's output rises (saturating around $t=16$), allowing the model to progressively resolve semantic ambiguities (e.g., distinguishing the context of "charge" in physics vs. finance) that earlier iterations fail to capture. This demonstrates that the principle of scaling expressivity with compute extends beyond continuous mathematical problems. **See Section 3.4 for details.**
>
> [1] Zou et al. (2023) "Deep equilibrium learning of explicit regularization functionals for imaging inverse problems."
>
> [2] Daniele et al. (2025) "Deep Equilibrium models for Poisson imaging inverse problems via Mirror Descent."
>
> [3] Marwah et al. (2023) "Deep equilibrium based neural operators for steady-state pdes."
>
> [4] Chen et al. (2023) "On representing linear programs by graph neural networks."
>
> [5] Chen et al. (2024) "Rethinking the capacity of graph neural networks for branching strategy."
>
> [6] Geiping et al. (2025) "Scaling up test-time compute with latent reasoning: A recurrent depth approach."

---

> ### Author Response · Authors · 2025-11-23
> **Official Rebuttal: Broader Contextual Discussion**
>
> > The paper overlooks key studies on implicit and equilibrium models, including those addressing training convergence, generalization, and the theoretical connections between implicit models, explicit deep networks, and kernel methods, which weakens its positioning within the broader literature and limits its conceptual impact.
>
> We respectfully clarify that our paper’s primary focus is the expressive power of implicit models, rather than training dynamics or generalization. Given the vastness of the field (seminal works like Bai et al. (2019) alone have nearly 1,000 citations), it is neither feasible nor necessary for a technical paper to survey every adjacent subfield, such as training dynamics or generalization bounds, which are orthogonal to our contributions.
>
> However, to accommodate the reviewer's request for a broader contextualization, **we have added a supplementary discussion covering these topics in Appendix J of the revised draft, and revised our introduction (See Lines 58-61).** Specifically, we have added citations for:
> * Training convergence: [1,2,3]
> * Generalization: [4]
> * Theoretical connections between implicit models, explicit deep networks, and kernel methods: [5,6,7]
>
> We placed this expanded survey in the Appendix J to strictly adhere to page limits while ensuring our main theoretical results and new experiments remain fully detailed in the main text. If there are specific additional papers the reviewer considers critical, we would be happy to incorporate them.
>
> [1] Geng et al. "On training implicit models." NeurIPS 2021.
>
> [2] Ling et al. "Global convergence of over-parameterized deep equilibrium models." AISTATS 2023.
>
> [3] Truong. "lobal convergence rate of deep equilibrium models with general activations." TMLR 2025.
>
> [4] Fung & Berkels. "A Generalization Bound for a Family of Implicit Networks." 2024.
>
> [5] Gao et al. "A global convergence theory for deep relu implicit networks via over-parameterization." ICLR 2022.
>
> [6] Feng & Kolter. "On the Neural Tangent Kernel of Equilibrium Models." 2023.
>
> [7] Ling, Z. et al. "Deep Equilibrium Models are Almost Equivalent to Not-so-deep Explicit Models for High-dimensional Gaussian Mixtures." ICML 2024.

---

> > ### Comment · Reviewer_Ybb3 · 2025-11-25
> >
> > Thank you for the authors’ detailed rebuttal and the additional experiments. After reviewing the response and the comments from other reviewers, I find that several clarifications actually reinforce my original concerns.
> >
> > In implicit models such as DEQ layers, the key idea is that "*the layer is defined by a joint equilibrium condition between the input and output*" (see the original DEQ paper and Slide 5 of the [NeurIPS implicit-layers tutorial](https://implicit-layers-tutorial.org/implicit_tutorial.pdf), not by the iterative trajectory used to approximate that equilibrium. Unrolling is only one numerical method, and different solvers can trace different approximation paths under the same compute budget. Thus, the expressivity of an implicit model is **capped** by the equilibrium map itself. Studying "expressive power as a function of test-time compute" therefore reflects the behavior of the solver’s approximation error, not an inherent increase in the model’s expressivity. In this sense, the rebuttal’s emphasis on the equivalence between unrolling and implicit differentiation further strengthens this conceptual concern. The review also want to emphasize that This setting is also fundamentally different from recurrent or looped networks, whose equilibrium state may not exist or unknown.
> >
> > Moreover, regardless of the existence of equilibrium state, the observation that repeatedly applying a simple operator increases representational complexity is not new from an ML standpoint. Prior work, such as [Hayou et al. (2021, Prop. 2)](https://proceedings.mlr.press/v130/hayou21a/hayou21a.pdf), already demonstrates how expressivity grows with depth or, in the authors’ terminology, with test-time compute. Although the architectures and proof techniques differ, the underlying insight remains similar.
> >
> > The new experiments, e.g., scaled explicit baselines, LLM reasoning, and implicit FNOs, further reinforce this assessment. The performance gains come from running more iterations and getting closer to the fixed point defined by the equilibrium, which naturally leads to diminishing returns as the solver converges. These improvements therefore reflect reduced approximation error, not increased expressive capacity of the implicit model itself. Consequently, the central conceptual claim remains unconvincing.
> >
> > Given these considerations, and noting that other reviewers also questioned the contribution from the purely mathematical perspective, I maintain my original evaluation.

---

> > > ### Author Response · Authors · 2025-11-26
> > > **Authors' response - Part II**
> > >
> > > **Paragraph 3: What our experiment tells?**
> > >
> > > We respectfully disagree with the interpretation that performance gains are merely "reduced approximation error" and not "increased expressivity." This perspective overlooks the capacity constraints of the model at early iterations of a solver.
> > >
> > > The reviewer notes that iterations reduce error. However, we must ask: why is the error high at low iterations?
> > >
> > > At low iteration count $t$, the high error is **not simply because the solver is "calculating inaccurately"**; it is because the function represented by the model $y_t(\cdot)$ is structurally **too simple to match the target $\mathcal{F}(\cdot)$**. As discussed in Section 2, at $t=1$, the mapping $y_1(\cdot)$ is restricted to be globally Lipschitz. Geometrically, it cannot "bend" sufficiently to reach a complex, locally steep Ground Truth (like the $1/x$ singularity 2 or a shock wave).
> > >
> > > Consider a concrete counter-example: If the target function were naturally simple — for instance, a linear map $\mathcal{F}(x) = 0.5x$ — a trivial implicit layer $\mathcal{G}(y,x) = 0.5x$ would recover it perfectly in just one step ($y_1 = \mathcal{G}(0, x) = 0.5x$), yielding zero error immediately using the same solver as before. While fixed-point solvers are not unique, e.g., Netwon, Halpern, and Anderson techniques, the common fact that more iterations permits higher complexity implies that the bottleneck is expressivity.
> > >
> > > Figure 1 visualizes this vividly: at $t=1$, the model is a smooth line that physically fails to represent the singularity; iterations are the resource that unlocks the capacity to model it.
> > >
> > > Our case-study experiments (Figures 2, 4, 7, and 8) support this dynamic. In all cases, we observe a low empirical Lipschitz constant at the start, confirming that $y_1$ is simple and smooth—which is precisely why the initial error is high. As $t$ increases, the effective Lipschitz constant grows, indicating that the model’s expressivity is being progressively unlocked, thereby allowing the error to decrease.
> > >
> > > **Paragraph 4: Contribution from the purely mathematical perspective**
> > >
> > > We've already addressed this concern in our response to Reviewer La71 regarding "Theory Novelty."
> > >
> > > **Others: Expanding the Horizon**
> > >
> > > Beyond the immediate scope of this discussion, our theoretical framework creates exciting new avenues for research. For instance, how simple can an operator $\mathcal{G}$ be while maintaining high expressivity in the fixed point? How do we mathematically formalize the link between fixed-point iteration depth and the reasoning capabilities observed in modern LLMs? We believe these insights can invite the community to look beyond past views and traditional constraints, and bring essential new vitality to the study of implicit models.
> > >
> > >
> > > We believe we have fully addressed the concerns raised. We respectfully ask that you consider improving your assessment score based on these clarifications.

---

> ### Author Response · Authors · 2025-11-26
> **Authors' response - Part I**
>
> Thanks for the reviewer's comments. Below you can find our response to each paragraph.
>
> **Paragraph 1: Expressivity vs solver's approximation error**
>
> We respectfully invite the reviewer to revisit our main theoretical results, Theorems 2.4 and 2.5 (Page 3), which directly address your concern regarding the "inherent increase" in expressivity due to the model being implicity. **These theorems strictly characterize the behavior of the fixed-point map (the equilibrium map $y_*(x)$) itself, independent of the numerical solver used.**
>
> Our Theorems highlight a *strict advantage* over explicit networks of any finite size. As long as an explicit feedforward network is constrained to be globally $L$-Lipschitz (a common constraint imposed for robustness), it is mathematically incapable of representing target maps with arbitrarily large local slopes (e.g., function $1/x$ near 0), yet its fixed-point map can. Even without the strict constraints, to approximate locally steep target maps, explicit networks require depths/widths (and thus parameter counts) to grow infinitely. In contrast, a single, fixed-size implicit operator can represent the same class of complex target maps through its fixed point. Hence, a greater expressive power is unlocked by fixed-point finding. (On the contrary, Hayou et al. (2021, Prop. 2), which you quote, does not show this result; see below.)
>
> It is precisely based on this primary theoretical result—regarding the capacity of the equilibrium map, not the solver's approximation error—that we can claim the final target map possesses high expressive power. **Had the equilibrium map itself lacked the capacity to represent complex functions, no solver would matter.**
>
> Therefore, the logic of our contribution is as follows:
> * **Main Theorem (Theorems 2.4 & 2.5):** the equilibrium map $y_*$ is highly expressive (covering all locally Lipschitz functions).
> * **Implication:** This theory, combined with the iterative reduction of solver approximation errors, indicates that the iterates $y_t(x)$ can progressively express more complex mappings.
>
> This relationship is explicitly structured in our revised paper: **Contribution 1 (Lines 62-64)** in the Introduction concerns the fixed-point map characterization, while **Contribution 2 (Lines 65-66)** states that "our theory yields a new viewpoint of implicit models" (i.e., scaling with compute, without adding model parameters). The discussion on **"What does our theory imply?" (Page 3)** further elucidates how the iterates evolve based on the properties of the fixed point.
>
> In summary, while we agree with the reviewer that "solver approximation error" is a known concept, your assessment overlooks our primary contribution and main theoretical results (Theorems 2.4 & 2.5), which are novel and strong. All subsequent inferences, including the "expressive power scales with test-time compute" perspective, are deductive consequences of these primary results (please also refer to our response to Reviewer La71 regarding "Theory Novelty").
>
> **Paragraph 2: Implicit map (DEQ) vs Hayou et al (2021): weight-tying vs new parameters per layer.**
>
> We respectfully disagree with "repeatedly applying a simple operator increases representational complexity is not new". In Hayou et al. (2021): **Each layer introduces new parameters** ($y\_{t+1} = \mathcal{G}(y\_t; \theta\_t)$). Their finding that "expressivity increases with depth" essentially means "expressivity scales with model size (parameter count)."
>
> In contrast, a DEQ/implicit layer is defined by a single operator $\mathcal{G}_\theta$. When approximated by iteration (or solved), the update $y\_{t+1} = \mathcal{G}\_\theta(y\_t, x)$ reuses the identical parameters $\theta$ at every step. Consequently, **increasing "test-time compute" (iterations) does not introduce new parameters or increase model size**. In other words, "expressivity scales with iteration of the same layer."

---

### Official Review · Reviewer_FMZf · 2025-11-01

**Soundness:** 3
**Presentation:** 3
**Contribution:** 4
**Rating:** 8
**Confidence:** 2

**Summary:**

Prior works in deep equilibrium models have shown that these networks can outperform regular feedforward models with the same architecture as well as larger models by using more test-time iterations. This paper identifies a broad class of functions called regular functions for which implicit representations provide simple update operators while expressing complex fixed-point mappings. These regular implicit operators gain expressive power with more test-time iterations and can represent more complex class of locally Lipschitz functions. The key assumption is that the domain is bounded and the function is locally Lipschitz on this bounded domain. Then, the paper states that the first iterate is a simple globally Lipschitz map but the subsequent iterates make the map more complex resulting in locally Lipschitz function. This increase in complexity is measured via increase in Lipschitz constant. This theory has been validated on three case studies: image reconstruction, Navier-stokes PDE (scientific computing), and linear programming (operations research). All the three cases demonstrate that increase in test-time iterations result in a better solution and the composition of the regular operator results in a more complex and expressive function.

**Strengths:**

1. The key findings are insightful to the community. Prior works have provided proof of universal approximation of implicit models such as deep equilibrium models. The key finding of this paper is the identification of the particular class of functions for which the operator  gains expressive power through additional test-time compute and can match more complex function class the includes singularities and unbounded Lipschitz behavior. This has been proved both theoretically and empirically.
2. The paper provides valid practical recommendations for implicit models which is to infuse the architecture for operator G with domain priors and constraints instead of constraining the network architecture itself. The paper demonstrates ways to incorporate these priors in the operator through examples in image reconstruction, Navier-stokes PDE and linear programming.

**Weaknesses:**

1. The paper demonstrates empirically that standard training results in such a regular implicit operator operator in some cases, but to the best of my understanding, it does not explore how or if the training guarantees this.
2. Another limitation of the theoretical result is the assumption of bounded domain. While the paper notes that this is not a major limitation for the results in the paper, how it affects other real-world problems is not well-understood.
3. Writing: Many parts of paper are dense and a bit difficult to read. It will be useful if proof sketches are included for the main theorems (especially 2.4 and 2.5).

**Questions:**

1. Some typos:
Line 69 - demonstrating -> deomonstrate
Line 395 - LInaer programm -> linear program

2. Line 947: $diam(X) + 1$ can be large, Why is it a meaningful upper bound? This uses the assumption that $X$ is bounded, but even then it can be really large.

---

> ### Author Response · Authors · 2025-11-23
> **Official Rebuttal: Part I**
>
> Thank you very much for your encouraging comments! Please find our responses below:
>
> > [W1] The paper demonstrates empirically that standard training results in such a regular implicit operator operator in some cases, but to the best of my understanding, it does not explore how or if the training guarantees this.
>
> During training, we did not use any means to enforce regularity. However, we empirical observed regularity in all of our training experiments. While we are happy to see this and excited to report it, we cannot provide a theoretical insight yet.
>
> Specifically, we assess regularity through two criteria: whether $\mathcal{G}(y,x)$ is globally Lipschitz with respect to $x$, and whether the fixed-point iteration $y_{t+1}=\mathcal{G}(y_t,x)$ successfully converges. **Regarding the dependency on $x$,** we observed that the first iterate $y_1(x)=\mathcal{G}(y_0,x)$ (where $y_0$ is constant) exhibits a mild Lipschitz constant—significantly smaller than that of the complex target mapping. This indicates that the learned operator tends to be simple and smooth with respect to the input. **Regarding the dynamics in $y$,** we monitored standard convergence metrics (PSNR for imaging and relative error for other tasks) and observed that the sequence $y_t$ converges. While these are empirical observations rather than theoretical guarantees, these two findings closely align with and support the key properties defined in regularity.
>
> Many standard techniques for enforcing global contraction (e.g., spectral norm constraints ensuring **uniform** Lipschitz < 1) can significantly restrict model expressivity, e.g., excluding mappings with large local Lipschitz constants. There is a tension between overly-strict regularity guarantees and expressivity. It is an interesting future research direction to discover how to enforce weaker, just-enough constraints without sacrificing models' adaptive capacity.
>
> Manuscript updates:
> * We now state **at the beginning of Section 3** that there is no use of any means to enforce regularity during training; that said, we numerically verified regularity in the experiments of image debluring, steady state NS equations, and linear programs.
> * In **Appendix F**, we explain why standard methods for guaranteeing regularity are too strict and may compromise model expressivity. We also discuss the specific technical challenges involved and outline potential roadmaps for developing new training techniques that can guarantee regularity without sacrificing adaptive capacity.
>
> > [W2] Another limitation of the theoretical result is the assumption of bounded domain. While the paper notes that this is not a major limitation for the results in the paper, how it affects other real-world problems is not well-understood.
>
> We are pleased to report that in the revised manuscript, **we have refined our proof techniques to completely remove the bounded domain assumption.** Theorems 2.4 and 2.5 now hold for any subset $\mathbb{X} \subset \mathbb{R}^d$, regardless of whether it is bounded.
>
> **How do we remove the assumption?** We adopted a "divide-and-conquer" spatial partitioning strategy. We discretize the entire space $\mathbb{R}^d$ into a grid of bounded cubic regions. Since our original theory holds for bounded domains, we construct the required functions locally within each cubic cell. We then seamlessly concatenate these local functions together using a smooth partition of unity to form a global function valid over the entire unbounded domain. The complete, updated proofs are highlighted in blue text in Appendices A.1 and A.2.
>
> **Why can we remove the assumption?**  The target function $\mathcal{F}(x)$ and the fixed-point map $y_*(x)$ are locally Lipschitz. This definition implies that the function's behavior is constrained only within local neighborhoods, not by a single global constant. Consequently, we can solve the construction problem piece-by-piece on bounded patches. Since the property we are preserving is local, ensuring it holds within every finite grid cell guarantees it holds globally for the union of those cells.

---

> ### Author Response · Authors · 2025-11-23
> **Official Rebuttal: Part II**
>
> > [W3] Writing: Many parts of paper are dense and a bit difficult to read. It will be useful if proof sketches are included for the main theorems (especially 2.4 and 2.5).
>
> To aid reader understanding without interrupting the narrative flow of the main text, we have added a dedicated "Intuition and Proof Sketch" section at the beginning of Appendix A.
>
> "The core intuition behind our proofs is an extension of the $1/x$ example discussed in the introduction.
>
> "For Theorem 2.4 (Sufficiency), we construct the implicit operator $\mathcal{G}$ as a dynamic interpolation: $\mathcal{G}(y,x) = (1-\varepsilon(x))y + \varepsilon(x)\mathcal{F}(x)$, which iteratively pulls the state $y$ toward the target $\mathcal{F}(x)$ with a step size $\varepsilon(x)$. The key theoretical innovation is making this step size adaptive: we construct $\varepsilon(x)$ to be inversely proportional to the local steepness (Lipschitz constant) of the target $y^*(x)$. In regions where the target function becomes extremely steep or singular (like $x \to 0$ for $1/x$), our constructed $\varepsilon(x)$ naturally vanishes. This effectively "slows down" the dynamics, ensuring the operator $\mathcal{G}$ itself remains globally smooth and contractive.
>
> "Theorem 2.5 (Necessity) establishes the converse: we show that for any regular operator, the local steepness of the fixed point is mathematically bounded by the operator's parameters ($y$-contraction modulus $\mu(x)$); and the fixed point map $y_\ast(x)$ can only become singular if the convergence rate slows down (contraction modulus $\to 1$), perfectly matching the mechanism used in our sufficiency construction."
>
> > [Q1] Some typos: Line 69 - demonstrating -> deomonstrate Line 395 - LInaer programm -> linear program
>
> Fixed in the revised paper.
>
> > [Q2] Line 947: $diam(X)+1$ can be large, Why is it a meaningful upper bound? This uses the assumption that $X$  is bounded, but even then it can be really large.
>
> In the revised manuscript, **we have refined our proof technique to eliminate this dependence.** By adopting the spatial partitioning strategy described in our response above, we decompose the domain into fixed-size cubic regions. Consequently, the term $\text{diam}(\mathbb{X})$ in the bound is replaced by the diameter of these local cubic cells (specifically a constant $3\sqrt{d}$), which is made independent of the global size of the domain. This ensures that the Lipschitz bounds remain tight, constant, and meaningful locally, regardless of how large the total domain $\mathbb{X}$ becomes.

---

### Author Response · Authors · 2025-11-23
**To all reviewers: Our revised paper has been uploaded**

Dear Reviewers,

We sincerely thank you for your thoughtful comments and constructive feedback. We have uploaded a revised version of the paper, where all major revisions are highlighted in blue.

Based on your suggestions, we have implemented significant technical improvements:
* **New LLM Experiment:** We added a completely new case study on LLM reasoning (Section 3.4), extending our validation to discrete domains. This demonstrates how the model utilizes test-time iterations to distinguish semantic nuances (e.g., "charge" in physics vs. finance). The associated test dataset is accessible via the anonymous link provided in the footnote; alternatively, it is available in the supplementary material.
* **Scaled-up Comparisons:** We significantly expanded the experiments in image reconstruction and scientific computing (Appendix G & H). We now compare against much larger explicit baselines (up to 16x deeper and 10x more parameters), confirming that the expressive advantage of implicit models cannot be matched simply by scaling up explicit model size.
* **Stronger Theoretical Results:** We improved our proof techniques to remove the boundedness assumption on the domain $\mathbb{X}$ (Theorem 2.4, 2.5, and Appendix A), thereby establishing our characterization of expressivity in a more general setting.

Additionally, we have incorporated the following modifications:
* Clarified our positioning and novel contributions.
* Clarified terminologies and elaborated on technical implementation details.
* Corrected typos and addressed specific requests raised by individual reviewers.

We believe these revisions and our detailed individual responses address your concerns. We would greatly appreciate it if you could consider raising your assessment score in light of these improvements.

Thank you again for your time and effort in reviewing our work.

Best regards,
The Authors

---

### Public Comment · ~Jialin_Liu1 · 2026-03-01
**Minor Title Update for Camera-Ready**

As we finalize our camera-ready submission, we are slightly updating the paper title from "Implicit Models: Expressive Power Scales with Test-Time Compute" to "Expressive Power of Implicit Models: Rich Equilibria and Test-Time Scaling".

This minor refinement is to further address Reviewer Ybb3's concern and better reflect the core theoretical focus of our work (specifically, Main Theorems 2.4 and 2.5). The scope of the paper remains exactly the same.

Best regards,
Authors

---

### Meta-Review · Area_Chair_uagt · 2026-01-12

**Summary:**

The main concerns raised during review centered on (i) the novelty and interpretation of the theoretical contributions (in particular whether Theorems 2.4 and 2.5 go beyond classical fixed-point theory), (ii) the conceptual distinction between expressivity of the equilibrium map versus solver approximation error, (iii) terminology and framing of “implicit models” and “test-time compute,” and (iv) the scale and breadth of experimental validation beyond applied mathematics benchmarks.

Three reviewers (FMZf, xk5a, KADN) viewed the paper as making a strong and novel contribution, especially in providing a sharp characterization of the function class representable by regular implicit models and in articulating a mechanism by which expressivity can scale with iterations without increasing parameters.

Two reviewers (Ybb3, La71) questioned novelty and argued that the observed gains primarily reflect solver accuracy or restate classical results.

The rebuttal and revision directly addressed most technical and experimental concerns, added substantial new material (notably the removal of the bounded-domain assumption in theory and a new LLM reasoning case study), and clarified the precise scope and interpretation of the claims, leaving the remaining disagreement mostly conceptual rather than factual.

**Reviewer Concerns:**

Addressed by the rebuttal and revision:

- Bounded domain assumption (FMZf): Fully resolved. The authors removed the bounded-domain requirement by refining the proofs using a partition-of-unity argument (Appendix A), strengthening Theorems 2.4 and 2.5.
- Training guarantees for regularity (FMZf): Adequately clarified. The authors explicitly state that regularity is empirically observed rather than theoretically guaranteed, and they discuss why stronger enforcement would harm expressivity (Appendix F).
- Terminology and definition of implicit models (Ybb3): Clarified in detail. The revision cleanly separates model definition (fixed-point map) from numerical solution (unrolling vs. implicit differentiation) and documents which solvers are used in each case study.
- Experimental scale and domains (Ybb3, La71, KADN): Substantially addressed. The authors added large-scale comparisons where explicit baselines are scaled to memory limits (Appendices G/H) and introduced a new LLM reasoning case study (Section 3.4), directly responding to requests for broader validation.
- Comparison to explicit networks (xk5a): Explicitly addressed in the Introduction and Appendix J, clarifying when explicit networks can represent locally Lipschitz targets and contrasting parameter-scaling versus iteration-scaling.
- Relation to Banach and classical theory (KADN, La71): Clarified with precise distinctions. The authors explain that their results address an inverse problem (existence of an operator realizing a given target map) and non-uniform contraction, which are not covered by standard Banach-style results.
- Perceived mathematical novelty (La71): The authors provided extensive arguments and references explaining why the main theorems are not covered by classical fixed-point literature; the reviewer did not point to actual references where the proposed theorems were already stated.


Concern that remains

- Conceptual interpretation of “expressivity scaling with test-time compute” (Ybb3): Despite detailed responses, this reviewer maintains that iteration-dependent improvements primarily reflect solver approximation error rather than increased expressivity of the equilibrium map. This is a fundamental interpretational disagreement rather than a gap in the technical development. In my view, most implicit network papers would align with the author's vision, that the networks approximately solve a fixed-point equation.

**Reviewer Scores:**

I don't think the reviewers with an 8 would change their score.

La71 might increase their score since the authors convincingly explain the novelty of the maths content of this work.

Ybb3 would not increase their score as stated.

---

### Decision · Program_Chairs · 2026-01-26

Accept (Poster)